

# Spatiotemporal Variability of Snow Depth across the Eurasian Continent from 1966 to 2012

Xinyue Zhong[1, 2, 4], Tingjun Zhang[3], Lei Zheng[5], Yuantao Hu[3], Huijuan Wang[3],

Shichang Kang[2, 6]

[1] *Cold and Arid Regions Environmental and Engineering Research Institute, Chinese Academy of*

*Sciences (CAS), Lanzhou 730000, China*

[2] *State Key Laboratory of Cryosphere Science, Cold and Arid Regions Environmental and*

*Engineering Research Institute, CAS, Lanzhou 730000, China*

[3] *Key Laboratory of Western China's Environmental Systems (Ministry of Education), College of*

*Earth and Environmental Sciences, Lanzhou University, Lanzhou 730000, China*

[4] *Key Laboratory of Remote Sensing, Gansu Province, Lanzhou 730000, China*

[5] *Chinese Antarctic Center of Surveying and Mapping, Wuhan University, Wuhan 430079, China*

[6] *CAS Center for Excellence in Tibetan Plateau Earth Sciences, Beijing 100101, China*

*Correspondence to:* T. Zhang (tjzhang@lzu.edu.cn)

**ABSTRACT**

Snow depth is one of key physical parameters for understanding the land surface energy

balance, soil thermal regimes, regional- and continental-scale water cycles, as well as assessing

water resources. In this study, snow depth climatology and spatiotemporal variations were

investigated using the long-term (1966-2012) ground-based measurements from 1814 stations

across the Eurasian continent. Spatially, mean snow depths of >20 cm were recorded in

northeastern European Russia, the Yenisey River basin, Kamchatka Peninsula, and Sakhalin.

Annual mean and maximum snow depth increased significantly during 1966-2012. Seasonally,

monthly snow depth decreased in autumn, and increased in winter and spring over that period of

time. Regionally, snow depth increased dramatically in the areas north of 50 °N. Compared with

air temperature, snowfall had more influence on snow depth and snow water equivalent during



1    November through March across the former Soviet Union. This study provides a baseline for

2    changes in snow cover, which are significant in climate system changes over the Eurasian

3    continent.



## 1 Introduction

Snow cover is a key part of the cryosphere, which is a critical component of the global climate system. Changes in snow cover serve as indicators of climate change because of its interactions and feedbacks with surface energy and moisture fluxes, hydrological processes, and atmospheric and oceanic circulation (Brown and Goodison, 1996; Armstrong and Brown, 2008; King et al., 2008). Snow depth, snow water equivalent (SWE) and snow density are also important parameters for water resource assessment, hydrological and climate model inputs and validation (Dressler et al., 2006; Lazar and Williams, 2008; Nayak et al., 2010).

Snow depth is a basic and important parameter of snow cover, which can provide additional information related to climate, surface energy balance, soil temperature, moisture budgets, spring runoff, water supply, and human activity (Sturm et al., 2001; Zhang, 2005; AMAP, 2011). Although snow cover extent reduced with climate warming, snow depth still increased in northern Eurasia (Kitaev et al., 2005; Bulygina et al., 2011). This is due to changes in the atmospheric moisture budget altering the atmospheric circulation, the warmer air led to greater moisture supply for precipitation as snowfall in winter (Ye et al., 1998; Kitaev et al., 2005; Rawlins et al., 2010). Meanwhile, snowmelt from increased snow depth may also lead to higher soil moisture in spring, which promotes enhanced precipitation with increased evapotranspiration (Groisman et al., 1994).

Snow depth is an important factor controlling the ground thermal regime (Goodrich, 1982; Zhang et al., 1996, 1997; Zhang, 2005). Kudryavtsev (1992) investigated that thin snow cover results in cooler soil surface, whereas thick snow cover leads to a warmer soil surface. Frauenfeld et al. (2004) indicated that the maximum snow depth by the end of winter has a significant influence on the active layer depth during the following summer. As an important parameter, snow depth was included in a surface energy balance-based one-dimensional heat transfer model for estimating the thermal regime of soil (Ling and Zhang, 2004, 2005). The numerical modeling results showed that the rate of mean annual ground surface temperature increase with the increasing maximum snow depth was about 0.1 ℃ cm$^{-1}$ for the



maximum snow depth at 15 cm. Over the Alaskan Arctic coastal plain, mean annual
ground surface temperature increased with snow depth. However, the rate of the mean
annual ground surface temperature increase fell dramatically for snow depth greater
than 40 cm (Zhang, 2005).
Furthermore, snow accumulation is one of the important freshwater resources
and has direct impact on the hydrological cycle. Snowmelt runoff in spring is a major
source of river recharge and water supply, on the other hand, snowmelt floods are of
great importance, threatening the ecological and human security (Li, 1988).
Approximately 95 % of water resources are derived from snowmelt in spring and
early summer in alpine and Arctic areas; in addition, in these areas, half or more of
floods are caused by melting snow (AMAP, 2011). Adam et al. (2009) suggested that
the variations of snow depth will significantly affect the hydrological regime of the
Arctic in the future.
Using in-situ observational data from meteorological stations and satellite remote
sensing data, several studies have documented changes in snow depth over the
Northern Hemisphere, demonstrating that snow depth varies regionally: overall, the
annual mean snow depth decreased in most areas over North America (Brown and
Braaten, 1998; Dyer and Mote, 2006), and increased in Eurasia and the Arctic (Ye et
al., 1998; Kitaev et al., 2005; Callaghan et al., 2011a; Liston and Hiemstra, 2011) but
there was regional differences (Bulygina et al., 2009, 2011; Ma and Qin, 2012;
Stuefer et al., 2013; Terzago et al., 2014). Changes in snow depth were primarily
affected by air temperature and precipitation. Ye et al. (1998) and Kitaev et al. (2005)
showed that higher air temperatures caused an increase in snowfall in winter, thus
greater snow depth was observed in northern Eurasia in response to global warming.
Furthermore, snow depth distribution and variation are also controlled by terrain (i.e.,
elevation, slope, aspect, and roughness) and vegetation (Lehning et al., 2011;
Grünewald et al., 2014; Revuelto et al., 2014; Rees et al., 2014; Dickerson-Lange et
al., 2015). Snow depth is closely related to other climatic variables such as the North
Atlantic Oscillation /Arctic Oscillation (NAO/AO) index. Beniston (1997) found that
the NAO played a crucial role in fluctuations in the amount of snowfall and snow





depth in the Swiss Alps from 1945 to 1994. Kitaev et al. (2002) reported that the
NAO index is positively related to snow depth in the northern part of the East
European Plain and over western Siberia; however, the NAO is negatively correlated
with snow depth in most southern regions of northern Eurasia. You et al. (2011)
indicated that there is a positive relationship between snow depth and the winter
AO/NAO index and Niño-3 region sea surface temperature (SST) in the eastern and
central Tibetan Plateau (TP) from 1961 through 2005.
In order to obtain a wider range of snow depth, researchers have used different
instruments (e.g., LIDAR, airborne laser scanning (ALS), and unmanned aerial
systems (UASs)) (Hopkinson et al., 2004; Grünewald et al., 2013; Bühler et al., 2016)
or have developed and improved the algorithms with passive microwave (Foster et al.,
1997; Derksen et al., 2003; Grippaa et al., 2004; Che et al., 2016). Although these
observations can mitigate the regional deficiency of in-situ snow depth observations,
the satellite data have low spatial resolution ($25 \times 25$ km) and the accuracy is always
affected by clouds, underlying surface conditions, and inversion algorithms; in
addition, data acquisition from the large airborne equipment or aerial systems is
always costly and some of them need to obtain official permission before using in
some countries. Ground-based snow measurement is the basis for verification of
remote sensing and instrumental data, which can provide more accurate and
longer-time-series information, and it is important for investigating climatology and
variability of snow depth.
During winter, the average maximum terrestrial snow cover is nearly $47 \times 10^6$
km$^2$ over Northern Hemisphere lands (Robinson et al., 1993; IGOS, 2007). A large
fraction of the Eurasian continent is covered by snow during the winter season, and
some areas are covered by snow for more than half a year. There are long-term and
large-scale snow cover measurements and observations across the Eurasian continent,
with the first snow cover record dating back to 1881 in Latvia (Armstrong, 2001).
These measurements provide valuable data and information for snow cover phenology
and snow cover change detection. In Eurasia, most studies of snow depth have mainly
focused on Russia (Ye et al., 1998; Kitaev et al., 2005; Bulygina et al., 2009, 2011),



the former Soviet Union (USSR) (Brasnett, 1999), and the TP (Li and Mi, 1983; Ma
and Qin, 2012). However, due to the lack of data and information, there has been no
integrated and systematic investigation of changes in snow depth across the entire
Eurasian continent using ground-based measurements. Using data from ground-based
measurements, the objective of this study is to provide a detailed description of snow
depth and to investigate the climatology and variability of snow depth as well as its
relationships with other topography and climate factors over the Eurasian continent
from 1966 to 2012. This study can provide basic information on climate system
changes in the region. The dataset and methodology are described in Section 2, with
the results, discussion, and conclusions presented in Sections 3, 4, and 5, respectively.
**2   Data and Methodology**
Measurements of daily snow depth were conducted at 1103 meteorological
stations in 17 countries on the Eurasian continent from 1881 to 2013 (Table 1). Snow
depth was measured at these stations on daily basis. Snow course data over the former
USSR were also used in this study from historical records from 1966 to 2011. Snow
course data include routine snow surveys that run throughout the accumulation season
(every 10 days) and during snowmelt (every 5 days) period over the former USSR.
Snow surveys were conducted for 1–2 km in both forest and open terrain around each
station. Snow depth was measured each 10 m in the forest, and each 20 m in open
terrain (Bulygina et al. 2011).
SWE is also an important parameter of snow cover that is usually used in
hydroclimate research. In this study, we analyzed the relationships among SWE, air
temperature, snowfall and snow depth during the accumulation season (from
November to March) over the former USSR. SWE was measured every 100 m at the
0.5-1.0 km courses and every 200 m at the 2 km course (Bulygina et al., 2011).
Precipitation data were divided proportionally into daily solid and liquid data, and the
solid-to-liquid fraction was determined according to daily mean temperature (Brown,
2000). The solid fraction of precipitation, $S_{rat}$, was estimated by the following
Equation (1):



$$S_{rat} = \begin{cases} 1.0 & for\ T_{mean} \leq -2.0°C, \\ 0.0 & for\ T_{mean} \geq +2.0°C, \\ 1.0 - 0.25(T_{mean} + 2.0) & for\ -2.0°C < T_{mean} < +2.0°C. \end{cases} \quad (1)$$

where $T_{mean}$ is the mean daily air temperature (℃).
Snow depth and SWE at each station were determined as the average value of a
series of measurements in each snow course survey (Bulygina et al., 2011). In
individual measurements, both random and systematic errors inevitably occur
(Kuusisto, 1984). To minimize these errors, quality control of the meteorological data
was undertaken prior to the datasets being stored at the Russian Research Institute for
Hydrometeorological Information-World Data Center (RIHMI-WDC) (Veselov, 2002).
We implemented a second quality control: (1) daily snow depth observations (equal to
or greater than 0 cm, not including missing data) for <15 days in one month were
omitted; (2) snow data from stations with <20 years of measurements during
1971-2000 were excluded; and 3) data exceeding two standard deviations compared
with the annual average value during 1966-2012 were omitted. In total, we used data
from 1814 stations to analyze the climatology and variability of snow depth over the
Eurasian continent (Fig. 1 and Table 1).
We defined a snow year as the period from July 1[st] of a current year to June 30[th]
of the following year. Because the procedures for taking snow observations had
changed in the past, there were some inhomogeneities in the data. However, there has
been no change in the observation procedure since 1965 (Bulygina et al., 2009).
Therefore, we used snow data for the snow years from 1966 to 2012 in this study. The
following variables were calculated for each station:
(1) Monthly mean snow depth: In this study, we defined a snow cover day with
snow depth equal to or greater than 0 cm according to the standard way for deriving
monthly mean snow depth in regular World Meteorological Organization (WMO)
climatological products. A threshold of 15 days was selected because the snow cover
duration in some areas of China was less than one month, and the data for 15 days'
snow depth in a month were relatively stable. The monthly mean snow depth was
computed as the arithmetic sum of daily snow depth divided by the number of days





with snow on the ground within each month.

2       In order to reflect the primary long-term spatial patterns of snow cover

distribution, we calculated the annual mean snow depth and annual mean maximum
snow depth during 1966-2012:

5       (2) Annual mean snow depth: the annual mean snow depth was calculated as the

arithmetic sum of the monthly mean snow depth divided by the number of available
snow months within each snow year. The annual mean snow depth was averaged from
the annual snow depth for ≥20 snow years during 1966-2012.

9       (3) Annual mean maximum snow depth: the annual mean maximum snow depth

was determined from the maximum daily snow depth in each snow year. It was
calculated using the average values of annual maximum snow depth from the stations
with ≥20 years of data during 1966-2012.
(4) Linear trend coefficient of snow depth: the linear trend coefficient of snow
depth for each station was the result of linear regression analysis with respect to time,
and was the rate of change in snow depth for a period of time. The rate of change in
snow depth was considered to be statistically significant at the 95 % level.
To overcome the systematic differences between stations related to
climate/elevation and station distributions, the anomaly of snow depth from the
long-term mean was used in this study. According to each 30 years as a climate
reference period, the annual mean snow depths of the period 1971-2000 were
computed as climate reference values in this study. We calculated the anomalies of
monthly, annual mean and maximum snow depth relative to the mean for the period
from 1971 to 2000 for each station and averaged the anomalies for all stations to the
anomalies for the whole Eurasian continent. Linear regression method was applied to
analyze the trend of the snow depth anomaly.
Wavelet analysis was performed to analyze the long-term variations of snow
depth. A wavelet is a wave-like oscillation with an amplitude that begins at 0,
increases, and then decreases back to 0. All wavelet transforms may be considered
forms of time-frequency representation for continuous-time (analog) signals and so
are related to harmonic analysis. Almost all practically useful discrete wavelet



transforms use discrete-time filter banks. These filter banks are called the wavelet and
scaling coefficients in wavelets nomenclature. These filter banks may contain either
finite impulse response (FIR) or infinite impulse response (IIR) filters. The wavelets
forming a continuous wavelet transform (CWT) are subject to the uncertainty
principle of Fourier analysis respective sampling theory: given a signal with some
event in it, one cannot assign simultaneously an exact time and frequency response
scale to that event. The product of the uncertainties of time and frequency response
scale has a lower bound. Thus, in the scale gram of a continuous wavelet transform of
this signal, such an event marks an entire region in the time-scale plane, instead of
just one point. Also, discrete wavelet bases may be considered in the context of other
forms of the uncertainty principle. This method is used to solve the problem of
recovering a true signal from indirect noisy data (Graps, 1995). We used an averaging
filter for wavelets analysis. Using this method, values that are too small or too large
may be excluded; however, the main features of the dataset are not significantly
affected. The wavelet coefficients obtained from filtering were used in an inverse
wavelet transformation to reconstruct the data set. The new data set was represented
as the smoothed lines of wavelet analysis in figures. Linear trend analysis of
anomalies was applied to obtain the temporal trends for the long-term period. The
linear trend coefficient of snow depth was calculated to represent the rate of change at
each station.

**3  Results**

**3.1 Climatology of Snow Depth**

The distributions of long-term mean snow depth generally represented the
latitudinal zonality: the snow depth for each station generally increased with the
latitude across the Eurasian continent (Fig. 2). The maximum annual mean snow
depth of 106.3 cm was observed in the west of the Yenisey River (dark blue circle)
(Fig. 2a). In contrast, the minimum values (~0.01 cm) were observed in some areas of
China (small gray circles) due to wind speed, topography, underlying ground surface,
and climatic conditions (Gray and Male, 1981; Sturm et al., 1995, 2001; Callaghan et



al., 2011b).

2       Annual mean snow depth for most areas in Russia was >10 cm. Depths were

even greater in the northeastern part of European Russia, the Yenisey River basin, the
Kamchatka Peninsula, and Sakhalin, with snow depths of >40 cm. The regions with
the smallest annual mean snow depth (<5 cm) were located in most areas of the
Caucasus Mountains. Snow depth in other areas of the former USSR was ~2-10 cm,
but shallow snow depths (no more than 1 cm) were observed in some southern regions
of Central Asia. The annual average snow depth in the central Mongolian Plateau was
lower than that in the northern areas, with values of no more than 5 cm. Snow depth
was >3 cm in the north of the Tianshan Mountains, Northeast China and some regions
of the southwestern TP. In the Altay Mountains and some areas of the northeastern
Inner Mongolia Plateau, annual mean snow depths were >5 cm.

13       Annual mean maximum snow depth also varied with the latitude (Fig. 2b), which

showed a spatial distribution pattern similar to the annual mean snow depth pattern.
The maximum value (~201.8 cm) was recorded in the same location as the greatest
annual mean snow depth. For the majority of Russia, the maximum snow depth
was >40 cm. The regions with the maximum snow depths (exceeding 80 cm) were
located in the northeastern regions of European Russia, the northern part of the West
Siberian Plain, the Yenisey River basin, the Kamchatka Peninsula, and Sakhalin;
however, along the coast of the Caspian Sea, the maximum snow depth was <10 cm.
Most of the rest of the former USSR had a maximum depth of >10 cm, except for
some regions of Ukraine and Uzbekistan. Maximum snow depth was >10 cm in
northern Mongolia, and 6–10 cm in the central and eastern parts of the country.
Maximum snow depths were higher over the northern part of the Xinjiang
Autonomous Region of China, Northeast China, and some regions of the eastern and
southwestern TP (>10 cm). The maximum snow depth in some areas was more than
20 cm. In other regions of China, the values were relatively small, ~8 cm or less.

28       Monthly mean snow depth varied across the Eurasian continent (Fig. 3). The

maximum monthly snow depths were recorded in northeastern European Russia,
northern part of the West Siberian Plain, the Yenisey River basin, the Kamchatka



Peninsula, and Sakhalin. The minimum values were observed in most areas of China.

2       In the autumn months (September to November), the snow depth was shallow

(Figs. 3a-c). Monthly mean snow depth was <20 cm in most areas of European Russia
and the south of Siberia, but ranged from ~20 cm to 40 cm in    northern Siberia and
the Russian Far East in November (Fig. 3c). Monthly mean snow depth was less than
5 cm in the north of Mongolia and most regions across China. From December to
February, the snow depth increased and the areas covered by snow expanded
significantly (Figs. 3d-f). Most monthly snow depth values were >20 cm over the
former USSR. Monthly mean snow depth was still <1 cm in most regions of China,
but more than 10 cm in the northern Xinjiang Autonomous Region of China,
Northeast China, and some regions of southwestern TP. The snow depth was even
more than 20 cm in some places of the Altai Mountains. In spring months, the snow
cover areas decreased significantly (Figs. 3g–i). However, the monthly mean snow
depth still exceeded 20 cm in most areas of Russia. Snow cover areas and snow depth
gradually decreased in April and May. Snow cover was observed only in Russia and
the TP in June (Fig. 3j).
**3.2 Variability of Snow Depth**
There were long-term significant increasing trends in the annual mean and
maximum snow depth from 1966 to 2012 over the Eurasian continent as a whole with
the increasing rate of snow depth of 0.2 cm decade$^{-1}$ and 0.6 cm decade$^{-1}$, respectively
(Fig. 4). Both annual mean snow depth and maximum snow depth exhibited a similar
pattern of changes over the four decades, although the amplitude of the maximum
snow depth anomaly (about $\pm2$ cm) was much larger than that of the mean snow
depth anomaly (about $\pm1$ cm). From the mid-1960s to the early 1970s, the annual
mean snow depth decreased slightly, then increased until the late 1970s (Fig. 4a).
Thereafter, it fluctuated from the late 1970s to the early 1990s. Subsequently, the
annual mean snow depth increased steadily from the early 1990s through the early
2000s, then decreased sharply until 2012.
Maximum snow depth decreased by 2.5 cm from the mid-1960s through the





early 1970s (Fig. 4b). There was a sharp increase of 3.5 cm in the maximum snow
depth during the 1970s, then fluctuated from the late 1970s to the early 1990s. The
maximum snow depth increased again from the early 1990s through the early 2010s.

4         Statistically significant trends of variations in monthly snow depth occurred from

1966 through 2012 except for November, February, and May (Fig. 5). During the
snow cover formation period (October and November), the monthly snow depth
decreased slightly (Figs. 5a-b). There was a significant decrease trend of monthly
snow depth in October, with a rate of decrease of approximately 0.1 cm decade$^{-1}$ (Fig.
5a).

10        Inter-annual variations of monthly snow depth were more significant in the

winter months (Figs. 5c-e). Snow depth was below its long-term mean value from the
mid-1960s through the mid-1980s, and then it was above the long-term mean. There
were statistically significant increasing trends in monthly snow depth in January and
February, and similar inter-annual variations in snow depth for these two months
during the period from 1966 to 2012 (Figs. 5d, e). Monthly snow depth sharply
decreased by about 2 cm prior to the early 1970s, then increased by 2-2.5 cm until the
late 1970s. Monthly snow depth displayed a fluctuating increase from the late1970s
through 2012.

19        Significant increasing trend of monthly snow depth also appeared in March and

April, the rate of increase was about 0.6 cm decade$^{-1}$ and 0.3 cm decade$^{-1}$, respectively
(Figs. 5f-g). The trend of monthly snow depth in March was consistent with the
change in winter from the mid-1960s through the late 1970s, then it was stable until
the early 1990s (Fig. 5f). Monthly snow depth rapidly increased by 2.5 cm from the
mid-1990s through the late 1990s, then it decreased slightly. Snow depth presented
fluctuant trend during the mid-1960s through the early 1980s (Fig. 5g). Subsequently,
snow depth increased dramatically by about 3 cm from the mid-1980s to the early
2000s. It declined rapidly during the early 2000s through 2012.

28        Figure 6 shows the spatial distributions of linear trend coefficients of annual

mean snow depth and maximum snow depth for each station during 1966-2012, with



p≤0.05. The significant increasing trends (blue circles) of annual mean snow depth
occurred in most of European Russia, the south of Siberia and the Russian Far East,
the northern Xinjiang Autonomous Region of China, and Northeast China (Fig. 6a). In
contrast, decreasing trends (red circles) were detected in western European Russia,
some regions of Siberia, the north of Russian Far East, and some regions to the south
of 40 °N across China. Over the entire Eurasian continent, the most significant linear
variability trends in annual mean snow depth were observed in the region north of 50 °
N, indicating that the increasing rate of annual mean snow depth was greater in higher
latitude regions.
Changes in the maximum snow depth were similar to those in annual mean snow
depth in most of Eurasian areas from 1966 to 2012, but the change rates of the
maximum snow depth were greater than the values of annual mean snow depth (Fig.
6b). The significant increasing trends were observed in the same regions as those with
increases in annual mean snow depth. The decreasing trends were found in generally
the same locations as decreases in annual mean snow depth, with greater reductions in
the south of Siberia and the Russian Far East.
In October and November, there were few stations with significant changes in
snow depth (at the 95 % level) (Figs. 7a, b). The increasing trends were mainly
observed in most areas across the Eurasian continent in October. But the increasing
trends of snow depth only appeared in Siberia and the Russian Far East in November.
The decreasing trends in monthly mean snow depth occurred in the eastern regions of
European Russia, the southern areas of the West Siberian Plain, and some areas of the
northeast Russian Far East.
In winter months (December, January and February), there was a gradual
expansion in areas with monthly mean snow depth variation at the 95 % level (Figs.
7c–e). There were increasing trends of monthly mean snow depth in the eastern
regions of European Russia, southern parts of Siberia, the northern Xinjiang
Autonomous Region of China, and Northeast China. In contrast, significant
decreasing trends were observed in the north and west of European Russia, scattered
in Siberia, the northeast of the Russian Far East, and most areas of China.



From March to May, the number of stations with significant changes (at the 95 %
level) in monthly mean snow depth fell, especially in May because of snow melt (only
78 stations) (Figs. 7f-h). Changes in monthly mean snow depth were consistent with
the trends in winter over the former USSR but more stations with the decreasing
trends in the southern Siberia. There were few stations with statistically significant
trends of snow depth across China and monthly snow depths tended to decrease in
most stations. Compared with the south of 50 °N, the changes in monthly mean snow
depth were more significant to the north of 50 °N.
**3.3 Variability of Snow Depth with Latitude and Elevation**

11       To explore the spatial features of snow depth, we conducted a linear regression

analysis of annual mean snow depth with latitude and elevation (Fig. 8). Snow depth
is positively correlated with latitude, i.e., snow depth generally increases with latitude
(Fig. 8a). The increase rate of snow depth was about 0.81 cm per 1 °N. We detected a
closer relationship between latitude and mean snow depth to the north of 40 °N (Figs.
8a, c). In these regions, snow cover was relatively stable (the number of annual mean
continuous snow cover days was more than 30) (Zhang and Zhong, 2014), in which
snow cover was easier to accumulate by the heavy snowfall and more difficult to melt
with low air temperature.

20       There was a negative correlation between snow depth and elevation across the

Eurasian continent (Fig. 8b): with every 100 m increase in elevation, snow depth
decreased by ~0.5 cm (P≤0.05). Annual mean snow depth was less than 1 cm in most
areas, with an elevation greater than 2000 m, because a  snow depth of 0 cm was
used to calculate the mean snow depth. Therefore, although the TP is at high elevation,
the shallow snow depth in this area resulted in the generally negative correlation
between snow depth and elevation across the Eurasian continent. However, we also
determined that snow depth increased with elevation in most regions north of 45 °N
(Fig. 8c). This result indicates that elevation is an important factor affecting snow
depth in these regions.





### 3.4 Relationships among Snow Depth, SWE, Air Temperature and Snowfall

Variations in snow depth are closely related to climate change. To examine the relationship between snow depth and climatic factors, we calculated the long-term mean snow depth, air temperature and snowfall of 386 stations from November through March across the USSR (Fig. 9). The period (snow cover years) spanned from 1966 through 2009 because data on air temperature and precipitation were recorded only until 2010. Snow depth significantly decreased with increasing air temperature ($P \leq 0.05$), but the Goodness of Fit of the relationship was only 16% (Fig. 9a). Compared with the air temperature, snowfall exhibited a better relationship with snow depth (Fig. 9b). The mean snow depth was less than 20 cm in most stations with the accumulated snowfall being <50 mm from November through March. It increased with the accumulated snowfall increased, and the thickest snow depth reached 120 cm when the maximum cumulative snowfall was 350 mm.

Comparing the long-term inter-annual trends of changes in snow depth, SWE, air temperature and snowfall, the variability of snow depth and SWE were mainly affected by the changes in snowfall. Overall, the trends in long-term air temperature, precipitation, snowfall and SWE displayed increases from November to March (Fig. 10). This was because the increase precipitation fell as snow in cold areas where the increased temperature was still below freezing (Ye et al., 1998; Kitaev et al., 2005). Warmer air led to greater supply of moisture for snowfall, hence the snow accumulation still increased (Ye et al., 1998). The significant increasing snowfall can explain the sudden drop in snow density from the mid-1990s through the early 2000s (Zhong et al., 2014): fresh snow with low snow density. There were basically consistent trends of variations in snow depth, SWE and snowfall accumulation from November through March during 1966-2009 (Figs. 10b-d). The results indicated that the increasing trend of changes in snow depth was the combined effect of the increasing air temperature and snowfall. In fact, the climatology of snow depth not only influenced by air temperature and precipitation, but also with other climatic factors and atmospheric circulation. The mechanism of increasing snow depth in the Eurasian continent requires further investigation in the future.





## 4   Discussion

### 4.1 Comparison with Previous Results

Comparing our results with previous research across the Eurasian continent, we found that the climatology of mean snow depth was basically consistent with that described in the previous studies in China (Ma and Qin, 2012), but was higher than that in northern Eurasia (Kitaev et al., 2005; Bulygina et al., 2011). These differences may result from differences in the time frame of data collection, number of stations, calculation methods, and data quality control. For example, Kitaev et al. (2005) reported a historical record of snow depth spanning the period from 1936 to 2000, with the onset and end of the snow year earlier than the definition used in this study. Nevertheless, the distributions of high snow depth in the two studies were located in the same regions and the regional and continental inter-annual and inter-decadal variations were consistent.

Previous research found that historical winter snow depth increased in most areas (30-140 °E, 50-70 °N), with the exception of European Russia, during 1936-1983 (Ye et al., 1998), similarly to our results. However, in the present study, we found that decreasing trends also appeared in some regions of the southern portion of western and central Siberia. The time sequence of observations may be the main reason for this difference. Compared with our study, the areas with increasing trends in snow depth reported by Ma and Qin (2012) were larger in China. Snow depth increased significantly in the northeastern TP in their results. The differences may have been caused by the different statistical methods and interpolation of nearby stations in the study of Ma and Qin.

In addition to the above reasons, these differences can be explained by the changes in climatic factors during different periods. The sensitivity of snow cover to air temperature and precipitation for each station showed regional differences (Fallot et al., 1997; Park et al., 2013). The amount of snowfall can be affected by climate change, and leading to differences in snow depth at different times (Ye et al., 1998; Kitaev et al., 2005).



## 4.2 Topographical effects in snow depth

Some important questions that are not addressed in the current research should

be resolved in the future. Topography is an important factor affecting the climatology

of snow depth, and    is the main reason causing the inhomogeneity of data. Previous

studies have analyzed the representation of snow depth for single stations to solve the

issue (Grünewald and Lehning, 2011, 2013; Grünewald et al., 2014). However, in the

present study, we did not discuss this question because of the complexity of spatial

difference. This issue should be addressed in future studies. Variations in snow depth

are significantly affected by the local climate factors. Therefore, we will select a

typical climate zone to research the climatology and variations of snow cover.

Furthermore, as there are few stations in high-latitude regions, southern Mongolia, the

basin areas of the southern Tianshan Mountains and the northwest of TP, collection of

additional data and comprehensive field measurements is required.

## 5   Conclusions

In this study, daily snow depth and snow course data from 1814 stations were

used to investigate spatial and temporal changes in annual mean snow depth and

maximum snow depth over the Eurasian continent for the period from 1966 to 2012.

Our results demonstrate that greater long-term average snow depth was observed in

northeastern European Russia, the Yenisey River basin, the Kamchatka Peninsula, and

Sakhalin. In contrast, the shallowest snow depths were recorded in China, except for

the northern Xinjiang Autonomous Region of China, Northeast China, and in some

regions of southwestern TP.

There were statistically significant trends of variations in long-term snow depth

over the Eurasian continent as a whole. A similar increase pattern of changes was

exhibited in both annual snow depth and maximum snow depth, although the

amplitude of the maximum snow depth anomaly was much larger than the equivalent

value for mean snow depth. Monthly snow depth in autumn presented decreasing

trend, while there were increasing trends of variations of snow depth during winter



and spring.

2        Significant increasing trends in snow depth were detected in the eastern regions

of European Russia, the southern Siberia, the Russian Far East, northern areas of the
Xinjiang Autonomous Region of China, and northeastern China. Decreasing linear
trends were observed in most western areas of European Russia, some regions of
southern Siberia, the northeastern Russian Far East and most areas in the southern
40 °N across China.

8        Compared with elevation, latitude played a more important role in the snow

depth climatology. The variations in snow depth and SWE were more affected by
snowfall: the greater the snowfall accumulation, the thicker the snow depth and SWE.
The mechanism controlling the increase in snow depth and the effects of topography
on snow depth will be addressed in future studies.
*Acknowledgements.* We express our gratitude to the researchers who assembled and
digitized the snow depth data at meteorological stations and snow surveys across the
Eurasian continent over a period of >40 years. This work was funded by the National
Key Scientific Research Program of China (2013CBA01802), the Open Foundation
from the State Key Laboratory of Cryospheric Sciences (SKLCS-OP-2016-12), the
Project for Incubation of Specialists in Glaciology and Geocryology of the National
Natural Science Foundation of China (J1210003/ J0109), and the Foundation for
Excellent Youth Scholar of Cold and Arid Research Environmental and Engineering
Research Institute, Chinese Academy of Sciences.



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

| Dataset | Spatial distribution | Number of stations | Source |
|---|---|---|---|
| Daily snow depth | the former USSR | 586 | Russian Research Institute for Hydrometeorological Information-World Data Center (RIHMI-WDC) National Snow and Ice Data Center (NSIDC), University of Colorado at Boulder |
| | China | 492 | National Meteorological Information Center (NMIC) of the China Meteorological Administration |
| | Mongolia | 25 | NSIDC |
| Snow depth from snow course | the former USSR | 1044 | RIHMI-WDC, NSIDC |
| Snow water equivalent (SWE) | the former USSR | 386 | RIHMI-WDC |
| Daily air temperature and precipitation | the former USSR | 386 | RIHMI-WDC |





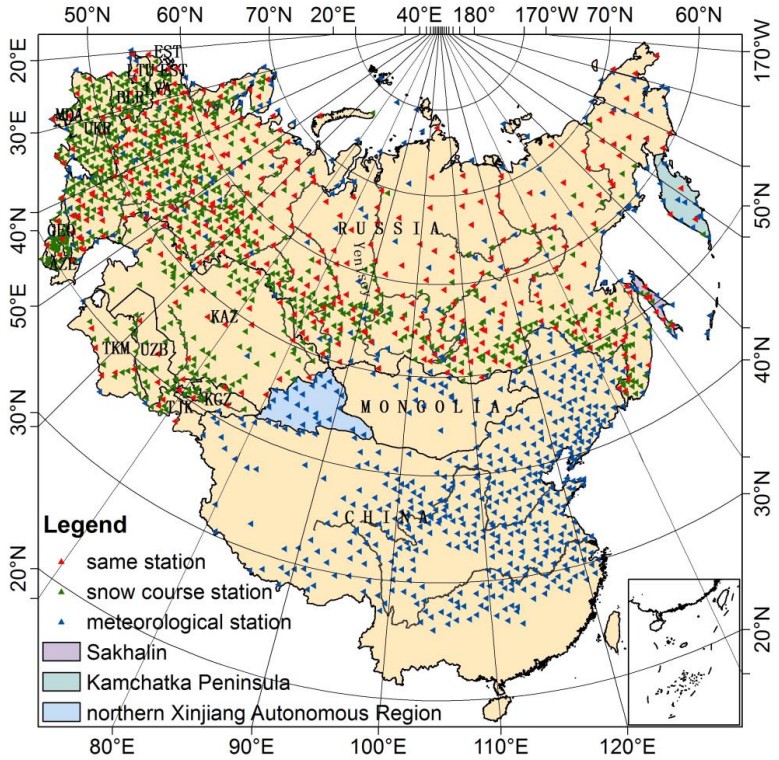

**Figure 1.** Geographical locations of meteorological and snow course stations across the Eurasian continent. The
red triangles represent stations where snow depth was measured at both meteorological stations and snow course
surveys, the green triangles show stations where snow depth was measured at snow surveys only, and the blue
triangles show stations where snow depth was measured at meteorological stations only.





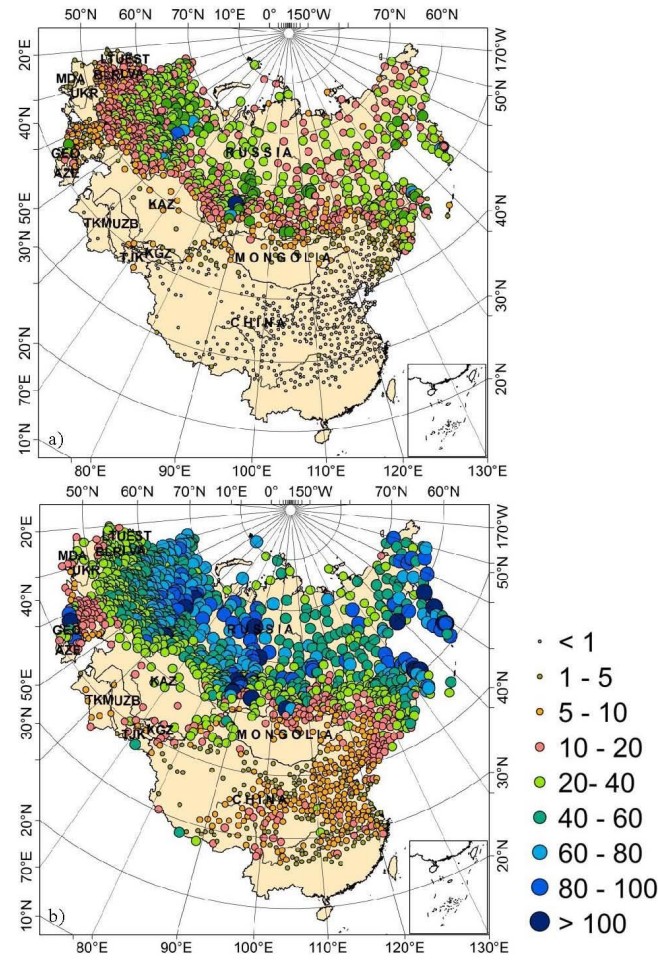

**Figure 2.** Annual mean snow depth (a) and maximum snow depth (b) across the Eurasian continent (cm) during 1966-2012.



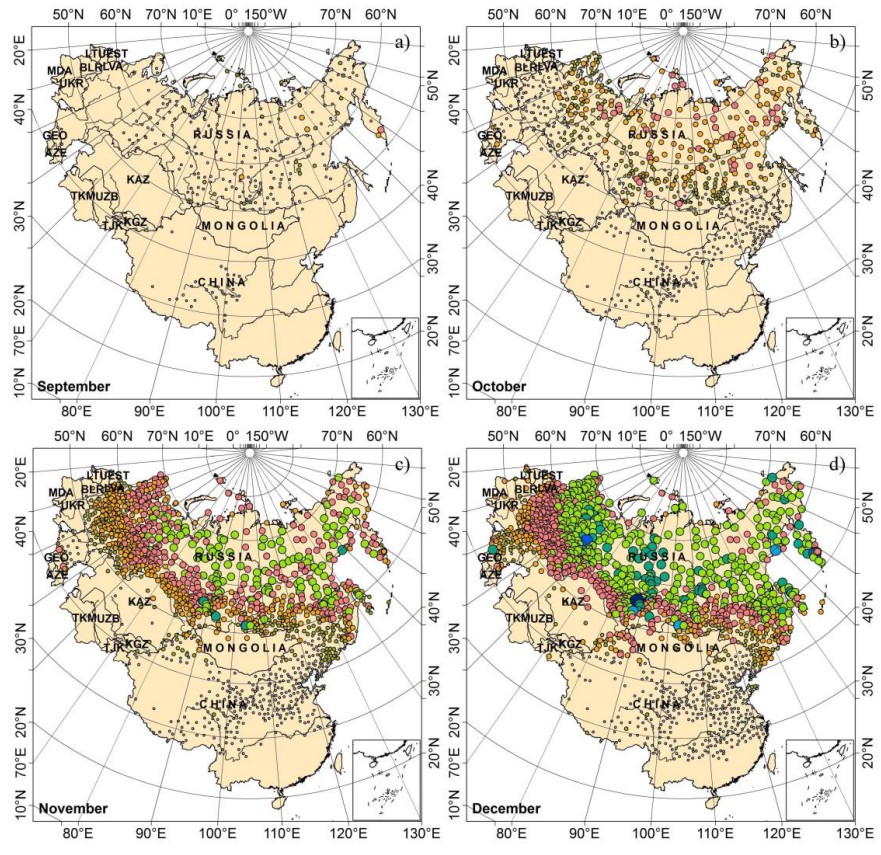



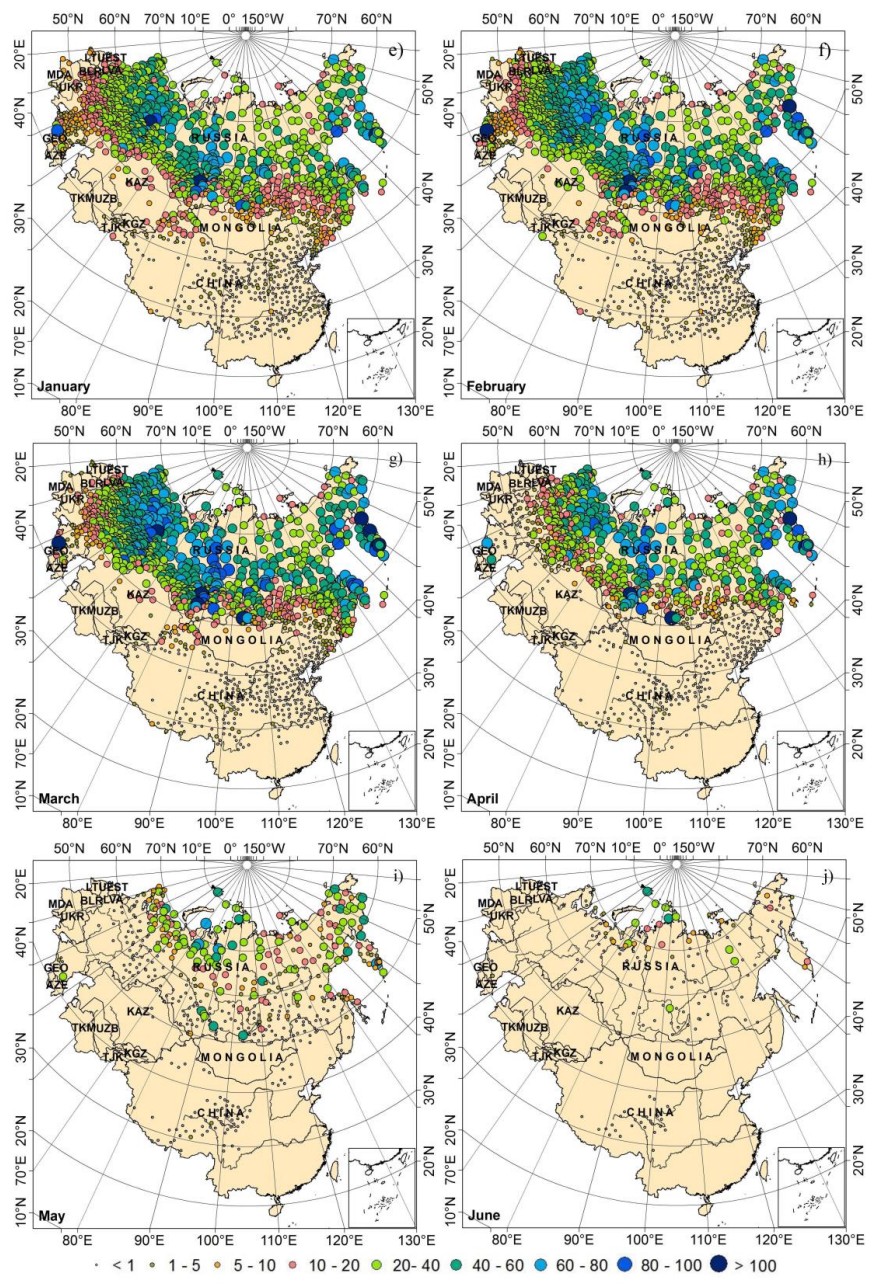

**Figure 3.** Monthly mean snow depth (from September to June) (cm) across the Eurasian continent (cm) during
1966-2012. (a) September, (b) October, (c) November, (d) December, (e) January, (f) February, (g) March, (h)
April, (i) May, (j) June.





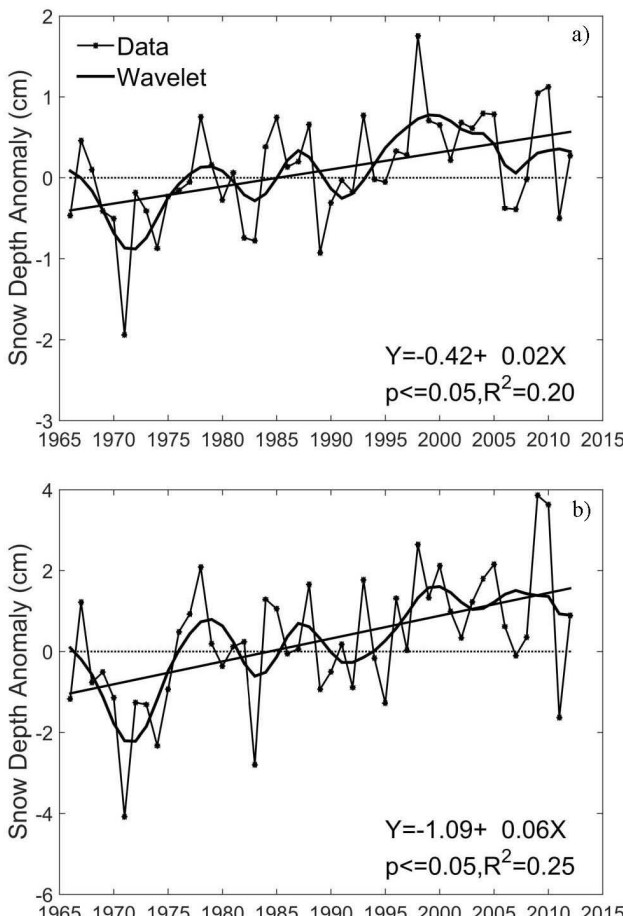

**Figure 4.** Composite of inter-annual variation of annual mean snow depth (a) and maximum snow depth (b) from
1966 through 2012 with respect to the 1971-2000 mean across the Eurasian continent. The line with dots is the
anomaly of snow depth; the thick curve represents the smoothed curve using wavelet analysis; the thick line
presents a linear regression trend. Y represents snow depth anomaly in cm and X represents time in snow cover
years, 1966 was the first snow cover year, therefore, X ranged from year 1 (1966) to year 47 (2012) in the
simulation of annual mean snow depth.





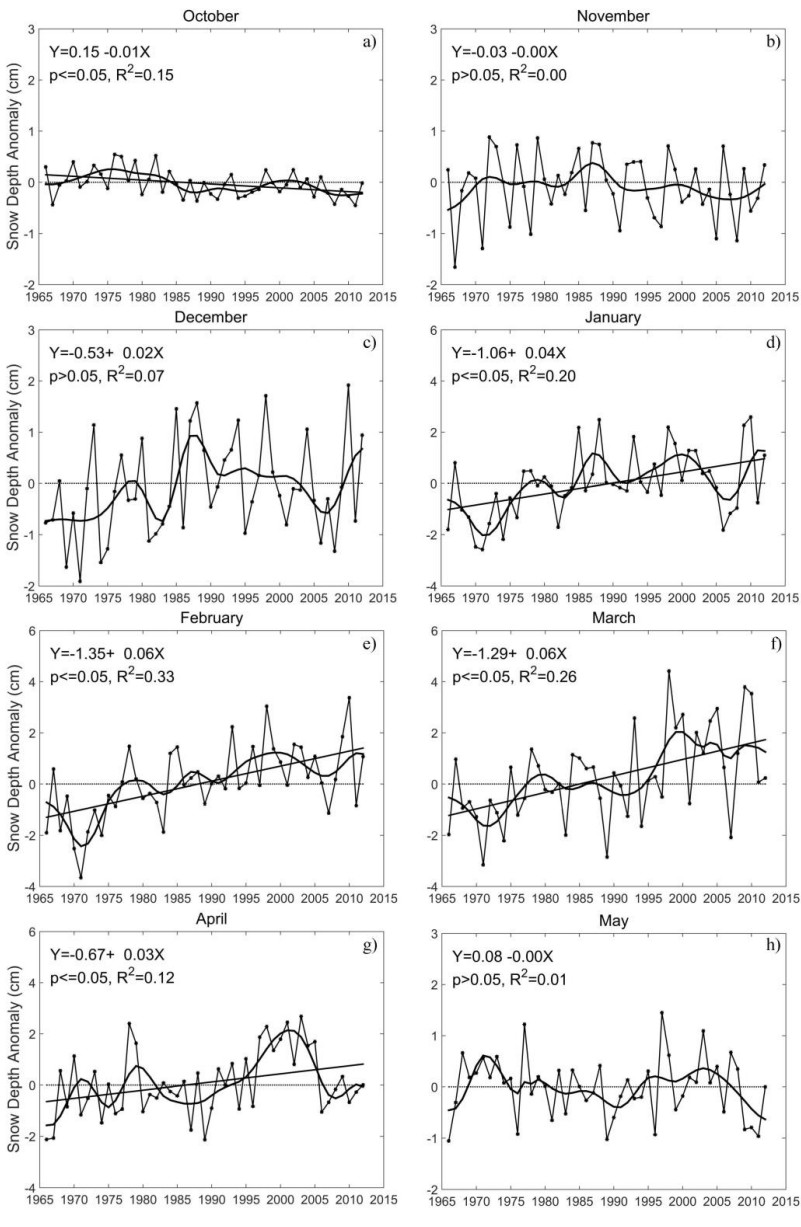

**Figure 5.** Composites of inter-annual variation of monthly mean snow depth (from October to May) from 1966
through 2012 with respect to the 1971-2000 mean across the Eurasian continent. (a) October, (b) November, (c)
December, (d) January, (e) February, (f) March, (g) April, (h) May. The line with dots is the anomaly of snow
depth; the thick curve represents the smoothed curve using wavelet analysis; the thick line presents a linear
regression trend.





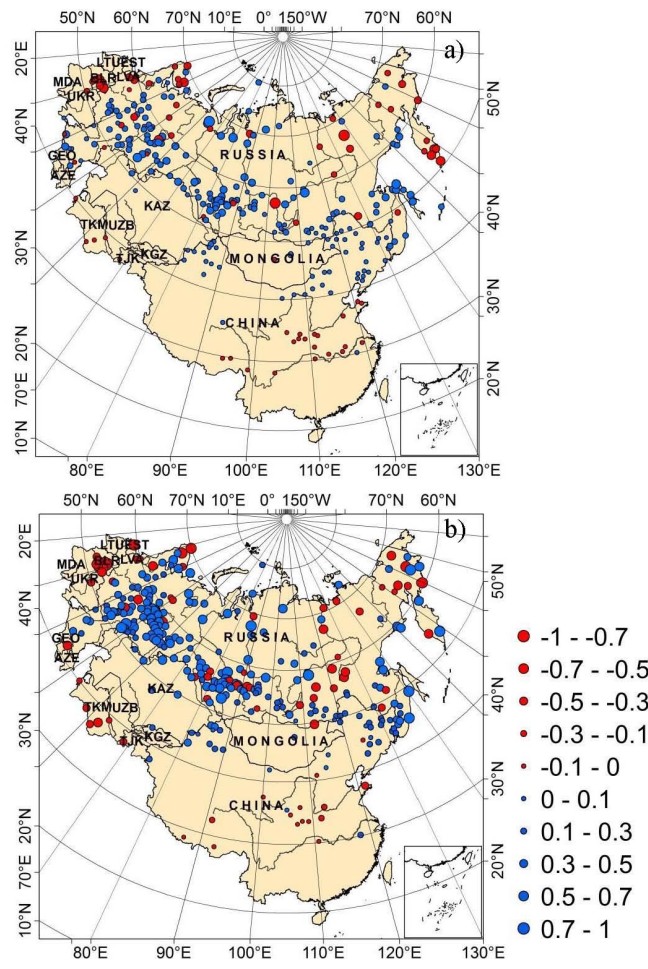

**Figure 6.** Spatial distribution of linear trend coefficients (cm yr$^{-1}$) of annual mean snow depth (a) and maximum snow depth (b) for each station in 1966-2012. The rate of change was at the 95% level. Red circles represent a decreasing trend, and blue circles represent an increasing trend.









**Figure 7.** Spatial distributions of linear trend coefficients (cm yr$^{-1}$) of monthly mean snow depth (from October to May) during 1966 to 2012. (a)October, (b) November, (c) December, (d) January, (e) February, (f) March, (g) April, (h) May. The rate of change was at the 95% level. Red circles represent a decreasing trend, and blue circles represent an increasing trend.

**Figure 8.** Annual mean snow depth changes with latitude (a) and elevation (b) for all stations across the Eurasian continent during 1966-2012. Asterisks show the mean snow depth of each station; the thick line is a linear regression trend; the different colors represent snow depth (cm) of each station (c).





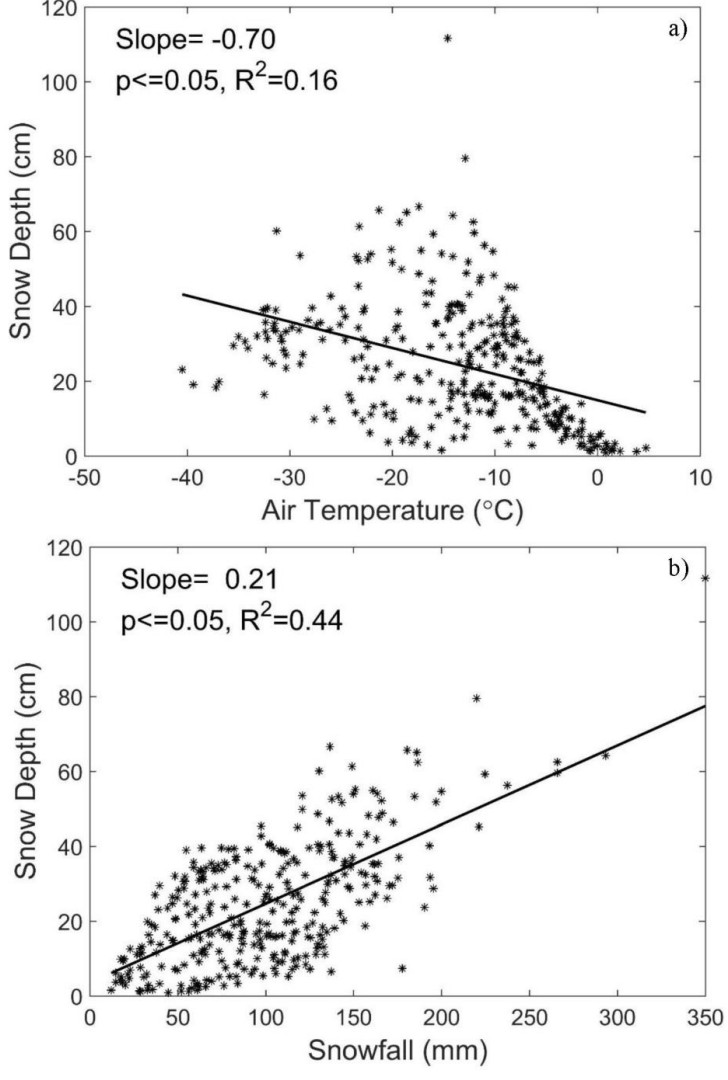

**Figure 9.** The relationships among annual mean snow depth, air temperature and snowfall for 386 stations from

November through March during 1966-2009 over the USSR. The thick line is a linear regression trend.





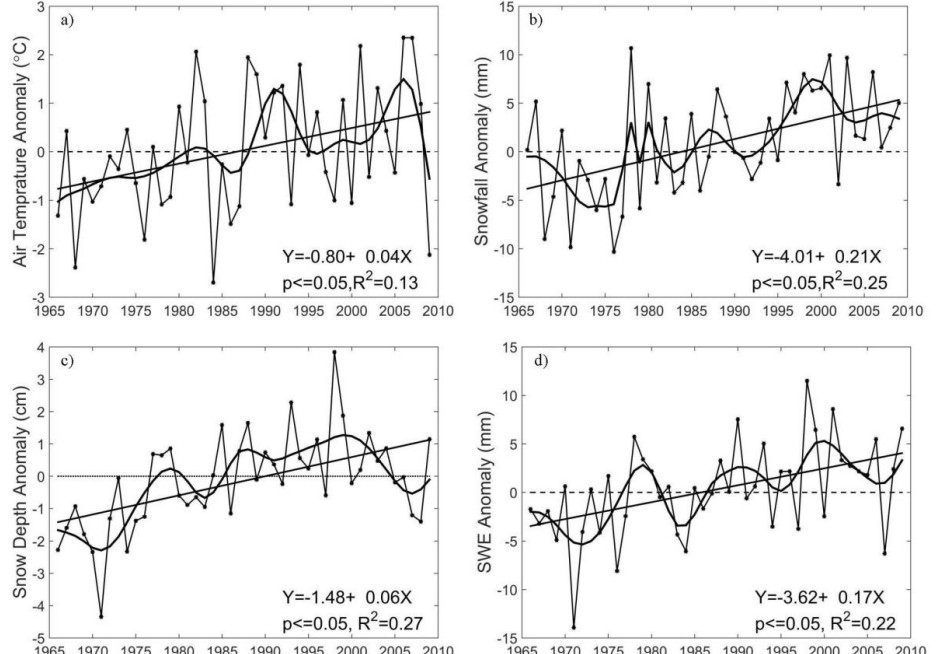

**Figure 10.** Composite of inter-annual variation of annual mean air temperature (a), annual snowfall (b), annual
snow depth (c) and snow water equivalent (d) from November through March during 1966-2009 with respect to
the 1971-2000 mean across the former USSR. The line with dots is the composite of the annual means; the thick
curve represents the smoothed curve using wavelet analysis; the thick line presents a linear regression trend.

