# Peer review of "Spatiotemporal Variability of Snow Depth across the Eurasian Continent from 1966 to 2012"

_The Cryosphere, 2016_

## Referee Comment (RC1) · Anonymous Referee #1 · 11 Oct 2016

Summary: In this paper, the authors develop a snow depth climatology across the Eurasian continent using ground-based observations over 1966-2012. A total of 1814 stations from 17 countries spanning Eurasia with snow data are used to assess mean annual and maximum snow depth and their trends for each site. The northern reaches of Eurasia typically have the greatest mean annual snow depth, revealing a latitudinal dependence on the results. Trends assessed from linear regressions show significant increases in snow depth poleward of 50°N. These trends are associated more so with increased snowfall rather than rising air temperatures.

This paper provides a comprehensive climatology and trends of snow depth across Eurasia over an extended period of time. However, it is unclear how novel the results

are compared to previous, similar studies (as cited in the paper) apart perhaps from the geographical extent of the study. My report provides guidance the authors may consider in revising their manuscript.

General Comments:

1) This study examines the characteristics and trends across the Eurasian continent from 1966 to 2012. To do so, the authors assemble snow depth data from 1103 stations across the study area. How representative are the station (point) snow depth data of the overall regional landscapes of interest? For instance, are snow depth data in forested areas collected at airports or other open areas, that may not represent the regional snow characteristics?

2) Further to this, snow course data from the former USSR are also employed in establishing the snow depth climatology (see Section 2). Is it therefore a fair comparison to present the station (point) data with those from local (spatially averaged) data?

3) The Introduction section is quite lengthy and could be abbreviated by focusing on past studies that report climatologies and trends in snow depth across Eurasia only and the gap being filled by the present study. Further to this, the Introduction should emphasize the novelty of this research compared to previous studies cited in the text.

4) The authors should consider the Mann-Kendall test to assess linear trends or other non-parametric trend analysis rather than linear regressions.

5) Do the linear trends reported in Section 3.2 exceed the variability in the snow depth data? In other words, are there "detectable" trends in snow depth, i.e. with the signal greater than the noise in the system?

6) All figures are rather small and difficult to interpret when printed on paper.

Specific Comments:

1) P. 1, line 21: Insert "a" before "snow depth". Then insert "its" before "spatiotemporal".

[Figure]

2) P. 1, line 27: Consider a word other than "dramatically" here. Are these statistically-significant trends?

3) P. 3, lines 10-20: Note that the tense for verbs changes throughout this paragraph.

4) P. 3, lines 22-24: Are the soil thermal conditions reported here for winter only?

5) P. 4, line 8: Delete "the" before "ecological".

6) P. 5, line 8: Delete "In order" and begin the sentence with "To obtain. . ."

7) P. 6, lines 4-8: Delete "Using data from ground-based measurements" as this repeats text from the previous sentence. Also, please rephrase the statement "a detailed description of snow depth", as this suggests the paper goes at length in describing how snow depth is defined, which is not the case. This entire sentence is awkward and quite long, so should be rephrased and perhaps divided into two sentences.

8) P. 6, line 14: Snow depth data from 17 countries are apparently used in the present study; yet Table 1 lists only three countries (former USSR, Mongolia and China) as sources for the snow depth data.

9) P. 6, line 15: Insert "a" before "daily".

10) P. 6, line 18: Replace "5" with "five".

11) P. 6, line 22: "SWE" has not yet been defined.

12) P. 8, line 2: Delete "In order" and start the sentence with "To reflect. . ."

13) P. 9, line 8: What is a "scale gram"?

14) P. 9, line 15: Delete extra spaces before "from".

15) P. 10, line 11: "TP" is not defined.

16) P. 11, line 4: Delete extra space before "northern".

17) P. 12, line 2: Insert "it" before "fluctuated".

18) P. 12, line 7: Change to "decreasing trend".

19) P. 12, line 25: Rephrase "fluctuant trend".

20) P. 13, line 7: Delete "variability" before "trends".

21) P. 13, line 25: Delete the space in "95%".

22) P. 14, line 23: Delete the extra space before "snow".

23) P. 15, line 2: Variations in hydrometeorological quantities such as snow depth are due to climate variability, not climate change.

24) P. 15, line 7: Here reports of significant declines in snow depth are provided, while the abstract (line 27) suggests the opposite pattern is being observed – which is correct?

25) P. 15, line 18: Change to "increased".

26) P. 15, line 27: Insert "is" before "not".

27) P. 16, lines 7-8: "differences" is used twice in succession.

28) P. 17, line 5: Delete the extra space before "is the".

29) P. 22, line 20: This should read "Liston".

30) P. 26, Figure 1: The colors highlighting three regions (Sakhalin, Kamchatka Peninsula, and northern Xinjiang Autonomous region) are nearly indistinguishable. Please consider using colors of greater contrast. Why are these regions highlighted in the first place? A number of abbreviations are used on the map that are not defined in the figure caption (this is an issue in other figures as well).

31) P. 27, Figure 2: Given the high number of sites with high average snow depth values in the northern reaches of the Eurasian continent, would the results be better depicted using contour lines instead? Consider adding the latitudinal averages of the snow data as secondary diagrams to these figures.

32) P. 30, Figure 4: Does the number of stations used in the composite snow depth anomalies vary over time? The statistical significance of the trends should use the symbol "$\leq$" rather than "<=". Why does the last sentence in the figure caption mention "simulation" of snow depth?

33) P. 31, Figure 5: See comments for Figure 4.

34) P. 34, Figure 8: The statistical significance of the linear regressions should use the symbol "$\leq$" rather than "<=". Are any of the stations below sea level? If not, panels (b) and (c) should have their x-axes begin at 0 m in elevation. The caption should also state that these are relationships between snow depth and latitude and elevation, not changes.

35) P. 35, Figure 9: See comments for Figure 8.

36) P. 36, Figure 10: See comments for Figure 8.

---

## Referee Comment (RC2) · Anonymous Referee #2 · 18 Oct 2016

Review of Spatiotemporal 1 Variability of Snow Depth across the Eurasian Continent from 1966 to 2012, by Zhong et al.

The paper by Zhong et al (2016) investigates the spatio-temporal variability of snow depth (and snow water equivalent or SWE to a lesser extent) over the Eurasian continent over the 1966-2012 period. For this they assembled a considerable dataset of historical snow data measured at meteorological stations and snow courses during the period. The authors are to be commended for assembling this dataset and performing the climatological analysis. While the breath of the interpretations and conclusions remains somewhat limited and could be improved, this paper could serve as a valuable reference for future snow studies on the Eurasian continent. My main reserve concern

the trend detection method, and the analysis of the physiographic control of snow depth spatial variability. Suggestions to improve the analyses are given below.

Main comments

1) methods ïĆğ Methods to measure snow depth. The authors describe the snow course data, How is snow depth/snowfall measured at other stations? Is the method similar in the different countries?

ïĆğ Choice of time window for analysis. The snow year was defined from July 1st to June 30th. Why this choice of period? Figure 3 shows that snow seems to begin accumulating in September but snow remains in June in northern Russia and the Tibetan Plateau. So the chosen analysis window may not be optimal and should probably begin, and end, later so as to capture the complete seasonal cycle over the studied area.

ïĆğ Trend detection. Trend detection in environmental time-series is a delicate topic and this is a big concern for this study, as the 'significant' trends reported could be cited in future works. The authors seem to have used ordinary linear regression (OLS) with classical hypothesis tests (Fisher or 'F-test' on the variance explained, and/or Student T test on regression coefficients). These parametric tests make the assumption that the data is normally and independently distributed. The authors have not reported on checking these assumptions, and I doubt that the time series presented in the figures are free of autocorrelation. As a result I question the validity of much of the 'significant' linear trends reported in this study and suggest the author should apply a statistical test which account for the serial correlation of time-series. A suggestion is given below.

If the data are normally distributed, OLS can be used but the degree of freedom for the significance test must be adjusted for the reduction in the degree of freedom caused by the auto- ('serial) correlation. If the data is not normally distributed, transformation or a non-parametric test is necessary. The Mann-Kendall trend test is commonly used on non-normal data. Here again serial correlation must accounted for. The authors
[Figure]

could quickly apply a normality test and the Durbin-Watson statistic to the residuals of their regression to diagnose these problems. One possible approach to take the autocorrelation into account using OLS is outlined in Weatherhead et al. (1998):

Weatherhead, E. C., et al. (1998), Factors affecting the detection of trends: Statistical considerations and applications to environmental data, J. Geophys. Res., 103(D14), 17149–17161, doi:10.1029/98JD00995.

Another possibility is to apply pre-whitening to the time series. A pertinent paper is:

Yue, S., Pilon, P., Phinney, B. and Cavadias, G. (2002), The influence of autocorrelation on the ability to detect trend in hydrological series. Hydrol. Process., 16: 1807–1829. doi:10.1002/hyp.1095 ïĆğ Wavelet analysis (P8, L26-30 and P9, L1-20) The description of the wavelet analysis is very confusing and seems unnecessarily complicated, due to their limited role in the paper. It does not allow the reader to understand what was done to the data and to replicate the analysis. This section really cuts the flow of reading and should be reworked altogether in order to bring out the essential, with proper supporting references. Which wavelet transform was used in the end, a continuous or discrete? Which wavelet family/filter? From my understanding of this paragraph you applied a discrete wavelet transform, excluded the high-frequency components and then used the inverse transform to reconstruct the lower frequency signal. Or is it that you applied an averaging filter on the wavelet coefficient before reconstructing the signal with the inverse wavelet transform?

2) Results and discussion

Physiographic and climatic control on spatial variability of snow depths The analysis of the factors controlling the spatial variability of snow depths is somewhat limited in breadth. The authors show a general increase in snow depth with latitude and a general decrease in snow depth with increasing altitude. Large-scale control on snow depth will be mainly dependent on the interplay between latitude and altitude but also distance to moisture sources (continentality) and position relative to orographic barriers. Together

these will determine the snowfall rate. For such a large and topographically contrasted region as the Eurasian continent, ignoring these last two effects does not allow a clear understanding of large-scale snow depth spatial variability in the region. The negative (but poor) relationship between snow depth and altitude shown by the authors is largely explained by the continentality and rain shadow effect of the high Tibetan Plateau, while at higher latitude snow depth does seem to increase with altitude in response to orographic enhancement of snowfall. The authors should try to incorporate quantitatively the effects of continentality and barrier effect into their analysis, or at least provide a more in depth discussion of their results in the light of known large-scale physiographic control on the snow cover, with proper supporting references.

The authors further investigate the spatial relationships between mean air temperature, mean snowfall and snow depth. They find that spatial variations in snowfall are positively correlated with snow depth while temperature is negatively, but poorly, correlated with snow depth. While generally interesting, these findings are somewhat expected and do not bring new insights on how the snow cover responds to climate across Eurasia.

Relationships between mean (Eurasian) climate and snow depth over time

The authors revealed interesting increases in snow depth, SWE, temperature and snowfall rates over the study period. However the analysis and interpretation of these tendencies remains somewhat superficial. This section could be enhanced by quantitative analysis, i.e. by performing correlation analysis, and/or multiple correlation/regression analysis to highlight the respective influence of temperature and snowfall changes on mean Eurasian snow depth and SWE. Even more interesting would be to see this analysis done spatially, perhaps in a future study. This would probably highlight the effect of continentality and position relative to orographic barriers on the response of the snow cover to climate.

Specific comments and editorial changes

P3, L13. Although snow cover extent reduced with climate warming, snow depth still increased in northern Eurasia (Kitaev et al., 2005; Bulygina, 2011). Over which period?

P3 L23, 'a' thin snow cover results...

P3 L24. Frauenfeld et al. (2004) indicated that in permafrost areas the maximum snow depth by the end of winter has a significant influence on the active layer depth during the following summer.

P3, L28. The numerical modeling results showed that the rate of mean annual ground surface temperature increase with the increasing maximum snow depth was about 0.1 °C cm-1 for the maximum snow depth at 15 cm. This sentence is convoluted and hard to understand, please rephrase more clearly. Maybe: The numerical modeling results showed that the mean annual ground surface temperature increases with increasing snow depth at a rate of 0.1 °C cm-1 until up to a snow depth of 15 cm...?

P4, L2, ...also increased with snow depth.

P4, L5: Furthermore, snow accumulation an important freshwater resources and has direct impacts on the hydrological cycle.

P4, L11. Adam et al. (2009) suggested that the variations in snow depth will significantly affect the hydrological regime of the Arctic in the future.

P4, L14-26: here you describe trends in snow cover and other variables. Please mention the period over which these changes were observed for respective studies

P4, L28. Snow depth is also closely related to other large-scale atmospheric circulation indices, such as the North Atlantic Oscillation /Arctic Oscillation (NAO/AO) indices. For example, Beniston (1997) found that the NAO....

P5, L8. In order to obtain a wider range of snow depth...Wider range is imprecise. In order to increase the spatial coverage? and/or spatial resolution?

P5, L18. Ground-based snow measurement remains the basis for verification of remote

sensing and instrumental data...

P6, L1. 'TP' = Tibetan Plateau I presume, but I don't think it was defined before.

P6, L15. on a daily basis.

P6, L15. Suggested change: Historical snow course data over the former USSR from 1966 to 2011 were also used in this study

P6, L17 snow surveys performed throughout the accumulation season

P6, L19. Snow surveys were conducted over 1–2 km-long transects.

P6, L20. Snow depth was measured every 10 m in the forest, and every 20 m in open terrain.

P6, L22. SWE: define once

P6. L25 ... over the former USSR. Why only over USSR? Maybe complete sentence with ... 'where SWE data are available...'

P6 L25. SWE was measured every 100 m along the 0.5-1.0 km courses and every 200 m along the 2 km course.

P6, L27. Precipitation data were divided proportionally into daily solid and liquid data, and the solid-to-liquid fraction was determined according to daily mean temperature (Brown,2000). I suggest replacing by: Daily precipitation was partitioned into a solid and liquid fraction, based on daily mean temperature (Brown,2000). You then describe the partitioning equation in the following sentence. $S_r at : what does r at stands for..?$

P7, L9. Quality control steps. (1) daily snow depth observations (equal to or greater than 0 cm, not including missing data) for <15 days in one month were omitted; This is confusing. Do you mean that months having less than 15 days with snow depth data were omitted from the analyse? If that so rephrase in that sense. (2) snow data from stations with <20 years of measurements during 1971-2000 were excluded; I suggest

replacing by: Stations with less than 20 years of data during the 1971-2000 period were excluded from the analysis. 3) data exceeding two standard deviations compared with the annual average value during 1966-2012 were omitted. Add: 'At each station,' before the sentence.

P7, L16. We defined a snow year as the period from July 1st of a current year to June 30th of the following year. Why this choice of period..? maybe add short complement to the sentence: '... so as to insure that the complete seasonal snow cycle is captured across the study region...' Also, I note in Figure 3 that snow remains in June in some areas, and seems to begin accumulating in September. So the chosen analysis window may not be optimal and should probably begin and end later.

P7, L17. Because the procedures for taking snow observations have changed over the course of the studied period, there were some inhomogeneities in the data.

P7, L25....World Meteorological Organization (WMO) climatological products. A reference would be needed here.

P7, L25. A threshold of 15 days was selected because the snow cover duration in some areas of China was less than one month, and the data for 15 days' snow depth in a month were relatively stable. Do you refer to the previously defined quality control step 1? If this is the case this sentence should go in the quality control paragraph. You can here recall it in short sentence.

P8. L2. In order to capture the primary...

P8, (4) Linear trend coefficient of snow depth: the linear trend coefficient of snow depth for each station was obtained by linear regression analysis with respect to time, and thus represents the rate of change in snow depth for a period of time. Replace 'for a period of time' by 'over time'? Or by: 'for a >20 year time period'. Statistical test on linear trend: see main comments...

P8. L23. ...each station and averaged the anomalies for all stations to the anomalies

for the whole Eurasian continent. : 'averaged the anomalies for all stations to obtain mean anomalies for the whole Eurasian continent'.

P8, L26 -. Description of wavelet analysis. See main comment on this. You need to include at least once key reference.

P9, L12. We used an averaging filter for wavelets analysis. Using this method, values that are too small or too large may be excluded; This description is really unclear. Please simplify and add proper references so that the interested reader can fin further explanations on the technique used, if wanted.

P9, L15. obtained from filtering. Remove extra space.

P9, L26. increased with the latitude... A maximum annual mean snow depth... in the west of the Yenisey River

P9, L28. ...were observed in some areas of China. 'some areas' is rather vague, can you be more descriptive? ....due to wind speed, topography, underlying ground surface, and climatic conditions (refs). This is a rather very general statement which does not bring any insights. Of course snow depth will vary everywhere due to these factors... if you do analyse in a later section how these factord affect the spatial variability, mention it. 'The relation between these factors and spatial snow depth variability is further investigated in section xxx'...

P10, L5. The regions with the smallest annual mean snow depth (<5 cm) were located in most areas of the Caucasus Mountains. This is a bit surprising given the high elevations. Is there an elevation bias here? (snow stations are at low elevations?)

P10, L13. varied with the latitude

P10, L15. . ( 201.8 cm) :here as elsewhere in the text you can should round the snow depth to the nearest centimeter, as this is the probably accuracy of the measurements.

P11, L4.... in northern Siberia. Remove extra space between in and northern

P11, L21. ...the increasing rate of snow depth. increasing rates of snow depth

P11. L19-22: linear trends and results plotted on Figure 4: were trends computed on annual anomalies or on the wavelet filtered series? You must provide a trend test that accounts for the autocorrelation of time-series (see main comment).

P12, L1. There was a sharp increase of 3.5 cm in the maximum snow depth during the 1970s, then fluctuated from the late 1970s to the early 1990s. : then fluctuations from... Perhaps be more precise: what type of fluctuation?

P12, L20. the rate of increase being about 0.6 cm decade

P14, L2. in monthly mean snow depth decreased,

P14, L3. Changes in monthly mean snow depth were consistent with the trends in winter over the former USSR but more stations with the decreasing trends in the southern Siberia. Do you mean: 'but more stations with decreasing trends were found in southern Siberia'?

P14, L5. There were few stations with statistically significant trends of snow depth across China; for these, monthly snow depths tended to decrease at most stations.

P14, L11. To explore the spatial variability of snow depth,

P14, L15. snow depth to the north of 40°N

P14, L23. because a snow depth. remove extra space between 'a' and 'snow'

P14. This result indicates that elevation is an important factor affecting snow depth in these regions. I find this statement and the preceding analysis a bit over simplistic. At large scales the snow cover can be tought to depend on latitude, altitude and distance to moisture source (continentality). I feel you are missing the third factor in our analysis. The poor, and generally negative relationship between elevation and snow depth is interesting because it is contrary to what would be expected from orographic effects on precipitation amounts and phase. What you show is that the high elevation of the TP

does not cause larger snow depth compared to surrounding lower lands. Continentality seems to be the main driving factor here: the TP is in the rain shadow of the Himalaya and as such is moisture-deprived. This should be better discussed, and analysed in the paper. This effect could be investigated, perhaps using a simple continentality index (e.g. http://glossary.ametsoc.org/wiki/Continentality). These indices rely on temperature annual ranges. You could use the closest distance to coast as another simple index.

P15: section 3.4.

You begin the section by stating that ' Variations in snow depth are closely related to climate change'. But what is investigated is the influence of spatially variable climate factors (mean temperature and mean snowfall) on snow depth, and NOT the effect of time-varying climate on snow depth. To do so you would have to test the influence of changing temperatures and snowfall/precipitation on snow depth over time. Rephrase the introduction of the section to clearly explain that you investigate spatial relationships between mean temperature and snowfall on mean snow depths.

The spatial relationship between air temperature and snow depth will be undoubtedly complex when considered an area as big and topographically diverse as the Eurasian continent. Your analysis is till interesting as it shows that snowfall is the main factor driving spatial variability in snow depth, at this spatial scale. However snowfall rates and air temperature must also be somewhat correlated, as snowfall depends on precipitation and temperature (precipitation phase). I suggest that you also calculate and report the partial correlation coefficients, i.e. to show the influence one variable while removing effect of the other, on snow depth.

You do examine the effect of changing climate on snow depth and mass in Figure 10 for the composite eurasian and russian records. This analysis is qualitative and while interesting and valuable, it could be enriched by calculating and presenting the correlation coefficients between series. Especially for the SWE series, how much of the

variance can be respectively explained by air temperature and snowfall? Even more instructive would be to perform this analysis on a station basis and map the results. We would then learn about the spatially variable climate control on snow.

P15, L7. Snow depth significantly decreases with increasing air temperature (P$\leq$0.05),

P15, L17. increases

P15, L21. The significant increasing snowfall can explain the sudden drop in snow density observed from the mid-1990s through the early 2000s (Zhong et al., 2014): fresh snow with low snow density. Explain the last statement better in a separate statement. Why does increasing snowfall decreases snow density? Is it the mean density of the snowpack? Increasing snowfall in response to warmer temperature should increase the density, both of fresh snow, and perhaps of the whole snowpack due to faster metamorphism and increased compaction...

P15, L26. increasing trend of changes in snow depth. trend in snow depth? or trend in the rate of change?

P15, L27. In fact, the climatology of snow depth not only influenced by air temperature and precipitation, but also with other climatic factors and atmospheric circulation. Poor formulation, rephrase.

P16. L7. These discrepancies may result from differences in the time frame

P16. L26. during the different study periods.

P16. L.26-28. The sensitivity of snow cover to air temperature and precipitation for each station showed regional differences (Fallot et al., 1997; Park et al., 2013). The amount of snowfall can be affected by climate change, and leading to differences in snow depth at different times (Ye et al., 1998; Kitaev et al., 2005). This is why simple spatial relationship between air temperature and snow depth do not exist...

P17, L5. and is the main reason. Extra space between 'and' and 'is'

P17, L10. Therefore, we will select a typical climate zone to research the climatology and variations of snow cover. This rather vague... your study looks at large scale control on snow cover and this is what this dataset allows. Studying small scale (topography, vegetation) effects on the snow cover requires other kind of data, sampled at a higher spatial resolution. I would remove this sentence.

You should better discuss your results in the light of what is known about large-scale control on snow cover: latitude, altitude and continentality are the main geographical factor which drive snowfall rates and hence snow depths. I find your analysis and the discussion on page 17 somewhat incomplete in this respect.

---

## Author Comment (AC1) · 6 Jan 2017

**Response to Referee #1**

**General Comments:**

1) This study examines the characteristics and trends across the Eurasian continent from 1966 to 2012. To do so, the authors assemble snow depth data from 1103 stations across the study area. How representative are the station (point) snow depth data of the overall regional landscapes of interest? For instance, are snow depth data in forested areas collected at airports or other open areas, that may not represent the regional snow characteristics?

Reply:Thank you for your comments and concerns. The spatial representativeness of stations is always a key and difficult problem in snow depth research or any ground-based studies at various regional scales. In fact, we did not do spatial interpolation of snow depth using these in-situ data across the study area just because the uneven distribution of stations spatially and among different landscapes. The passive microwave remote sensing snow depth products may mitigate the regional coverage problem, their low spatial resolution ($25 \times 25$ km) and high uncertainties (up to 200%) provide no better help to the issue. The combination of in-situ snow depth data with satellite remote sensing snow depth will be a better approach but it is out of the scope of this study. Here for the first time, we present all data we can possibly collect from various countries over the continent and show snow depth spatial variations and temporal changes. We are fully aware the shortcomings of station distribution but this in-situ dataset and its coverage is unprecedented. We may read a lot of published literatures regarding snow cover extent in regional or hemispheric scales, but not snow depth. In this study, we present spatial and temporal changes in snow depth using available in-situ data.

2) Further to this, snow course data from the former USSR are also employed in establishing the snow depth climatology (see Section 2). Is it therefore a fair comparison to present the station (point) data with those from local (spatially averaged) data?

Reply: In our study, 440 stations have both snow course and station data. We compared the snow course averaged data and the station point data and found that they were statistically significantly correlated, and the goodness of fit reached to 94% (Fig. 1). Therefore, we are confident that it was a fair practice to combine the snow course average data and the station point data together in this study.

[Figure]

Figure1. The relationship between snow depth of meteorological station and snow course at 440 stations.

3) The Introduction section is quite lengthy and could be abbreviated by focusing on past studies that report climatologies and trends in snow depth across Eurasia only and the gap being filled by the present study. Further to this, the Introduction should emphasize the novelty of this research compared to previous studies cited in the text.

Reply: We have abbreviated the introduction, and focused on the report of climatologies and trends in snow depth, the existing problems of the previous studies, as well as the characteristics of our study.

4) The authors should consider the Mann-Kendall test to assess linear trends or other non-parametric trend analysis rather than linear regressions.

Reply: Any trend analysis is an approximate and simple approach to obtain what has happened on average during the study period. Linear trend analysis provides an average rate of this change. Despite there is a nonlinearity, the linear trend analysis is also a useful approximation when a systematic low-frequency variations emerged. Meanwhile, to overcome the strong assumption in ordinary least squares (independent and normal distribution), we added a Mann-Kendall (MK) test to identify the monotonic trend in snow depth. These two test methods could provide more robust and comprehensive information of the trend analysis. We have added the method introduction in the "data and methodology" section and discussed the similarities and differences of the two kind of trend analysis results in the "results" section.

"Any trend analysis is an approximate and simple approach to get what has happened on average during the study period. Linear trend analysis provides an average rate of this change. Despite there is a nonlinearity, the linear trend analysis is also an useful approximation when a systematic low-frequency variations emerged. (Folland and Karl, 2001; Groisman et al., 2006). The linear trend coefficient of snow depth was calculated to represent the rate of change at each station. The Student T test was used to assess the statistical significant of the slope in the linear regression analysis and the partial correlation coefficients, and the confidence level above 95% was considered in our study. Meanwhile, to overcome the strong assumption in ordinary least squares (independent and normal distribution), we applied a Mann-Kendall (MK) test to identify the monotonic trend in snow depth. Confidence level above 95% was used to determine the statistically significant increase or decrease in snow depth. These two test methods could provide more robust and comprehensive information of the trend analysis."

"The Mann-Kendall statistical curves of annual and maximum snow depth were consistent with the linear trend analysis (Fig. 5). The increasing trend of annual snow depth reached to the 0.05 confident level in the late 1980s and from the early 1990s to the mid-1990s; it reached to the 0.01 confident level in the late 1990s. The decreasing trend reached to the 0.05 confident level from the early 2000s through the mid-2000s. The intersection of the UF curve and UB curve appeared in the mid-1970s, it indicated that the rising trend was an abrupt change during this period. The abrupt change point of the maximum snow depth was in the mid-1980s, then it increased significantly ($p \leqslant 0.05$) from the early 1990s through the mid-1990s, and it reached to the 0.01 confident level from the late 1990s to the early 2010s."

[Figure]

Figure 5. Mann-Kendall statistical curve of annual mean snow depth (a) and maximum snow depth (b) from 1966 through 2012 across the Eurasian continent. Straight line presents significance level at 0.05.

"In order to identify the monotonic trend in monthly snow depth, we conducted the MK test (Fig. 7). In October, snow depth represented a decreasing trend and it reached to the 0.05 confident level only after 2010. The statistically significant changes of monthly snow depth in November during the period of the late 1980s through the early 2000s, though it was not statistically significant with the linear regression. From December through March, there were increasing trends in monthly snow depth and the abrupt change point appeared in the mid-1970s. In the linear regression analysis, the variation of snow depth was not significant in December. However, the results of M-K test showed that the increasing trend of monthly snow depth reached to the 0.01 confident level during the mid-1980s through the late 1990s, and then it decreased during the 2000s. From January to March, monthly snow depth increased significantly ($p \leqslant 0.01$) from the mid-1980s to the early 2010s. In April, the statistically significant increase was found from the late 1990s to the late 2000s, and it

reached to the 0.01 confident level after 2000. Consistent with the linear regression, the trend in monthly snow depth was not significant in May."

[Figure]

Figure 7. Mann-Kendall statistical curve of monthly mean snow depth (from October to May) from 1966 through 2012 across the Eurasian continent. (a) October, (b) November, (c) December, (d) January, (e) February, (f) March,

(g) April, (h) May. Straight line presents significance level at 0.05.

5) Do the linear trends reported in Section 3.2 exceed the variability in the snow depth data? In other words, are there "detectable" trends in snow depth, i.e. with the signal greater than the noise in the system?

Reply: The Student T test was used to assess the statistical significant of the slope in the linear regression analysis, and the confidence level above 95% was considered in our study. We have tested the results of the linear trends in Section 3.2, and the results show that all of the "detectable" trends in snow depth were greater than the noise in the system.

6) All figures are rather small and difficult to interpret when printed on paper.

Reply: Thank you for pointing this out. We have expanded all figures.

**Specific Comments:**
1) P. 1, line 21: Insert "a" before "snow depth". Then insert "its" before "spatiotemporal".

Reply: Has been done.

2) P. 1, line 27: Consider a word other than "dramatically" here. Are these statistically significant trends?

Reply: We replaced it with "significantly". In our study, the trends with the confidence level above 95% were only considered.

3) P. 3, lines 10-20: Note that the tense for verbs changes throughout this paragraph.

Reply: We replaced "is" with "was" in line 15, and replaced "promotes" with "prompted" in line 19.

4) P. 3, lines 22-24: Are the soil thermal conditions reported here for winter only?

Reply: Yes, the soil thermal conditions are in winter.

5) P. 4, line 8: Delete "the" before "ecological".

Reply: Has been deleted.

6) P. 5, line 8: Delete "In order" and begin the sentence with "To obtain…"

Reply: Thank you for your suggestions. We revised it.

7) P. 6, lines 4-8: Delete "Using data from ground-based measurements" as this repeats text from the previous sentence. Also, please rephrase the statement "a detailed description of snow depth", as this suggests the paper goes at length in describing how snow depth is defined, which is not the case. This entire sentence is awkward and quite long, so should be rephrased and perhaps divided into two sentences.

Reply: Thank you for your suggestions. We rephrased the sentence: "The objective of this study is to investigate the climatology and variability of snow depth, and analyze snow depth relationships with the topography and climate factors over the Eurasian continent from 1966 to 2012."

8) P. 6, line 14: Snow depth data from 17 countries are apparently used in the present study; yet Table 1 lists only three countries (former USSR, Mongolia and China) as sources for the snow depth data.

Reply: Seventeen countries includes China, Mongolia and 15 countries previously belonged to the former USSR. In order to avoid the readers' confusion, we deleted "in 17 countries".

9) P. 6, line 15: Insert "a" before "daily".

Reply: We inserted it.

10) P. 6, line 18: Replace "5" with "five".

Reply: We revised it.

11) P. 6, line 22: "SWE" has not yet been defined.

Reply: We have defined SWE at P. 3, line 7.

12) P. 8, line 2: Delete "In order" and start the sentence with "To reflect…"

Reply: We revised it.

13) P. 9, line 8: What is a "scale gram"?

Reply: We deleted "gram".

14) P. 9, line 15: Delete extra spaces before "from".

Reply: We deleted it.

15) P. 10, line 11: "TP" is not defined.

Reply: We have defined TP at P. 5, line 7.

16) P. 11, line 4: Delete extra space before "northern".

Reply: We deleted it.

17) P. 12, line 2: Insert "it" before "fluctuated".

Reply: Thank you very much for your suggestion. We inserted it.

18) P. 12, line 7: Change to "decreasing trend".

Reply: We revised it.

19) P. 12, line 25: Rephrase "fluctuant trend".

Reply: We inserted "increasing" before "trend".

20) P. 13, line 7: Delete "variability" before "trends".

Reply: We deleted it.

21) P. 13, line 25: Delete the space in "95%".

Reply: We deleted it.

22) P. 14, line 23: Delete the extra space before "snow".

Reply: We deleted it.

23) P. 15, line 2: Variations in hydrometeorological quantities such as snow depth are due to climate variability, not climate change.

Reply: Thank you very much for your suggestion. We replaced "climate change" with "climate variability".

24) P. 15, line 7: Here reports of significant declines in snow depth are provided, while the abstract (line 27) suggests the opposite pattern is being observed – which is correct?

Reply: Here the result showed the relationship between snow depth and air temperature. There was a negative correlation between them. Increasing air temperature result in the snow depth decreased. However, in the abstract, the increasing trend represented the interannual variation in snow depth. They are different.

25) P. 15, line 18: Change to "increased".

Reply: We revised it.

26) P. 15, line 27: Insert "is" before "not".

Reply: We inserted it.

27) P. 16, lines 7-8: "differences" is used twice in succession.

Reply: We replaced the first "differences" with "discrepancies"

28) P. 17, line 5: Delete the extra space before "is the".

Reply: We deleted it.

29) P. 22, line 20: This should read "Liston".

Reply: Thank you for your suggestions. We revised it.

30) P. 26, Figure 1: The colors highlighting three regions (Sakhalin, Kamchatka Peninsula, and northern Xinjiang Autonomous region) are nearly indistinguishable. Please consider using colors of greater contrast. Why are these regions highlighted in the first place? A number of abbreviations are used on the map that are not defined in the figure caption (this is an issue in other figures as well).

Reply: We highlighted the three regions due to snow depths were greater in these areas. We wanted to indicate the accurately locations for readers who are not familiar with the geography of Eurasia. We have canceled the highlight because it may cause confusion for the reader. The country abbreviations were used because the space is limited and cannot be spelled out. We have spelled out the names of countries as an annex.

Abbreviation Description

| Country | Abbreviation |
| --- | --- |
| Kazakhstan | KAZ |
| Ukraine | UKR |
| Turkmenistan | TKM |
| Uzbekistan | UZB |
| Tajikistan | TJK |
| Belarus | BLR |
| Estonia | EST |
| Georgia | GEO |
| Latvia | LVA |
| Lithuania | LTU |

| Azerbaijan | AZE |
| Kyrgyzstan | KGZ |
| Moldova | MDA |

31) P. 27, Figure 2: Given the high number of sites with high average snow depth values in the northern reaches of the Eurasian continent, would the results be better depicted using contour lines instead? Consider adding the latitudinal averages of the snow data as secondary diagrams to these figures.

Reply: We tried to use the contour lines instead of the point values. But we found that there was a problem of the accuracy of interpolation with Kriging interpolation in AcrGIS, in which there was snow cover in some no snow areas. Therefore, the results cannot be depicted using contour lines instead. Snow depth distributions are affected by the topographic factors over the Eurasia. Snow depth is also affected by elevation, slope, aspect in the same latitude. The latitudinal average of snow data cannot fully reflect the snow depth distribution.

32) P. 30, Figure 4: Does the number of stations used in the composite snow depth anomalies vary over time? The statistical significance of the trends should use the symbol "$\leqslant$" rather than "<=". Why does the last sentence in the figure caption mention "simulation" of snow depth?

Reply: Thank you for your suggestions. The number of stations used in the composite snow depth anomalies vary over time. First, we calculated the snow depth anomaly at each site, and then took the average of the anomalies as the general anomaly. We replaced "<=" with "$\leqslant$". In the figure, X means the value of the linear regression trend, which is calculated by the snow depth anomaly with linear regression. Therefore, it is a calculation of snow depth.

33) P. 31, Figure 5: See comments for Figure 4.

Reply: We replaced "<=" with "$\leqslant$".

34) P. 34, Figure 8: The statistical significance of the linear regressions should use the symbol "$\leqslant$" rather than "<=". Are any of the stations below sea level? If not, panels (b) and (c) should have their x-axes begin at 0 m in elevation. The caption should also state that these are relationships between snow depth and latitude and elevation, not changes.

Reply: Thank you for your suggestions. We replaced "<=" with "$\leqslant$". There are 9 stations below sea level in the former USSR. We revised the caption: "The relationships among annual mean snow depth, air temperature and snowfall for 386 stations from November through March during 1966-2009 over the USSR. The thick line is a linear regression trend."

35) P. 35, Figure 9: See comments for Figure 8.

Reply: We replaced "<=" with "$\leqslant$".

36) P. 36, Figure 10: See comments for Figure 8.

Reply: We replaced "<=" with "$\leqslant$".

---

## Author Comment (AC2) · 6 Jan 2017

**Response to Referee #2**

**General Comments:**

1) Methods to measure snow depth. The authors describe the snow course data, how is snow depth/snowfall measured at other stations? Is the method similar in the different countries?

Reply: All measurements at the meteorological stations across the Eurasian continent was set at the same standard by the former USSR after the World War II, including China in the 1950s. All snow depth measurements at these meteorological stations were conducted with the same kind of instruments, the same standard, and the same local time. Therefore, snow depth at all meteorological stations are all consistent. Snow courses were established and operated the same way as the snow depth measurements.

Choice of time window for analysis. The snow year was defined from July 1st to June 30th. Why this choice of period? Figure 3 shows that snow seems to begin accumulating in September but snow remains in June in northern Russia and the Tibetan Plateau. So the chosen analysis window may not be optimal and should probably begin, and end, later so as to capture the complete seasonal cycle over the studied area.

Reply: We checked that the snow cover extent is the smallest in July and August. We also conducted analysis about snow depth in July and snow was essentially gone based on the data we have. Snowfall can happen anytime of the year on the Qinghai-Tibetan Plateau but it only lasts from a few hours to a few days during summer months such as June through August. We believe it is safe to say that a snow-year from June 30 through July 1 would capture the entire seasonal snow cycle.

Trend detection. Trend detection in environmental time-series is a delicate topic and this is a big concern for this study, as the 'significant' trends reported could be cited in future works. The authors seem to have used ordinary linear regression (OLS) with classical hypothesis tests (Fisher or 'F-test' on the variance explained, and/or Student T test on regression coefficients). These parametric tests make the assumption that the data is normally and independently distributed. The authors have not reported on checking these assumptions, and I doubt that the time series presented in the figures are free of autocorrelation. As a result I question the validity of much of the 'significant' linear trends reported in this study and suggest the author should apply a statistical test which account for the serial correlation of time-series. A suggestion is given below.

If the data are normally distributed, OLS can be used but the degree of freedom for the significance test must be adjusted for the reduction in the degree of freedom caused by the auto- ('serial) correlation. If the data is not normally distributed, transformation or a non-parametric test is necessary. The Mann-Kendall trend test is commonly used on

non-normal data. Here again serial correlation must accounted for. The authors could quickly apply a normality test and the Durbin-Watson statistic to the residuals of their regression to diagnose these problems. One possible approach to take the autocorrelation into account using OLS is outlined in Weatherhead et al. (1998):

Weatherhead, E. C., et al. (1998), Factors affecting the detection of trends: Statistical considerations and applications to environmental data, J. Geophys. Res., 103(D14), 17149–17161, doi:10.1029/98JD00995.

Another possibility is to apply pre-whitening to the time series. A pertinent paper is:

Yue, S., Pilon, P., Phinney, B. and Cavadias, G. (2002), The influence of autocorrelation on the ability to detect trend in hydrological series. Hydrol. Process., 16: 1807–1829. doi:10.1002/hyp.1095 ï´C ˘g Wavelet analysis (P8, L26-30 and P9, L1-20)

Reply: Thank you very much for your comments and suggestions. Any trend analysis is an approximate and simple approach to obtain what has happened on average during the study period. Linear trend analysis provides an average rate of this change. Despite there is a nonlinearity, the linear trend analysis is also a useful approximation when a systematic low-frequency variations emerged (Folland and Karl, 2001; Groisman et al., 2006). The linear trend coefficient of snow depth was calculated to represent the rate of change at each station. We checked and found that the data were normally distributed. The Student T test was used to assess the statistical significant of the slope in the linear regression analysis and the partial correlation coefficients, and the confidence level above 95% was considered in our study. Meanwhile, to overcome the strong assumption in ordinary least squares (independent and normal distribution), we also added the Mann-Kendall (MK) test to identify the monotonic trend in snow depth. Confidence level above 95% was used to determine the statistically significant increase or decrease in snow depth. These two test methods could provide more robust and comprehensive information of the trend analysis.

Folland, C.K. and Karl, T.R. (2001), Observed climate variability and change. Climate Change 2001: The Scientific Basis, edited by: J. T. Houghton et al., Cambridge University Press, Cambridge, UK, 99–181.

Groisman, P. et al. (2006), State of the Ground: Climatology and Changes during the Past 69 Years over Northern Eurasia for a Rarely Used Measure of Snow Cover and Frozen Land, J. Climate, 19, 4933–4955.

The description of the wavelet analysis is very confusing and seems unnecessarily complicated, due to their limited role in the paper. It does not allow the reader to understand what was done to the data and to replicate the analysis. This section really cuts the flow of reading and should be reworked altogether in order to bring out the essential, with proper supporting references. Which wavelet transform was used in the end, a continuous or discrete? Which wavelet family/filter? From my understanding of this paragraph you applied a discrete wavelet transform, excluded the high-frequency components and then used the inverse transform to reconstruct the lower frequency signal. Or is it that you applied an averaging filter on the wavelet coefficient before reconstructing the signal with the inverse wavelet

transform?

Reply: We have revised the description of the wavelet analysis. As your comments, we applied a discrete wavelet transform, excluded the high-frequency components and then used the inverse transform to reconstruct the lower frequency signal. The new description as following:

"Wavelet analysis was performed to reveal the long-term low-frequency variations of snow depth over the study area as a whole. A wavelet is a wave-like oscillation with an amplitude that begins at 0, increases, and then decreases back to 0 (Graps, 1995). We applied a discrete wavelet transform, excluded the high-frequency components and then used the inverse transform to reconstruct the lower frequency signal."

2) Results and discussion
Physiographic and climatic control on spatial variability of snow depths. The analysis of the factors controlling the spatial variability of snow depths is somewhat limited in breadth. The authors show a general increase in snow depth with latitude and a general decrease in snow depth with increasing altitude. Large-scale control on snow depth will be mainly dependent on the interplay between latitude and altitude but also distance to moisture sources (continentality) and position relative to orographic barriers. Together these will determine the snowfall rate. For such a large and topographically contrasted region as the Eurasian continent, ignoring these last two effects does not allow a clear understanding of large-scale snow depth spatial variability in the region. The negative (but poor) relationship between snow depth and altitude shown by the authors is largely explained by the continentality and rain shadow effect of the high Tibetan Plateau, while at higher latitude snow depth does seem to increase with altitude in response to orographic enhancement of snowfall. The authors should try to incorporate quantitatively the effects of continentality and barrier effect into their analysis, or at least provide a more in depth discussion of their results in the light of known large-scale physiographic control on the snow cover, with proper supporting references.

Reply: Thank you very much for your comments and suggestions. We indeed conducted analysis on impact of continentality on snow depth over the study area. To our surprise, the correlation between the two are not significant. However, we strongly agree with the reviewer that at local and regional (less than continental) scales, topography can play key roles. In Section 4.2, we added more discussions about topographical effects on snow depth, especially on Tibetan Plateau (TP). The Tibetan Plateau's largest snow accumulation occurred in the winter, but the snowfall during winter months is the smallest of the year. This was mainly due to the terrain factors: the water vapor from the east and west was blocked by the Hengduan Mountains and Nyainqentanglha Mountains, respectively, which resulted in less snowfall. Although there was more snowfall in spring, snow cover was not easy to accumulate with higher temperatures. Therefore, snow depth was shallow on TP in general. In addition to topographic factors, spatial distribution of snow depth was also affected by atmospheric

circulation. We will discuss this issue in the future studies.

Relationships between mean (Eurasian) climate and snow depth over time.
The authors revealed interesting increases in snow depth, SWE, temperature and snowfall rates over the study period. However the analysis and interpretation of these tendencies remains somewhat superficial. This section could be enhanced by quantitative analysis, i.e. by performing correlation analysis, and/or multiple correlation/regression analysis to highlight the respective influence of temperature and snowfall changes on mean Eurasian snow depth and SWE. Even more interesting would be to see this analysis done spatially, perhaps in a future study. This would probably highlight the effect of continentality and position relative to orographic barriers on the response of the snow cover to climate.

Reply: We have added the quantitative analysis by the partial correlation analysis between snow depth, SWE, air temperature and snowfall. The results showed that the significant negative correlation ($p \leq 0.05$) between snow depth and air temperature presented in most areas of European Russia and the southern Siberia (Fig 13a). The stations with negative effects of air temperature on SWE were fewer, and there were no statistically significant correlation in the northern Siberia (Fig 13b). It was because the air temperature was below 0℃ in most areas of Siberia during December through March, the increasing temperature did not have an obvious effect on snow depth. Consistent with the interannual variation, changes in snow depth and SWE were more affected by snowfall in most areas across the former USSR from December through March. The greater partial correlation coefficients (>0.6) between snow cover and snowfall appeared in the northern European Russia, the southern Siberia, the northeast and southeast of the Russian Far East. Variations in snow depth and SWE were more sensitive to snowfall and snowfall rate in these areas.

**Specific comments and editorial changes:**
1) P3, L13. Although snow cover extent reduced with climate warming, snow depth still increased in northern Eurasia (Kitaev et al., 2005; Bulygina, 2011). Over which period?

Reply: The sentence is revised as "Although snow cover extent reduced with climate warming, snow depth still increased in the northern Eurasia during 1936 to 1995 (Kitaev et al., 2005) and 1966-2010 (Bulygina et al., 2011)."

2) P3 L23, 'a' thin snow cover results...

Reply: We deleted the third paragraph in P.3 and the second paragraph in P.4 according to the suggestion of referee #1.

3) P3 L24. Frauenfeld et al. (2004) indicated that in permafrost areas the maximum snow depth by the end of winter has a significant influence on the active layer depth during the following summer.

Reply: We deleted the third paragraph in P.3 and the second paragraph in P.4 according to the suggestion of referee #1.

4) P3, L28. The numerical modeling results showed that the rate of mean annual ground surface temperature increase with the increasing maximum snow depth was about 0.1℃ cm-1 for the maximum snow depth at 15 cm. This sentence is convoluted and hard to understand, please rephrase more clearly. Maybe: The numerical modeling results showed that the mean annual ground surface temperature increases with increasing snow depth at a rate of 0.1 _C cm-1 until up to a snow depth of 15 cm...?

Reply: We deleted the third paragraph in P.3 and the second paragraph in P.4 according to the suggestion of referee #1.

5) P4, L2, ...also increased with snow depth.

Reply: We deleted the third paragraph in P.3 and the second paragraph in P.4 according to the suggestion of referee #1.

6) P4, L5: Furthermore, snow accumulation an important freshwater resources and has direct impacts on the hydrological cycle.

Reply: We deleted the third paragraph in P.3 and the second paragraph in P.4 according to the suggestion of referee #1.

7) P4, L11. Adam et al. (2009) suggested that the variations in snow depth will significantly affect the hydrological regime of the Arctic in the future.

Reply: We deleted the third paragraph in P.3 and the second paragraph in P.4 according to the suggestion of referee #1.

8) P4, L14-26: here you describe trends in snow cover and other variables. Please mention the period over which these changes were observed for respective studies

Reply: Thank you very much for your suggestion. We added the period over these changes for respective studies:

"overall, the annual mean snow depth decreased in most areas over North America during 1946 to 2000 (Brown and Braaten, 1998; Dyer and Mote, 2006), and increased in Eurasia and the Arctic during the recent 70 years (Ye et al., 1998; Kitaev et al., 2005; Callaghan et al., 2011a; Liston and Hiemstra, 2011) but there was regional differences (Bulygina et al., 2009, 2011; Ma and Qin, 2012; Stuefer et al., 2013; Terzago et al., 2014). Changes in snow depth were primarily affected by air temperature and precipitation. Ye et al. (1998) and Kitaev et al. (2005) showed that higher air temperatures caused an increase in snowfall in winter from

1936 through 1995, thus greater snow depth was observed in northern Eurasia in response to global warming. Furthermore, snow depth distribution and variation are also controlled by terrain (i.e., elevation, slope, aspect, and roughness) and vegetation (Lehning et al., 2011; Grünewald et al., 2014; Revuelto et al., 2014; Rees et al., 2014; Dickerson-Lange et al., 2015). Snow depth is also closely related to other large-scale atmospheric circulation indices, such as the North Atlantic Oscillation /Arctic Oscillation (NAO/AO) indices. For example, Beniston (1997) found that the NAO played a crucial role in fluctuations in the amount of snowfall and snow depth in the Swiss Alps from 1945 to 1994. Kitaev et al. (2002) reported that the NAO index is positively related to snow depth in the northern part of the East European Plain and over western Siberia during the period 1966-1990;"

9) P4, L28. Snow depth is also closely related to other large-scale atmospheric circulation indices, such as the North Atlantic Oscillation /Arctic Oscillation (NAO/AO) indices. For example, Beniston (1997) found that the NAO....

Reply: We have revised the sentence.

10) P5, L8. In order to obtain a wider range of snow depth...Wider range is imprecise. In order to increase the spatial coverage? and/or spatial resolution?

Reply: We revised it with "increase the spatial coverage"

11) P5, L18. Ground-based snow measurement remains the basis for verification of remote sensing and instrumental data...

Reply: We replace "is" with "remains"

12) P6, L1. 'TP' = Tibetan Plateau I presume, but I don't think it was defined before.

Reply: We have defined it in P.5, line7.

13) P6, L15. on a daily basis.

Reply: We inserted it.

14) P6, L15. Suggested change: Historical snow course data over the former USSR from 1966 to 2011 were also used in this study

Reply: We revised it.

15) P6, L17 snow surveys performed throughout the accumulation season

Reply: We revised it.

16) P6, L19. Snow surveys were conducted over 1–2 km-long transects.

Reply: We revised it.

17) P6, L20. Snow depth was measured every 10 m in the forest, and every 20 m in open terrain.

Reply: We replace "each" with "every"

18) P6, L22. SWE: define once

Reply: We have defined it in P.3, line7.

19) P6. L25 ... over the former USSR. Why only over USSR? Maybe complete sentence with ... 'where SWE data are available...'

Reply: We added it.

20) P6 L25. SWE was measured every 100 m along the 0.5-1.0 km courses and every 200m along the 2 km course.

Reply: We replace "each" with "every"

21) P6, L27. Precipitation data were divided proportionally into daily solid and liquid data, and the solid-to-liquid fraction was determined according to daily mean temperature (Brown,2000). I suggest replacing by: Daily precipitation was partitioned into a solid and liquid fraction, based on daily mean temperature (Brown,2000). You then describe the partitioning equation in the following sentence. $S_r$at : whatdoesratstandsfor…?

Reply: Thank you very much for your suggestion. We replaced this sentence.

22) P7, L9. Quality control steps. (1) daily snow depth observations (equal to or greater than 0 cm, not including missing data) for <15 days in one month were omitted; This is confusing. Do you mean that months having less than 15 days with snow depth data were omitted from the analyse? If that so rephrase in that sense. (2) snow data from stations with <20 years of measurements during 1971-2000 were excluded; I suggest replacing by: Stations with less than 20 years of data during the 1971-2000 period were excluded from the analysis. 3) data exceeding two standard deviations compared with the annual average value during 1966-2012 were omitted. Add: 'At each station,' before the sentence.

Reply: We rephrased this sentence as "Months having less than 15 days with snow depth data were omitted from the analysis"

23) P7, L16. We defined a snow year as the period from July 1st of a current year to June

30th of the following year. Why this choice of period..? maybe add short complement to the sentence: '... so as to insure that the complete seasonal snow cycle is captured across the study region...' Also, I note in Figure 3 that snow remains in June in some areas, and seems to begin accumulating in September. So the chosen analysis window may not be optimal and should probably begin and end later.

Reply: Thank you very much for your suggestion. The snow cover extent is smallest during July to August, in order to capture the beginning of snow cover, we defined a snow year as the period from July 1st of a current year to June 30th of the following year. We have added the explanation of the choice of time window before we defined the snow year.

24) P7, L17. Because the procedures for taking snow observations have changed over the course of the studied period, there were some inhomogeneities in the data.

Reply: We revised it.

25) P7, L25....World Meteorological Organization (WMO) climatological products. A reference would be needed here.

Reply: We added the reference "World Meteorological Organization (WMO) climatological products (Ma and Qin, 2012)"

26) P7, L25. A threshold of 15 days was selected because the snow cover duration in some areas of China was less than one month, and the data for 15 days' snow depth in a month were relatively stable. Do you refer to the previously defined quality control step 1? If this is the case this sentence should go in the quality control paragraph. You can here recall it in short sentence.

Reply: Thank you very much for your suggestion. We moved this sentence to the quality control paragraph.

27) P8. L2. In order to capture the primary...

Reply: We revised it.

28) P8, (4) Linear trend coefficient of snow depth: the linear trend coefficient of snow depth for each station was obtained by linear regression analysis with respect to time, and thus represents the rate of change in snow depth for a period of time. Replace 'for a period of time' by 'over time'? Or by: 'for a >20 year time period'. Statistical test on linear trend: see main comments...

Reply: We replaced it. And the analysis of the statistical test on linear trend is answered in the main comments.

29) P8. L23. ...each station and averaged the anomalies for all stations to the anomalies for the whole Eurasian continent. : 'averaged the anomalies for all stations to obtain mean anomalies for the whole Eurasian continent'.

Reply: We revised it.

30) P8, L26 -. Description of wavelet analysis. See main comment on this. You need to include at least once key reference.

Reply: Thank you very much for your suggestion. We have revised the description of the wavelet analysis. As your understanding, we applied a discrete wavelet transform, excluded the high-frequency components and then used the inverse transform to reconstruct the lower frequency signal. The new description as following:

"Wavelet analysis was performed to reveal the long-term low-frequency variations of snow depth over the study area as a whole. A wavelet is a wave-like oscillation with an amplitude that begins at 0, increases, and then decreases back to 0 (Graps, 1995). We applied a discrete wavelet transform, excluded the high-frequency components and then used the inverse transform to reconstruct the lower frequency signal."

31) P9, L12. We used an averaging filter for wavelets analysis. Using this method, values that are too small or too large may be excluded; This description is really unclear. Please simplify and add proper references so that the interested reader can fin further explanations on the technique used, if wanted.

Reply: We deleted this sentence.

32) P9, L15. obtained from filtering. Remove extra space.

Reply: We deleted the space.

33) P9, L26. increased with the latitude... A maximum annual mean snow depth... in the west of the Yenisey River

Reply: We replace "the" with "A"

34) P9, L28. ...were observed in some areas of China. 'some areas' is rather vague, can you be more descriptive? ....due to wind speed, topography, underlying ground surface, and climatic conditions (refs). This is a rather very general statement which does not bring any insights. Of course snow depth will vary everywhere due to these factors... if you do analyse in a later section how these factord affect the spatial variability, mention it. 'The relation between these factors and spatial snow depth variability is further investigated in section xxx'...

Reply: We added more description L28: "in some areas of the south of Yangtze River in China". We deleted "due to wind speed, topography, underlying ground surface, and climatic conditions (Gray and Male, 1981; Sturm et al., 1995, 2001; Callaghan et al., 2011b)."

35) P10, L5. The regions with the smallest annual mean snow depth (<5 cm) were located in most areas of the Caucasus Mountains. This is a bit surprising given the high elevations. Is there an elevation bias here? (snow stations are at low elevations?)

Reply: The stations with the smallest annual mean snow depth are located in the eastern and western areas of the Caucasus Mountains, close to the coast of the Caspian Sea and the Black Sea, where the elevations are below 2000 m. We have added the specific location in the text: "were located in the eastern and western areas of the Caucasus Mountains"

36) P10, L13. varied with the latitude

Reply: We deleted "also"

37) P10, L15. . ( 201.8 cm) :here as elsewhere in the text you can should round the snow depth to the nearest centimeter, as this is the probably accuracy of the measurements.

Reply: The maximum value is calculated using the average values of annual maximum snow depth. The average is kept one decimal place, so it is an approximation.

38) P11, L4.... in northern Siberia. Remove extra space between in and northern

Reply: We removed the extra space.

39) P11, L21. ...the increasing rate of snow depth. increasing rates of snow depth

Reply: We replaced "rate" with "rates"

40) P11. L19-22: linear trends and results plotted on Figure 4: were trends computed on annual anomalies or on the wavelet filtered series? You must provide a trend test that accounts for the autocorrelation of time-series (see main comment).

Reply: The trends were computed on the annual anomalies. Any trend analysis is an approximate and simple approach to get what has happened on average during the study period. Linear trend analysis provides an average rate of this change. Despite there is a nonlinearity, the linear trend analysis is also a useful approximation when a systematic low-frequency variations emerged. The linear trend coefficient of snow depth was calculated to represent the rate of change at each station. The Student T test was used to assess the statistical significant of the slope in the linear regression analysis and the partial correlation coefficients, and the confidence level above 95% was considered in our study. Meanwhile, to overcome the strong assumption in ordinary least squares (independent and normal

distribution), we also added the Mann-Kendall (MK) test to identify the monotonic trend in snow depth. Confidence level above 95% was used to determine the statistically significant increase or decrease in snow depth. These two test methods could provide more robust and comprehensive information of the trend analysis. We have added the MK test and compare the results of the two methods:

"The Mann-Kendall statistical curves of annual and maximum snow depth were consistent with the linear trend analysis (Fig. 5). The increasing trend of annual snow depth reached to the 0.05 confident level in the late 1980s and from the early 1990s to the mid-1990s; it reached to the 0.01 confident level in the late 1990s. The decreasing trend reached to the 0.05 confident level from the early 2000s through the mid-2000s. The intersection of the UF curve and UB curve appeared in the mid-1970s, it indicated that the rising trend was an abrupt change during this period. The abrupt change point of the maximum snow depth was in the mid-1980s, then it increased significantly ($p \leqslant 0.05$) from the early 1990s through the mid-1990s, and it reached to the 0.01 confident level from the late 1990s to the early 2010s."

[Figure]

Figure 5. Mann-Kendall statistical curve of annual mean snow depth (a) and maximum snow depth (b) from 1966 through 2012 across the Eurasian continent. Straight line presents significance level at 0.05.

"In order to identify the monotonic trend in monthly snow depth, we conducted the MK test (Fig. 7). In October, snow depth represented a decreasing trend and it reached to the 0.05 confident level only after 2010. The statistically significant changes of monthly snow depth in November during the period of the late 1980s through the early 2000s, though it was not statistically significant with the linear regression. From December through March, there were increasing trends in monthly snow depth and the abrupt change point appeared in the mid-1970s. In the linear regression analysis, the variation of snow depth was not significant in December. However, the results of M-K test showed that the increasing trend of monthly snow depth reached to the 0.01 confident level during the mid-1980s through the late 1990s, and then it decreased during the 2000s. From January to March, monthly snow depth increased significantly ($p \leq 0.01$) from the mid-1980s to the early 2010s. In April, the statistically significant increase was found from the late 1990s to the late 2000s, and it reached to the 0.01 confident level after 2000. Consistent with the linear regression, the trend in monthly snow depth was not significant in May."

[Figure]

Figure 7. Mann-Kendall statistical curve of monthly mean snow depth (from October to May) from 1966 through 2012 across the Eurasian continent. (a) October, (b) November, (c) December, (d) January, (e) February, (f) March, (g) April, (h) May. Straight line presents significance level at 0.05.

41) P12, L1. There was a sharp increase of 3.5 cm in the maximum snow depth during the 1970s, then fluctuated from the late 1970s to the early 1990s. : then fluctuations from... Perhaps be more precise: what type of fluctuation?

Reply: There is no significant increase or decrease trend in fluctuation. We revised it as "then it fluctuating changed from the early 1990s through the early 2010s."

42) P12, L20. the rate of increase being about 0.6 cm decade

Reply: We replaced "was" with "being".

43) P14, L2. in monthly mean snow depth decreased,

Reply: We replaced "fell" with "decreased".

44) P14, L3. Changes in monthly mean snow depth were consistent with the trends in winter over the former USSR but more stations with the decreasing trends in the southern Siberia. Do you mean: 'but more stations with decreasing trends were found in southern Siberia'?

Reply: We revised it with "but more stations with decreasing trends were found in southern Siberia"

45) P14, L5. There were few stations with statistically significant trends of snow depth across China; for these, monthly snow depths tended to decrease at most stations.

Reply: We added "; for these,"

46) P14, L11. To explore the spatial variability of snow depth,

Reply: We replaced "features" with "variability".

47) P14, L15. snow depth to the north of 40° N

Reply: We deleted "mean".

48) P14, L23. because a snow depth. remove extra space between 'a' and 'snow'

Reply: We deleted the extra space.

49) P14. This result indicates that elevation is an important factor affecting snow depth in these regions. I find this statement and the preceding analysis a bit over simplistic. At large scales the snow cover can be tought to depend on latitude, altitude and distance to moisture source (continentality). I feel you are missing the third factor in our analysis. The poor, and generally negative relationship between elevation and snow depth is interesting because it is contrary to what would be expected from orographic effects on precipitation amounts and phase. What you show is that the high elevation of the TP does not cause larger snow depth compared to surrounding lower lands. Continentality seems to be the main driving factor here: the TP is in the rain shadow of the Himalaya and as

such is moisture-deprived. This should be better discussed, and analysed in the paper. This effect could be investigated, perhaps using a simple continentality index (e.g. http://glossary.ametsoc.org/wiki/Continentality). These indices rely on temperature annual ranges. You could use the closest distance to coast as another simple index.

Reply: We indeed conducted analysis on impact of continentality on snow depth over the study area. To our surprise, the correlation between the two are not significant. However, we strongly agree with the reviewer that at local and regional (less than continental) scales, topography can play key roles. In Section 4.2, we added more discussions about topographical effects on snow depth, especially on Tibetan Plateau (TP). The Tibetan Plateau's largest snow accumulation occurred in the winter, but the snowfall during winter months is the smallest of the year. This was mainly due to the terrain factors: the water vapor from the east and west was blocked by the Hengduan Mountains and Nyainqentanglha Mountains, respectively, which resulted in less snowfall. Although there was more snowfall in spring, snow cover was not easy to accumulate with higher temperatures. Therefore, snow depth was shallow on TP in general. In addition to topographic factors, spatial distribution of snow depth was also affected by atmospheric circulation. We will discuss this issue in the future studies.

50) P15: section 3.4.
You begin the section by stating that ' Variations in snow depth are closely related to climate change'. But what is investigated is the influence of spatially variable climate factors (mean temperature and mean snowfall) on snow depth, and NOT the effect of time-varying climate on snow depth. To do so you would have to test the influence of changing temperatures and snowfall/precipitation on snow depth over time. Rephrase the introduction of the section to clearly explain that you investigate spatial relationships between mean temperature and snowfall on mean snow depths. The spatial relationship between air temperature and snow depth will be undoubtedly complex when considered an area as big and topographically diverse as the Eurasian continent. Your analysis is till interesting as it shows that snowfall is the main factor driving spatial variability in snow depth, at this spatial scale. However snowfall rates and air temperature must also be somewhat correlated, as snowfall depends on precipitation and temperature (precipitation phase). I suggest that you also calculate and report the partial correlation coefficients, i.e. to show the influence one variable while removing effect of the other, on snow depth. You do examine the effect of changing climate on snow depth and mass in Figure 10 for the composite Eurasian and Russian records. This analysis is qualitative and hile interesting and valuable, it could be enriched by calculating and presenting the correlation coefficients between series. Especially for the SWE series, how much of the variance can be respectively explained by air temperature and snowfall? Even more constructive would be to perform this analysis on a station basis and map the results. We would then learn about the spatially variable climate control on snow.

Reply: Thank you very much for your suggestion. We have added the quantitative analysis by the partial correlation analysis between snow depth, SWE, air temperature and snowfall. The results showed that the significant negative correlation ($p \leq 0.05$) between snow depth and air

temperature presented in most areas of European Russia and the southern Siberia (Fig 13a). The stations with negative effects of air temperature on SWE were less, and there were no statistically significant correlation in the northern Siberia (Fig 13b). It was because the air temperature was below 0℃ in most areas of Siberia during December through March, the increasing temperature did not have an obvious effect on snow depth. Consistent with the interannual variation, changes in snow depth and SWE were more affected by snowfall in most areas across the former USSR from December through March. The greater partial correlation coefficients (>0.6) between snow cover and snowfall appeared in the northern European Russia, the southern Siberia, the northeast and southeast of the Russian Far East. Variations in snow depth and SWE were more sensitive to snowfall and snowfall rate in these areas. Variations of snow depth were explained by air temperature and snowfall in most areas of the European Russia and some regions of the southern Siberia, the effects of the two factors on SWE only appeared in some of these areas; however, snowfall was the main driver force of the variance of snow depth and SWE in the former USSR.

[Figure]

**Figure 13.** Spatial distributions of partial correlation coefficients of snow depth and air temperature (a), snow depth and snowfall (b), SWE and air temperature (c), SWE and snowfall from November through March during 1966-2009. The coefficients reaching to 0.05 confident level are displayed. Red circles represent a negative relationship, and blue circles indicate a positive relationship.

51) P15, L7. Snow depth significantly decreases with increasing air temperature (P≤0.05),

Reply: We replaced "decreased" with "decreases"

52) P15, L17. Increases

Reply: We replaced "increases" with "increasing trends"

53) P15, L21. The significant increasing snowfall can explain the sudden drop in snow density observed from the mid-1990s through the early 2000s (Zhong et al., 2014): fresh snow with low snow density. Explain the last statement better in a separate statement. Why does increasing snowfall decreases snow density? Is it the mean density of the snowpack? Increasing snowfall in response to warmer temperature should increase the density, both of fresh snow, and perhaps of the whole snowpack due to faster metamorphism and increased compaction...

Reply: The snow density means the bulk snow density of the snow profile. Increasing snow fall should decrease the density of the surface snowpack, which lowed the whole density of snowpack.

54) P15, L26. increasing trend of changes in snow depth. trend in snow depth? or trend in the rate of change?

Reply: It means the trend in snow depth. We deleted "of changes".

55) P15, L27. In fact, the climatology of snow depth not only influenced by air temperature and precipitation, but also with other climatic factors and atmospheric circulation. Poor formulation, rephrase.

Reply: We have deleted the sentences and added the analysis of the partial correlation coefficients between snow cover and air temperature, as well as snow cover and snowfall.

"The partial correlation coefficients between snow cover and air temperature, as well as snow cover and snowfall were calculated to discuss the spatial relationship between them (Fig. 13). The significant negative correlation (p≤0.05) between snow depth and air temperature presented in most areas of European Russia and the southern Siberia (Fig 13a). The stations with negative effects of air temperature on SWE were less, and there were no statistically significant correlation in the northern Siberia (Fig 13b). It was because the air temperature was below 0℃ in most areas of Siberia during December through March, the increasing temperature did not have an obvious effect on snow depth.
Consistent with the interannual variation, changes in snow depth and SWE were more affected by snowfall in most areas across the former USSR from December through March. The greater partial correlation coefficients (>0.6) between snow cover and snowfall appeared in the northern European Russia, the southern Siberia, the northeast and southeast of the Russian Far East. Variations in snow depth and SWE were more sensitive to snowfall and snowfall rate in these areas."

56) P16. L7. These discrepancies may result from differences in the time frame

Reply: We replaced the first "differences" with "discrepancies".

57) P16. L26. during the different study periods.

Reply: We revised it.

58) P16. L.26-28. The sensitivity of snow cover to air temperature and precipitation for each station showed regional differences (Fallot et al., 1997; Park et al., 2013). The amount of snowfall can be affected by climate change, and leading to differences in snow depth at different times (Ye et al., 1998; Kitaev et al., 2005). This is why simple spatial relationship between air temperature and snow depth do not exist...

Reply: We added a sentence after it.

"The results of our study showed that there was significant negative relationship between snow depth and air temperature in the southern Siberia, however, it did not exist in the northern Siberia. This may explain the difference in the results of these studies."

59) P17, L5. and is the main reason. Extra space between 'and' and 'is'

Reply: We deleted the extra space.

60) P17, L10. Therefore, we will select a typical climate zone to research the climatology and variations of snow cover. This rather vague... your study looks at large scale control on snow cover and this is what this dataset allows. Studying small scale (topography, vegetation) effects on the snow cover requires other kind of data, sampled at a higher spatial resolution. I would remove this sentence.

Reply: Thank you very much for your suggestion. We removed the sentence.

61) You should better discuss your results in the light of what is known about large-scale control on snow cover: latitude, altitude and continentality are the main geographical factor which drive snowfall rates and hence snow depths. I find your analysis and the discussion on page 17 somewhat incomplete in this respect.

Reply: We have analyzed the relationship between snow depth and continentality. But the correlation coefficient was not high (r=0.1). This indicated that the continentality is not an important driving factor of snow cover climatology over Eurasia. However, we strongly agree with the reviewer that at local and regional (less than continental) scales, topography can play key roles. In Section 4.2, we added more discussions about topographical effects on snow depth, especially on Tibetan Plateau (TP):

"Some important questions that are not addressed in the current research should be resolved in the future. Topography is an important factor affecting the climatology of snow depth, and

is the main reason causing the inhomogeneity of data. Previous studies have analyzed the representation of snow depth for single stations to solve the issue (Grünewald and Lehning, 2011, 2013; Grünewald et al., 2014). However, in the present study, we did not discuss this question because of the complexity of spatial difference. But we still got some interesting conclusions: There was a closely relationship between snow depth and elevation at the local scale. However, compared with latitude, the correlation between them was not so significant in the whole Eurasian Continent. Moreover, the continentality did not play a great role in spatial distribution of snow depth, especially on TP. The previous studies showed that the Tibetan Plateau's largest snow accumulation occurred in the winter, but the snowfall during winter months is the smallest of the year (Ma, 2008). This was mainly due to majority of annual precipitation occurs during the summer monsoon season on TP which cause very less snowfall during winter half year (or snow accumulated season). Furthermore, the water vapor from the east and west was blocked by the Hengduan Mountains and Nyainqentanglha Mountains, respectively, which resulted in less snowfall. Although there was more snowfall in spring, snow cover was not easy to accumulate with higher temperatures. Therefore, snow depth was shallow on TP in general. In addition to topographic factors, spatial distribution of snow depth was also affected by atmospheric circulation. We will discuss this issue in the future studies."

---

## Referee Report (RR1)

**Response to Referee #1**

**General Comments:**

1) This study examines the characteristics and trends across the Eurasian continent from 1966 to 2012. To do so, the authors assemble snow depth data from 1103 stations across the study area. How representative are the station (point) snow depth data of the overall regional landscapes of interest? For instance, are snow depth data in forested areas collected at airports or other open areas, that may not represent the regional snow characteristics?

Reply:Thank you for your comments and concerns. The spatial representativeness of stations is always a key and difficult problem in snow depth research or any ground-based studies at various regional scales. In fact, we did not do spatial interpolation of snow depth using these in-situ data across the study area just because the uneven distribution of stations spatially and among different landscapes. The passive microwave remote sensing snow depth products may mitigate the regional coverage problem, their low spatial resolution (25×25 km) and high uncertainties (up to 200%) provide no better help to the issue. The combination of in-situ snow depth data with satellite remote sensing snow depth will be a better approach but it is out of the scope of this study. Here for the first time, we present all data we can possibly collect from various countries over the continent and show snow depth spatial variations and temporal changes. We are fully aware the shortcomings of station distribution but this in-situ dataset and its coverage is unprecedented. We may read a lot of published literatures regarding snow cover extent in regional or hemispheric scales, but not snow depth. In this study, we present spatial and temporal changes in snow depth using available in-situ data.

2) Further to this, snow course data from the former USSR are also employed in establishing the snow depth climatology (see Section 2). Is it therefore a fair comparison to present the station (point) data with those from local (spatially averaged) data?

Reply: In our study, 440 stations have both snow course and station data. We compared the snow course averaged data and the station point data and found that they were statistically significantly correlated, and the goodness of fit reached to 94% (Fig. 1). Therefore, we are confident that it was a fair practice to combine the snow course average data and the station point data together in this study.

[Figure]

Figure1. The relationship between snow depth of meteorological station and snow course at 440 stations.

3) The Introduction section is quite lengthy and could be abbreviated by focusing on past studies that report climatologies and trends in snow depth across Eurasia only and the gap being filled by the present study. Further to this, the Introduction should emphasize the novelty of this research compared to previous studies cited in the text.

Reply: We have abbreviated the introduction, and focused on the report of climatologies and trends in snow depth, the existing problems of the previous studies, as well as the characteristics of our study.

4) The authors should consider the Mann-Kendall test to assess linear trends or other non-parametric trend analysis rather than linear regressions.

Reply: Any trend analysis is an approximate and simple approach to obtain what has happened on average during the study period. Linear trend analysis provides an average rate of this change. Despite there is a nonlinearity, the linear trend analysis is also a useful approximation when a systematic low-frequency variations emerged. Meanwhile, to overcome the strong assumption in ordinary least squares (independent and normal distribution), we added a Mann-Kendall (MK) test to identify the monotonic trend in snow depth. These two test methods could provide more robust and comprehensive information of the trend analysis. We have added the method introduction in the "data and methodology" section and discussed the similarities and differences of the two kind of trend analysis results in the "results" section.

"Any trend analysis is an approximate and simple approach to get what has happened on average during the study period. Linear trend analysis provides an average rate of this change. Despite there is a nonlinearity, the linear trend analysis is also an useful approximation when a systematic low-frequency variations emerged. (Folland and Karl, 2001; Groisman et al., 2006). The linear trend coefficient of snow depth was calculated to represent the rate of change at each station. The Student T test was used to assess the statistical significant of the slope in the linear regression analysis and the partial correlation coefficients, and the confidence level above 95% was considered in our study. Meanwhile, to overcome the strong assumption in ordinary least squares (independent and normal distribution), we applied a Mann-Kendall (MK) test to identify the monotonic trend in snow depth. Confidence level above 95% was used to determine the statistically significant increase or decrease in snow depth. These two test methods could provide more robust and comprehensive information of the trend analysis."

"The Mann-Kendall statistical curves of annual and maximum snow depth were consistent with the linear trend analysis (Fig. 5). The increasing trend of annual snow depth reached to the 0.05 confident level in the late 1980s and from the early 1990s to the mid-1990s; it reached to the 0.01 confident level in the late 1990s. The decreasing trend reached to the 0.05 confident level from the early 2000s through the mid-2000s. The intersection of the UF curve and UB curve appeared in the mid-1970s, it indicated that the rising trend was an abrupt change during this period. The abrupt change point of the maximum snow depth was in the mid-1980s, then it increased significantly ($p \leqslant 0.05$) from the early 1990s through the mid-1990s, and it reached to the 0.01 confident level from the late 1990s to the early 2010s."

[Figure]

Figure 5. Mann-Kendall statistical curve of annual mean snow depth (a) and maximum snow depth (b) from 1966 through 2012 across the Eurasian continent. Straight line presents significance level at 0.05.

"In order to identify the monotonic trend in monthly snow depth, we conducted the MK test (Fig. 7). In October, snow depth represented a decreasing trend and it reached to the 0.05 confident level only after 2010. The statistically significant changes of monthly snow depth in November during the period of the late 1980s through the early 2000s, though it was not statistically significant with the linear regression. From December through March, there were increasing trends in monthly snow depth and the abrupt change point appeared in the mid-1970s. In the linear regression analysis, the variation of snow depth was not significant in December. However, the results of M-K test showed that the increasing trend of monthly snow depth reached to the 0.01 confident level during the mid-1980s through the late 1990s, and then it decreased during the 2000s. From January to March, monthly snow depth increased significantly ($p \leqslant 0.01$) from the mid-1980s to the early 2010s. In April, the statistically significant increase was found from the late 1990s to the late 2000s, and it

reached to the 0.01 confident level after 2000. Consistent with the linear regression, the trend in monthly snow depth was not significant in May."

[Figure]

Figure 7. Mann-Kendall statistical curve of monthly mean snow depth (from October to May) from 1966 through 2012 across the Eurasian continent. (a) October, (b) November, (c) December, (d) January, (e) February, (f) March,

(g) April, (h) May. Straight line presents significance level at 0.05.

5) Do the linear trends reported in Section 3.2 exceed the variability in the snow depth data? In other words, are there "detectable" trends in snow depth, i.e. with the signal greater than the noise in the system?

Reply: The Student T test was used to assess the statistical significant of the slope in the linear regression analysis, and the confidence level above 95% was considered in our study. We have tested the results of the linear trends in Section 3.2, and the results show that all of the "detectable" trends in snow depth were greater than the noise in the system.

6) All figures are rather small and difficult to interpret when printed on paper.

Reply: Thank you for pointing this out. We have expanded all figures.

**Specific Comments:**
1) P. 1, line 21: Insert "a" before "snow depth". Then insert "its" before "spatiotemporal".

Reply: Has been done.

2) P. 1, line 27: Consider a word other than "dramatically" here. Are these statistically significant trends?

Reply: We replaced it with "significantly". In our study, the trends with the confidence level above 95% were only considered.

3) P. 3, lines 10-20: Note that the tense for verbs changes throughout this paragraph.

Reply: We replaced "is" with "was" in line 15, and replaced "promotes" with "prompted" in line 19.

4) P. 3, lines 22-24: Are the soil thermal conditions reported here for winter only?

Reply: Yes, the soil thermal conditions are in winter.

5) P. 4, line 8: Delete "the" before "ecological".

Reply: Has been deleted.

6) P. 5, line 8: Delete "In order" and begin the sentence with "To obtain…"

Reply: Thank you for your suggestions. We revised it.

7) P. 6, lines 4-8: Delete "Using data from ground-based measurements" as this repeats text from the previous sentence. Also, please rephrase the statement "a detailed description of snow depth", as this suggests the paper goes at length in describing how snow depth is defined, which is not the case. This entire sentence is awkward and quite long, so should be rephrased and perhaps divided into two sentences.

Reply: Thank you for your suggestions. We rephrased the sentence: "The objective of this study is to investigate the climatology and variability of snow depth, and analyze snow depth relationships with the topography and climate factors over the Eurasian continent from 1966 to 2012."

8) P. 6, line 14: Snow depth data from 17 countries are apparently used in the present study; yet Table 1 lists only three countries (former USSR, Mongolia and China) as sources for the snow depth data.

Reply: Seventeen countries includes China, Mongolia and 15 countries previously belonged to the former USSR. In order to avoid the readers' confusion, we deleted "in 17 countries".

9) P. 6, line 15: Insert "a" before "daily".

Reply: We inserted it.

10) P. 6, line 18: Replace "5" with "five".

Reply: We revised it.

11) P. 6, line 22: "SWE" has not yet been defined.

Reply: We have defined SWE at P. 3, line 7.

12) P. 8, line 2: Delete "In order" and start the sentence with "To reflect…"

Reply: We revised it.

13) P. 9, line 8: What is a "scale gram"?

Reply: We deleted "gram".

14) P. 9, line 15: Delete extra spaces before "from".

Reply: We deleted it.

15) P. 10, line 11: "TP" is not defined.

Reply: We have defined TP at P. 5, line 7.

16) P. 11, line 4: Delete extra space before "northern".

Reply: We deleted it.

17) P. 12, line 2: Insert "it" before "fluctuated".

Reply: Thank you very much for your suggestion. We inserted it.

18) P. 12, line 7: Change to "decreasing trend".

Reply: We revised it.

19) P. 12, line 25: Rephrase "fluctuant trend".

Reply: We inserted "increasing" before "trend".

20) P. 13, line 7: Delete "variability" before "trends".

Reply: We deleted it.

21) P. 13, line 25: Delete the space in "95%".

Reply: We deleted it.

22) P. 14, line 23: Delete the extra space before "snow".

Reply: We deleted it.

23) P. 15, line 2: Variations in hydrometeorological quantities such as snow depth are due to climate variability, not climate change.

Reply: Thank you very much for your suggestion. We replaced "climate change" with "climate variability".

24) P. 15, line 7: Here reports of significant declines in snow depth are provided, while the abstract (line 27) suggests the opposite pattern is being observed – which is correct?

Reply: Here the result showed the relationship between snow depth and air temperature. There was a negative correlation between them. Increasing air temperature result in the snow depth decreased. However, in the abstract, the increasing trend represented the interannual variation in snow depth. They are different.

25) P. 15, line 18: Change to "increased".

Reply: We revised it.

26) P. 15, line 27: Insert "is" before "not".

Reply: We inserted it.

27) P. 16, lines 7-8: "differences" is used twice in succession.

Reply: We replaced the first "differences" with "discrepancies"

28) P. 17, line 5: Delete the extra space before "is the".

Reply: We deleted it.

29) P. 22, line 20: This should read "Liston".

Reply: Thank you for your suggestions. We revised it.

30) P. 26, Figure 1: The colors highlighting three regions (Sakhalin, Kamchatka Peninsula, and northern Xinjiang Autonomous region) are nearly indistinguishable. Please consider using colors of greater contrast. Why are these regions highlighted in the first place? A number of abbreviations are used on the map that are not defined in the figure caption (this is an issue in other figures as well).

Reply: We highlighted the three regions due to snow depths were greater in these areas. We wanted to indicate the accurately locations for readers who are not familiar with the geography of Eurasia. We have canceled the highlight because it may cause confusion for the reader. The country abbreviations were used because the space is limited and cannot be spelled out. We have spelled out the names of countries as an annex.

Abbreviation Description

| Country | Abbreviation |
| --- | --- |
| Kazakhstan | KAZ |
| Ukraine | UKR |
| Turkmenistan | TKM |
| Uzbekistan | UZB |
| Tajikistan | TJK |
| Belarus | BLR |
| Estonia | EST |
| Georgia | GEO |
| Latvia | LVA |
| Lithuania | LTU |

| Azerbaijan | AZE |
| Kyrgyzstan | KGZ |
| Moldova | MDA |

31) P. 27, Figure 2: Given the high number of sites with high average snow depth values in the northern reaches of the Eurasian continent, would the results be better depicted using contour lines instead? Consider adding the latitudinal averages of the snow data as secondary diagrams to these figures.

Reply: We tried to use the contour lines instead of the point values. But we found that there was a problem of the accuracy of interpolation with Kriging interpolation in AcrGIS, in which there was snow cover in some no snow areas. Therefore, the results cannot be depicted using contour lines instead. Snow depth distributions are affected by the topographic factors over the Eurasia. Snow depth is also affected by elevation, slope, aspect in the same latitude. The latitudinal average of snow data cannot fully reflect the snow depth distribution.

32) P. 30, Figure 4: Does the number of stations used in the composite snow depth anomalies vary over time? The statistical significance of the trends should use the symbol "$\leqslant$" rather than "<=". Why does the last sentence in the figure caption mention "simulation" of snow depth?

Reply: Thank you for your suggestions. The number of stations used in the composite snow depth anomalies vary over time. First, we calculated the snow depth anomaly at each site, and then took the average of the anomalies as the general anomaly. We replaced "<=" with "$\leqslant$". In the figure, X means the value of the linear regression trend, which is calculated by the snow depth anomaly with linear regression. Therefore, it is a calculation of snow depth.

33) P. 31, Figure 5: See comments for Figure 4.

Reply: We replaced "<=" with "$\leqslant$".

34) P. 34, Figure 8: The statistical significance of the linear regressions should use the symbol "$\leqslant$" rather than "<=". Are any of the stations below sea level? If not, panels (b) and (c) should have their x-axes begin at 0 m in elevation. The caption should also state that these are relationships between snow depth and latitude and elevation, not changes.

Reply: Thank you for your suggestions. We replaced "<=" with "$\leqslant$". There are 9 stations below sea level in the former USSR. We revised the caption: "The relationships among annual mean snow depth, air temperature and snowfall for 386 stations from November through March during 1966-2009 over the USSR. The thick line is a linear regression trend."

35) P. 35, Figure 9: See comments for Figure 8.

Reply: We replaced "<=" with "$\leqslant$".

36) P. 36, Figure 10: See comments for Figure 8.

Reply: We replaced "<=" with "$\leqslant$".

**Response to Referee #2**

**General Comments:**

1) Methods to measure snow depth. The authors describe the snow course data, how is snow depth/snowfall measured at other stations? Is the method similar in the different countries?

Reply: All measurements at the meteorological stations across the Eurasian continent was set at the same standard by the former USSR after the World War II, including China in the 1950s. All snow depth measurements at these meteorological stations were conducted with the same kind of instruments, the same standard, and the same local time. Therefore, snow depth at all meteorological stations are all consistent. Snow courses were established and operated the same way as the snow depth measurements.

Choice of time window for analysis. The snow year was defined from July 1st to June 30th. Why this choice of period? Figure 3 shows that snow seems to begin accumulating in September but snow remains in June in northern Russia and the Tibetan Plateau. So the chosen analysis window may not be optimal and should probably begin, and end, later so as to capture the complete seasonal cycle over the studied area.

Reply: We checked that the snow cover extent is the smallest in July and August. We also conducted analysis about snow depth in July and snow was essentially gone based on the data we have. Snowfall can happen anytime of the year on the Qinghai-Tibetan Plateau but it only lasts from a few hours to a few days during summer months such as June through August. We believe it is safe to say that a snow-year from June 30 through July 1 would capture the entire seasonal snow cycle.

Trend detection. Trend detection in environmental time-series is a delicate topic and this is a big concern for this study, as the 'significant' trends reported could be cited in future works. The authors seem to have used ordinary linear regression (OLS) with classical hypothesis tests (Fisher or 'F-test' on the variance explained, and/or Student T test on regression coefficients). These parametric tests make the assumption that the data is normally and independently distributed. The authors have not reported on checking these assumptions, and I doubt that the time series presented in the figures are free of autocorrelation. As a result I question the validity of much of the 'significant' linear trends reported in this study and suggest the author should apply a statistical test which account for the serial correlation of time-series. A suggestion is given below.
If the data are normally distributed, OLS can be used but the degree of freedom for the significance test must be adjusted for the reduction in the degree of freedom caused by the auto- ('serial) correlation. If the data is not normally distributed, transformation or a non-parametric test is necessary. The Mann-Kendall trend test is commonly used on

non-normal data. Here again serial correlation must accounted for. The authors could quickly apply a normality test and the Durbin-Watson statistic to the residuals of their regression to diagnose these problems. One possible approach to take the autocorrelation into account using OLS is outlined in Weatherhead et al. (1998):

Weatherhead, E. C., et al. (1998), Factors affecting the detection of trends: Statistical considerations and applications to environmental data, J. Geophys. Res., 103(D14), 17149–17161, doi:10.1029/98JD00995.

Another possibility is to apply pre-whitening to the time series. A pertinent paper is:

Yue, S., Pilon, P., Phinney, B. and Cavadias, G. (2002), The influence of autocorrelation on the ability to detect trend in hydrological series. Hydrol. Process., 16: 1807–1829. doi:10.1002/hyp.1095 ï´C ˘g Wavelet analysis (P8, L26-30 and P9, L1-20)

Reply: Thank you very much for your comments and suggestions. Any trend analysis is an approximate and simple approach to obtain what has happened on average during the study period. Linear trend analysis provides an average rate of this change. Despite there is a nonlinearity, the linear trend analysis is also a useful approximation when a systematic low-frequency variations emerged (Folland and Karl, 2001; Groisman et al., 2006). The linear trend coefficient of snow depth was calculated to represent the rate of change at each station. We checked and found that the data were normally distributed. The Student T test was used to assess the statistical significant of the slope in the linear regression analysis and the partial correlation coefficients, and the confidence level above 95% was considered in our study. Meanwhile, to overcome the strong assumption in ordinary least squares (independent and normal distribution), we also added the Mann-Kendall (MK) test to identify the monotonic trend in snow depth. Confidence level above 95% was used to determine the statistically significant increase or decrease in snow depth. These two test methods could provide more robust and comprehensive information of the trend analysis.

Folland, C.K. and Karl, T.R. (2001), Observed climate variability and change. Climate Change 2001: The Scientific Basis, edited by: J. T. Houghton et al., Cambridge University Press, Cambridge, UK, 99–181.

Groisman, P. et al. (2006), State of the Ground: Climatology and Changes during the Past 69 Years over Northern Eurasia for a Rarely Used Measure of Snow Cover and Frozen Land, J. Climate, 19, 4933–4955.

The description of the wavelet analysis is very confusing and seems unnecessarily complicated, due to their limited role in the paper. It does not allow the reader to understand what was done to the data and to replicate the analysis. This section really cuts the flow of reading and should be reworked altogether in order to bring out the essential, with proper supporting references. Which wavelet transform was used in the end, a continuous or discrete? Which wavelet family/filter? From my understanding of this paragraph you applied a discrete wavelet transform, excluded the high-frequency components and then used the inverse transform to reconstruct the lower frequency signal. Or is it that you applied an averaging filter on the wavelet coefficient before reconstructing the signal with the inverse wavelet

transform?

2) Results and discussion

Physiographic and climatic control on spatial variability of snow depths. The analysis of the factors controlling the spatial variability of snow depths is somewhat limited in breadth. The authors show a general increase in snow depth with latitude and a general decrease in snow depth with increasing altitude. Large-scale control on snow depth will be mainly dependent on the interplay between latitude and altitude but also distance to moisture sources (continentality) and position relative to orographic barriers. Together these will determine the snowfall rate. For such a large and topographically contrasted region as the Eurasian continent, ignoring these last two effects does not allow a clear understanding of large-scale snow depth spatial variability in the region. The negative (but poor) relationship between snow depth and altitude shown by the authors is largely explained by the continentality and rain shadow effect of the high Tibetan Plateau, while at higher latitude snow depth does seem to increase with altitude in response to orographic enhancement of snowfall. The authors should try to incorporate quantitatively the effects of continentality and barrier effect into their analysis, or at least provide a more in depth discussion of their results in the light of known large-scale physiographic control on the snow cover, with proper supporting references.

Reply: Thank you very much for your comments and suggestions. We indeed conducted analysis on impact of continentality on snow depth over the study area. To our surprise, the correlation between the two are not significant. However, we strongly agree with the reviewer that at local and regional (less than continental) scales, topography can play key roles. In Section 4.2, we added more discussions about topographical effects on snow depth, especially on Tibetan Plateau (TP). The Tibetan Plateau's largest snow accumulation occurred in the winter, but the snowfall during winter months is the smallest of the year. This was mainly due to the terrain factors: the water vapor from the east and west was blocked by the Hengduan Mountains and Nyainqentanglha Mountains, respectively, which resulted in less snowfall. Although there was more snowfall in spring, snow cover was not easy to accumulate with higher temperatures. Therefore, snow depth was shallow on TP in general. In addition to topographic factors, spatial distribution of snow depth was also affected by atmospheric

circulation. We will discuss this issue in the future studies.

Relationships between mean (Eurasian) climate and snow depth over time.
The authors revealed interesting increases in snow depth, SWE, temperature and snowfall rates over the study period. However the analysis and interpretation of these tendencies remains somewhat superficial. This section could be enhanced by quantitative analysis, i.e. by performing correlation analysis, and/or multiple correlation/regression analysis to highlight the respective influence of temperature and snowfall changes on mean Eurasian snow depth and SWE. Even more interesting would be to see this analysis done spatially, perhaps in a future study. This would probably highlight the effect of continentality and position relative to orographic barriers on the response of the snow cover to climate.

Reply: We have added the quantitative analysis by the partial correlation analysis between snow depth, SWE, air temperature and snowfall. The results showed that the significant negative correlation ($p \leq 0.05$) between snow depth and air temperature presented in most areas of European Russia and the southern Siberia (Fig 13a). The stations with negative effects of air temperature on SWE were fewer, and there were no statistically significant correlation in the northern Siberia (Fig 13b). It was because the air temperature was below 0℃ in most areas of Siberia during December through March, the increasing temperature did not have an obvious effect on snow depth. Consistent with the interannual variation, changes in snow depth and SWE were more affected by snowfall in most areas across the former USSR from December through March. The greater partial correlation coefficients (>0.6) between snow cover and snowfall appeared in the northern European Russia, the southern Siberia, the northeast and southeast of the Russian Far East. Variations in snow depth and SWE were more sensitive to snowfall and snowfall rate in these areas.

**Specific comments and editorial changes:**
1) P3, L13. Although snow cover extent reduced with climate warming, snow depth still increased in northern Eurasia (Kitaev et al., 2005; Bulygina, 2011). Over which period?

Reply: The sentence is revised as "Although snow cover extent reduced with climate warming, snow depth still increased in the northern Eurasia during 1936 to 1995 (Kitaev et al., 2005) and 1966-2010 (Bulygina et al., 2011)."

2) P3 L23, 'a' thin snow cover results...

Reply: We deleted the third paragraph in P.3 and the second paragraph in P.4 according to the suggestion of referee #1.

3) P3 L24. Frauenfeld et al. (2004) indicated that in permafrost areas the maximum snow depth by the end of winter has a significant influence on the active layer depth during the following summer.

Reply: We deleted the third paragraph in P.3 and the second paragraph in P.4 according to the suggestion of referee #1.

4) P3, L28. The numerical modeling results showed that the rate of mean annual ground surface temperature increase with the increasing maximum snow depth was about 0.1℃ cm-1 for the maximum snow depth at 15 cm. This sentence is convoluted and hard to understand, please rephrase more clearly. Maybe: The numerical modeling results showed that the mean annual ground surface temperature increases with increasing snow depth at a rate of 0.1 _C cm-1 until up to a snow depth of 15 cm...?

Reply: We deleted the third paragraph in P.3 and the second paragraph in P.4 according to the suggestion of referee #1.

5) P4, L2, ...also increased with snow depth.

Reply: We deleted the third paragraph in P.3 and the second paragraph in P.4 according to the suggestion of referee #1.

6) P4, L5: Furthermore, snow accumulation an important freshwater resources and has direct impacts on the hydrological cycle.

Reply: We deleted the third paragraph in P.3 and the second paragraph in P.4 according to the suggestion of referee #1.

7) P4, L11. Adam et al. (2009) suggested that the variations in snow depth will significantly affect the hydrological regime of the Arctic in the future.

Reply: We deleted the third paragraph in P.3 and the second paragraph in P.4 according to the suggestion of referee #1.

8) P4, L14-26: here you describe trends in snow cover and other variables. Please mention the period over which these changes were observed for respective studies

Reply: Thank you very much for your suggestion. We added the period over these changes for respective studies:

"overall, the annual mean snow depth decreased in most areas over North America during 1946 to 2000 (Brown and Braaten, 1998; Dyer and Mote, 2006), and increased in Eurasia and the Arctic during the recent 70 years (Ye et al., 1998; Kitaev et al., 2005; Callaghan et al., 2011a; Liston and Hiemstra, 2011) but there was regional differences (Bulygina et al., 2009, 2011; Ma and Qin, 2012; Stuefer et al., 2013; Terzago et al., 2014). Changes in snow depth were primarily affected by air temperature and precipitation. Ye et al. (1998) and Kitaev et al. (2005) showed that higher air temperatures caused an increase in snowfall in winter from

1936 through 1995, thus greater snow depth was observed in northern Eurasia in response to global warming. Furthermore, snow depth distribution and variation are also controlled by terrain (i.e., elevation, slope, aspect, and roughness) and vegetation (Lehning et al., 2011; Grünewald et al., 2014; Revuelto et al., 2014; Rees et al., 2014; Dickerson-Lange et al., 2015). Snow depth is also closely related to other large-scale atmospheric circulation indices, such as the North Atlantic Oscillation /Arctic Oscillation (NAO/AO) indices. For example, Beniston (1997) found that the NAO played a crucial role in fluctuations in the amount of snowfall and snow depth in the Swiss Alps from 1945 to 1994. Kitaev et al. (2002) reported that the NAO index is positively related to snow depth in the northern part of the East European Plain and over western Siberia during the period 1966-1990;"

9) P4, L28. Snow depth is also closely related to other large-scale atmospheric circulation indices, such as the North Atlantic Oscillation /Arctic Oscillation (NAO/AO) indices. For example, Beniston (1997) found that the NAO....

Reply: We have revised the sentence.

10) P5, L8. In order to obtain a wider range of snow depth...Wider range is imprecise. In order to increase the spatial coverage? and/or spatial resolution?

Reply: We revised it with "increase the spatial coverage"

11) P5, L18. Ground-based snow measurement remains the basis for verification of remote sensing and instrumental data...

Reply: We replace "is" with "remains"

12) P6, L1. 'TP' = Tibetan Plateau I presume, but I don't think it was defined before.

Reply: We have defined it in P.5, line7.

13) P6, L15. on a daily basis.

Reply: We inserted it.

14) P6, L15. Suggested change: Historical snow course data over the former USSR from 1966 to 2011 were also used in this study

Reply: We revised it.

15) P6, L17 snow surveys performed throughout the accumulation season

Reply: We revised it.

16) P6, L19. Snow surveys were conducted over 1–2 km-long transects.

Reply: We revised it.

17) P6, L20. Snow depth was measured every 10 m in the forest, and every 20 m in open terrain.

Reply: We replace "each" with "every"

18) P6, L22. SWE: define once

Reply: We have defined it in P.3, line7.

19) P6. L25 ... over the former USSR. Why only over USSR? Maybe complete sentence with ... 'where SWE data are available...'

Reply: We added it.

20) P6 L25. SWE was measured every 100 m along the 0.5-1.0 km courses and every 200m along the 2 km course.

Reply: We replace "each" with "every"

21) P6, L27. Precipitation data were divided proportionally into daily solid and liquid data, and the solid-to-liquid fraction was determined according to daily mean temperature (Brown,2000). I suggest replacing by: Daily precipitation was partitioned into a solid and liquid fraction, based on daily mean temperature (Brown,2000). You then describe the partitioning equation in the following sentence. $S_r$at : whatdoesratstandsfor…?

Reply: Thank you very much for your suggestion. We replaced this sentence.

22) P7, L9. Quality control steps. (1) daily snow depth observations (equal to or greater than 0 cm, not including missing data) for <15 days in one month were omitted; This is confusing. Do you mean that months having less than 15 days with snow depth data were omitted from the analyse? If that so rephrase in that sense. (2) snow data from stations with <20 years of measurements during 1971-2000 were excluded; I suggest replacing by: Stations with less than 20 years of data during the 1971-2000 period were excluded from the analysis. 3) data exceeding two standard deviations compared with the annual average value during 1966-2012 were omitted. Add: 'At each station,' before the sentence.

Reply: We rephrased this sentence as "Months having less than 15 days with snow depth data were omitted from the analysis"

23) P7, L16. We defined a snow year as the period from July 1st of a current year to June

30th of the following year. Why this choice of period..? maybe add short complement to the sentence: '... so as to insure that the complete seasonal snow cycle is captured across the study region...' Also, I note in Figure 3 that snow remains in June in some areas, and seems to begin accumulating in September. So the chosen analysis window may not be optimal and should probably begin and end later.

Reply: Thank you very much for your suggestion. The snow cover extent is smallest during July to August, in order to capture the beginning of snow cover, we defined a snow year as the period from July 1st of a current year to June 30th of the following year. We have added the explanation of the choice of time window before we defined the snow year.

24) P7, L17. Because the procedures for taking snow observations have changed over the course of the studied period, there were some inhomogeneities in the data.

Reply: We revised it.

25) P7, L25....World Meteorological Organization (WMO) climatological products. A reference would be needed here.

Reply: We added the reference "World Meteorological Organization (WMO) climatological products (Ma and Qin, 2012)"

26) P7, L25. A threshold of 15 days was selected because the snow cover duration in some areas of China was less than one month, and the data for 15 days' snow depth in a month were relatively stable. Do you refer to the previously defined quality control step 1? If this is the case this sentence should go in the quality control paragraph. You can here recall it in short sentence.

Reply: Thank you very much for your suggestion. We moved this sentence to the quality control paragraph.

27) P8. L2. In order to capture the primary...

Reply: We revised it.

28) P8, (4) Linear trend coefficient of snow depth: the linear trend coefficient of snow depth for each station was obtained by linear regression analysis with respect to time, and thus represents the rate of change in snow depth for a period of time. Replace 'for a period of time' by 'over time'? Or by: 'for a >20 year time period'. Statistical test on linear trend: see main comments...

Reply: We replaced it. And the analysis of the statistical test on linear trend is answered in the main comments.

29) P8. L23. ...each station and averaged the anomalies for all stations to the anomalies for the whole Eurasian continent. : 'averaged the anomalies for all stations to obtain mean anomalies for the whole Eurasian continent'.

Reply: We revised it.

30) P8, L26 -. Description of wavelet analysis. See main comment on this. You need to include at least once key reference.

Reply: Thank you very much for your suggestion. We have revised the description of the wavelet analysis. As your understanding, we applied a discrete wavelet transform, excluded the high-frequency components and then used the inverse transform to reconstruct the lower frequency signal. The new description as following:

"Wavelet analysis was performed to reveal the long-term low-frequency variations of snow depth over the study area as a whole. A wavelet is a wave-like oscillation with an amplitude that begins at 0, increases, and then decreases back to 0 (Graps, 1995). We applied a discrete wavelet transform, excluded the high-frequency components and then used the inverse transform to reconstruct the lower frequency signal."

31) P9, L12. We used an averaging filter for wavelets analysis. Using this method, values that are too small or too large may be excluded; This description is really unclear. Please simplify and add proper references so that the interested reader can fin further explanations on the technique used, if wanted.

Reply: We deleted this sentence.

32) P9, L15. obtained from filtering. Remove extra space.

Reply: We deleted the space.

33) P9, L26. increased with the latitude... A maximum annual mean snow depth... in the west of the Yenisey River

Reply: We replace "the" with "A"

34) P9, L28. ...were observed in some areas of China. 'some areas' is rather vague, can you be more descriptive? ....due to wind speed, topography, underlying ground surface, and climatic conditions (refs). This is a rather very general statement which does not bring any insights. Of course snow depth will vary everywhere due to these factors... if you do analyse in a later section how these factord affect the spatial variability, mention it. 'The relation between these factors and spatial snow depth variability is further investigated in section xxx'...

Reply: We added more description L28: "in some areas of the south of Yangtze River in China". We deleted "due to wind speed, topography, underlying ground surface, and climatic conditions (Gray and Male, 1981; Sturm et al., 1995, 2001; Callaghan et al., 2011b)."

35) P10, L5. The regions with the smallest annual mean snow depth (<5 cm) were located in most areas of the Caucasus Mountains. This is a bit surprising given the high elevations. Is there an elevation bias here? (snow stations are at low elevations?)

Reply: The stations with the smallest annual mean snow depth are located in the eastern and western areas of the Caucasus Mountains, close to the coast of the Caspian Sea and the Black Sea, where the elevations are below 2000 m. We have added the specific location in the text: "were located in the eastern and western areas of the Caucasus Mountains"

36) P10, L13. varied with the latitude

Reply: We deleted "also"

37) P10, L15. . ( 201.8 cm) :here as elsewhere in the text you can should round the snow depth to the nearest centimeter, as this is the probably accuracy of the measurements.

Reply: The maximum value is calculated using the average values of annual maximum snow depth. The average is kept one decimal place, so it is an approximation.

38) P11, L4.... in northern Siberia. Remove extra space between in and northern

Reply: We removed the extra space.

39) P11, L21. ...the increasing rate of snow depth. increasing rates of snow depth

Reply: We replaced "rate" with "rates"

40) P11. L19-22: linear trends and results plotted on Figure 4: were trends computed on annual anomalies or on the wavelet filtered series? You must provide a trend test that accounts for the autocorrelation of time-series (see main comment).

Reply: The trends were computed on the annual anomalies. Any trend analysis is an approximate and simple approach to get what has happened on average during the study period. Linear trend analysis provides an average rate of this change. Despite there is a nonlinearity, the linear trend analysis is also a useful approximation when a systematic low-frequency variations emerged. The linear trend coefficient of snow depth was calculated to represent the rate of change at each station. The Student T test was used to assess the statistical significant of the slope in the linear regression analysis and the partial correlation coefficients, and the confidence level above 95% was considered in our study. Meanwhile, to overcome the strong assumption in ordinary least squares (independent and normal

distribution), we also added the Mann-Kendall (MK) test to identify the monotonic trend in snow depth. Confidence level above 95% was used to determine the statistically significant increase or decrease in snow depth. These two test methods could provide more robust and comprehensive information of the trend analysis. We have added the MK test and compare the results of the two methods:

"The Mann-Kendall statistical curves of annual and maximum snow depth were consistent with the linear trend analysis (Fig. 5). The increasing trend of annual snow depth reached to the 0.05 confident level in the late 1980s and from the early 1990s to the mid-1990s; it reached to the 0.01 confident level in the late 1990s. The decreasing trend reached to the 0.05 confident level from the early 2000s through the mid-2000s. The intersection of the UF curve and UB curve appeared in the mid-1970s, it indicated that the rising trend was an abrupt change during this period. The abrupt change point of the maximum snow depth was in the mid-1980s, then it increased significantly ($p \leqslant 0.05$) from the early 1990s through the mid-1990s, and it reached to the 0.01 confident level from the late 1990s to the early 2010s."

[Figure]

Figure 5. Mann-Kendall statistical curve of annual mean snow depth (a) and maximum snow depth (b) from 1966 through 2012 across the Eurasian continent. Straight line presents significance level at 0.05.

"In order to identify the monotonic trend in monthly snow depth, we conducted the MK test (Fig. 7). In October, snow depth represented a decreasing trend and it reached to the 0.05 confident level only after 2010. The statistically significant changes of monthly snow depth in November during the period of the late 1980s through the early 2000s, though it was not statistically significant with the linear regression. From December through March, there were increasing trends in monthly snow depth and the abrupt change point appeared in the mid-1970s. In the linear regression analysis, the variation of snow depth was not significant in December. However, the results of M-K test showed that the increasing trend of monthly snow depth reached to the 0.01 confident level during the mid-1980s through the late 1990s, and then it decreased during the 2000s. From January to March, monthly snow depth increased significantly ($p \leqslant 0.01$) from the mid-1980s to the early 2010s. In April, the statistically significant increase was found from the late 1990s to the late 2000s, and it reached to the 0.01 confident level after 2000. Consistent with the linear regression, the trend in monthly snow depth was not significant in May."

[Figure]

Figure 7. Mann-Kendall statistical curve of monthly mean snow depth (from October to May) from 1966 through 2012 across the Eurasian continent. (a) October, (b) November, (c) December, (d) January, (e) February, (f) March, (g) April, (h) May. Straight line presents significance level at 0.05.

41) P12, L1. There was a sharp increase of 3.5 cm in the maximum snow depth during the 1970s, then fluctuated from the late 1970s to the early 1990s. : then fluctuations from... Perhaps be more precise: what type of fluctuation?

42) P12, L20. the rate of increase being about 0.6 cm decade

43) P14, L2. in monthly mean snow depth decreased,

44) P14, L3. Changes in monthly mean snow depth were consistent with the trends in winter over the former USSR but more stations with the decreasing trends in the southern Siberia. Do you mean: 'but more stations with decreasing trends were found in southern Siberia'?

45) P14, L5. There were few stations with statistically significant trends of snow depth across China; for these, monthly snow depths tended to decrease at most stations.

46) P14, L11. To explore the spatial variability of snow depth,

47) P14, L15. snow depth to the north of 40° N

48) P14, L23. because a snow depth. remove extra space between 'a' and 'snow'

49) P14. This result indicates that elevation is an important factor affecting snow depth in these regions. I find this statement and the preceding analysis a bit over simplistic. At large scales the snow cover can be tought to depend on latitude, altitude and distance to moisture source (continentality). I feel you are missing the third factor in our analysis. The poor, and generally negative relationship between elevation and snow depth is interesting because it is contrary to what would be expected from orographic effects on precipitation amounts and phase. What you show is that the high elevation of the TP does not cause larger snow depth compared to surrounding lower lands. Continentality seems to be the main driving factor here: the TP is in the rain shadow of the Himalaya and as

such is moisture-deprived. This should be better discussed, and analysed in the paper. This effect could be investigated, perhaps using a simple continentality index (e.g. http://glossary.ametsoc.org/wiki/Continentality). These indices rely on temperature annual ranges. You could use the closest distance to coast as another simple index.

Reply: We indeed conducted analysis on impact of continentality on snow depth over the study area. To our surprise, the correlation between the two are not significant. However, we strongly agree with the reviewer that at local and regional (less than continental) scales, topography can play key roles. In Section 4.2, we added more discussions about topographical effects on snow depth, especially on Tibetan Plateau (TP). The Tibetan Plateau's largest snow accumulation occurred in the winter, but the snowfall during winter months is the smallest of the year. This was mainly due to the terrain factors: the water vapor from the east and west was blocked by the Hengduan Mountains and Nyainqentanglha Mountains, respectively, which resulted in less snowfall. Although there was more snowfall in spring, snow cover was not easy to accumulate with higher temperatures. Therefore, snow depth was shallow on TP in general. In addition to topographic factors, spatial distribution of snow depth was also affected by atmospheric circulation. We will discuss this issue in the future studies.

50) P15: section 3.4.
You begin the section by stating that ' Variations in snow depth are closely related to climate change'. But what is investigated is the influence of spatially variable climate factors (mean temperature and mean snowfall) on snow depth, and NOT the effect of time-varying climate on snow depth. To do so you would have to test the influence of changing temperatures and snowfall/precipitation on snow depth over time. Rephrase the introduction of the section to clearly explain that you investigate spatial relationships between mean temperature and snowfall on mean snow depths. The spatial relationship between air temperature and snow depth will be undoubtedly complex when considered an area as big and topographically diverse as the Eurasian continent. Your analysis is till interesting as it shows that snowfall is the main factor driving spatial variability in snow depth, at this spatial scale. However snowfall rates and air temperature must also be somewhat correlated, as snowfall depends on precipitation and temperature (precipitation phase). I suggest that you also calculate and report the partial correlation coefficients, i.e. to show the influence one variable while removing effect of the other, on snow depth. You do examine the effect of changing climate on snow depth and mass in Figure 10 for the composite Eurasian and Russian records. This analysis is qualitative and hile interesting and valuable, it could be enriched by calculating and presenting the correlation coefficients between series. Especially for the SWE series, how much of the variance can be respectively explained by air temperature and snowfall? Even more constructive would be to perform this analysis on a station basis and map the results. We would then learn about the spatially variable climate control on snow.

Reply: Thank you very much for your suggestion. We have added the quantitative analysis by the partial correlation analysis between snow depth, SWE, air temperature and snowfall. The results showed that the significant negative correlation ($p \leqslant 0.05$) between snow depth and air

temperature presented in most areas of European Russia and the southern Siberia (Fig 13a). The stations with negative effects of air temperature on SWE were less, and there were no statistically significant correlation in the northern Siberia (Fig 13b). It was because the air temperature was below 0℃ in most areas of Siberia during December through March, the increasing temperature did not have an obvious effect on snow depth. Consistent with the interannual variation, changes in snow depth and SWE were more affected by snowfall in most areas across the former USSR from December through March. The greater partial correlation coefficients (>0.6) between snow cover and snowfall appeared in the northern European Russia, the southern Siberia, the northeast and southeast of the Russian Far East. Variations in snow depth and SWE were more sensitive to snowfall and snowfall rate in these areas. Variations of snow depth were explained by air temperature and snowfall in most areas of the European Russia and some regions of the southern Siberia, the effects of the two factors on SWE only appeared in some of these areas; however, snowfall was the main driver force of the variance of snow depth and SWE in the former USSR.

[Figure]

**Figure 13.** Spatial distributions of partial correlation coefficients of snow depth and air temperature (a), snow depth and snowfall (b), SWE and air temperature (c), SWE and snowfall from November through March during 1966-2009. The coefficients reaching to 0.05 confident level are displayed. Red circles represent a negative relationship, and blue circles indicate a positive relationship.

51) P15, L7. Snow depth significantly decreases with increasing air temperature (P≤0.05),

Reply: We replaced "decreased" with "decreases"

52) P15, L17. Increases

Reply: We replaced "increases" with "increasing trends"

53) P15, L21. The significant increasing snowfall can explain the sudden drop in snow density observed from the mid-1990s through the early 2000s (Zhong et al., 2014): fresh snow with low snow density. Explain the last statement better in a separate statement. Why does increasing snowfall decreases snow density? Is it the mean density of the snowpack? Increasing snowfall in response to warmer temperature should increase the density, both of fresh snow, and perhaps of the whole snowpack due to faster metamorphism and increased compaction...

Reply: The snow density means the bulk snow density of the snow profile. Increasing snow fall should decrease the density of the surface snowpack, which lowed the whole density of snowpack.

54) P15, L26. increasing trend of changes in snow depth. trend in snow depth? or trend in the rate of change?

Reply: It means the trend in snow depth. We deleted "of changes".

55) P15, L27. In fact, the climatology of snow depth not only influenced by air temperature and precipitation, but also with other climatic factors and atmospheric circulation. Poor formulation, rephrase.

Reply: We have deleted the sentences and added the analysis of the partial correlation coefficients between snow cover and air temperature, as well as snow cover and snowfall.

"The partial correlation coefficients between snow cover and air temperature, as well as snow cover and snowfall were calculated to discuss the spatial relationship between them (Fig. 13). The significant negative correlation (p≤0.05) between snow depth and air temperature presented in most areas of European Russia and the southern Siberia (Fig 13a). The stations with negative effects of air temperature on SWE were less, and there were no statistically significant correlation in the northern Siberia (Fig 13b). It was because the air temperature was below 0℃ in most areas of Siberia during December through March, the increasing temperature did not have an obvious effect on snow depth.
Consistent with the interannual variation, changes in snow depth and SWE were more affected by snowfall in most areas across the former USSR from December through March. The greater partial correlation coefficients (>0.6) between snow cover and snowfall appeared in the northern European Russia, the southern Siberia, the northeast and southeast of the Russian Far East. Variations in snow depth and SWE were more sensitive to snowfall and snowfall rate in these areas."

56) P16. L7. These discrepancies may result from differences in the time frame

Reply: We replaced the first "differences" with "discrepancies".

57) P16. L26. during the different study periods.

Reply: We revised it.

58) P16. L.26-28. The sensitivity of snow cover to air temperature and precipitation for each station showed regional differences (Fallot et al., 1997; Park et al., 2013). The amount of snowfall can be affected by climate change, and leading to differences in snow depth at different times (Ye et al., 1998; Kitaev et al., 2005). This is why simple spatial relationship between air temperature and snow depth do not exist...

Reply: We added a sentence after it.

"The results of our study showed that there was significant negative relationship between snow depth and air temperature in the southern Siberia, however, it did not exist in the northern Siberia. This may explain the difference in the results of these studies."

59) P17, L5. and is the main reason. Extra space between 'and' and 'is'

Reply: We deleted the extra space.

60) P17, L10. Therefore, we will select a typical climate zone to research the climatology and variations of snow cover. This rather vague... your study looks at large scale control on snow cover and this is what this dataset allows. Studying small scale (topography, vegetation) effects on the snow cover requires other kind of data, sampled at a higher spatial resolution. I would remove this sentence.

Reply: Thank you very much for your suggestion. We removed the sentence.

61) You should better discuss your results in the light of what is known about large-scale control on snow cover: latitude, altitude and continentality are the main geographical factor which drive snowfall rates and hence snow depths. I find your analysis and the discussion on page 17 somewhat incomplete in this respect.

Reply: We have analyzed the relationship between snow depth and continentality. But the correlation coefficient was not high (r=0.1). This indicated that the continentality is not an important driving factor of snow cover climatology over Eurasia. However, we strongly agree with the reviewer that at local and regional (less than continental) scales, topography can play key roles. In Section 4.2, we added more discussions about topographical effects on snow depth, especially on Tibetan Plateau (TP):

"Some important questions that are not addressed in the current research should be resolved in the future. Topography is an important factor affecting the climatology of snow depth, and

is the main reason causing the inhomogeneity of data. Previous studies have analyzed the representation of snow depth for single stations to solve the issue (Grünewald and Lehning, 2011, 2013; Grünewald et al., 2014). However, in the present study, we did not discuss this question because of the complexity of spatial difference. But we still got some interesting conclusions: There was a closely relationship between snow depth and elevation at the local scale. However, compared with latitude, the correlation between them was not so significant in the whole Eurasian Continent. Moreover, the continentality did not play a great role in spatial distribution of snow depth, especially on TP. The previous studies showed that the Tibetan Plateau's largest snow accumulation occurred in the winter, but the snowfall during winter months is the smallest of the year (Ma, 2008). This was mainly due to majority of annual precipitation occurs during the summer monsoon season on TP which cause very less snowfall during winter half year (or snow accumulated season). Furthermore, the water vapor from the east and west was blocked by the Hengduan Mountains and Nyainqentanglha Mountains, respectively, which resulted in less snowfall. Although there was more snowfall in spring, snow cover was not easy to accumulate with higher temperatures. Therefore, snow depth was shallow on TP in general. In addition to topographic factors, spatial distribution of snow depth was also affected by atmospheric circulation. We will discuss this issue in the future studies."

**List of all relevant changes**

(1) P1, L.4-5: advanced the sixth co-author to the third, interchanged "2" and "3" in superscript.

(2) P1, L.8-11: interchanged the second and third affiliations.

(3) P1, L21: inserted "a" before "snow", inserted "its" before "spatiotemporal".

(4) P1, L27: inserted "significantly" after "depth", deleted "dramatically".

(5) P2, L2: replaced "cover" with "depth".

(6) P3, L7: replaced "also" with "all".

(7) P3, L14: inserted "the" before "northern", inserted "during 1936 to 2010" after "Eurasia".

(8) P3, L15: replaced "is" with "was".

(9) P3, L19: replaced "promotes" with "promoted".

(10) P3, L21-P4, L13: deleted the two paragraphs.

(11) P4, L17: inserted "during 1946 to 2000" after "America".

(12) P4, L18: inserted "during the recent 70 years" after "Arctic".

(13) P4, L23: inserted "from 1936 through 1995" after "winter".

(14) P4, L28: inserted "also" after "is", inserted "large-scale atmospheric circulation indices," after "other", deleted "climatic variables".

(15) P4, L29: replaced "index" with "indices", inserted "For example," before "Beniston".

(16) P5, L3: inserted "during the period from 1966 to 1990".

(17) P5, L8: replaced "In order to obtain a wider rang" with "To increase the spatial coverage".

(18) P5, L18: replaced "is" with "remains".

(19) P6, L.4-6: deleted "Using data from ground based measurements,", replaced "the" with "The", deleted "provide a detailed description of snow depth and to", replaced "as well as its" with "and analyze snow depth".

(20) P6, L7: replaced "other" with "the".

(21) P6, L14: deleted "in 17 countries", inserted "over" before "on".

(22) P6, L15: inserted "a" after "on", inserted "Historical" before "snow".

(23) P6, L16: inserted "from 1966 to 2011" after "USSR", deleted "from historical records from 1966 to 2011"

(24) P6, L17: replaced "that run" with "performed".

(25) P6, L18: replaced "10" with "ten", replaced "5" with "five".

(26) P6, L19: replaced "for" with "over", inserted "-long transects" after "km".

(27) P6, L20: replaced both "each" with "every".

(28) P6, L.25-26: inserted "where SWE data are available" after "USSR", replaced both "at" with "along".

(29) P6, L.27-29: replaced the sentence by "Daily precipitation was divided partitioned into a solid and liquid fraction, based on daily mean temperature (Brown, 2000)".

(30) P7, L. 9-12: replaced the sentences by "(1) A threshold of 15 days was selected because the snow cover duration in some areas of China was less than one month, and the data for 15 days' snow depth in a month were relatively stable. Months having less than 15 days with snow depth data were omitted from the analysis. (2) Stations with less than 20 years of data during the 1971-2000 period were excluded from the analysis."

(31) P7, L12: deleted "and", inserted "At each station," before "data".

(32) P7, L16: inserted "The snow cover extent is the smallest in July and August, in order to capture the entire seasonal snow cycle," before "we".

(33) P7, L17: replaced "had" with "have".

(34) P7, L18: replaced "in the past" with "over the course of the studies period".

(35) P7, L25: inserted "(Ma and Qin, 2012)" after "products".

(36) P7, L.25-27: deleted the sentence "A threshold of 15 … were relatively stable", and added a sentence "According to the quality control, months having more than 15 days with snow data were used."

(37) P8, L2: replaced "In order to reflect" with "To capture".

(38) P8, L.13-16: deleted the sentence "(4) Linear trend…at the 95% level."

(39) P8, L23: deleted the last "the", inserted "obtain mean" after "stations to".

(40) P8, L.24-25: deleted the last sentence.

(41) P8, L.26: replaced "analyze" with "reveal", inserted "low-frequency" before "variations".

(42) P8, L.27: inserted "over the study area as a whole" after "depth".

(43) P8, L26 - P9, L20: replaced the paragraph with "Wavelet analysis was performed to reveal the long-term low-frequency variations of snow depth over the study area as a whole. A wavelet is a wave-like oscillation with an amplitude that begins at 0, increases, and then decreases back to 0 (Graps, 1995). We applied a discrete wavelet transform, excluded the high-frequency components and then used the inverse transform to reconstruct the lower frequency signal. Any trend analysis is an approximate and simple approach to obtain what has happened on average during the study period. Linear trend analysis provides an average rate of this change. Despite there is a nonlinearity, the linear trend analysis is also a useful approximation when a systematic low-frequency variations emerged. (Folland and Karl, 2001; Groisman et al., 2006). The linear trend coefficient of snow depth was calculated to represent the rate of change at each station. The Student T test was used to assess the statistical significant of the slope in the linear regression analysis and the partial correlation coefficients, and the confidence level above 95% was considered in our study. Meanwhile, to overcome the strong assumption in ordinary least squares (independent and normal distribution), we applied a Mann-Kendall (MK) test to identify the monotonic trend in snow depth. Confidence level above 95% was used to determine the statistically significant increase or decrease in snow depth. These two test methods could provide more robust and comprehensive information of the trend analysis. In order to evaluate the influence of single climatic factor on snow cover, the partial correlation coefficients were calculated and reported the relationships between snow depth, SWE, air temperature and snowfall. The way to do significant test of the correlation coefficient is same to the trend analysis, which includes T-test and MK-test."

(44) P9, L26: replaced "The" with "A".

(45) P9, L28: inserted "the south of Yangtze River in" after "of".

(46)P9, L29 – P10, L1: deleted "due to wind speed…Callaghan et al.,2011b)"

(47)P10, L5: replaced "most" with "the eastern and western".

(48)P10, L13: deleted "also".

(49)P11, L26: inserted "it" after "then".

(50)P11, L.26-28: deleted "late 1970s…early 1990s through the".

(51)P11, L29: inserted "and" before "then", inserted "(Fig. 4a)" after "2012"

(52)P12, L2: replaced "fluctuated" with "it fluctuating changed".

(53)P12, L4: inserted a new paragraph "The Mann-Kendall statistical curves of annual and maximum snow depth were consistent with the linear trend analysis (Fig. 5). The increasing trend of annual snow depth reached to the 0.05 confident level in the late 1980s and from the early 1990s to the mid-1990s; it reached to the 0.01 confident level in the late 1990s. The decreasing trend reached to the 0.05 confident level from the early 2000s through the mid-2000s. The intersection of the UF curve and UB curve appeared in the mid-1970s, it indicated that the rising trend was an abrupt change during this period. The abrupt change point of the maximum snow depth was in the mid-1980s, then it increased significantly (p≤0.05) from the early 1990s through the mid-1990s, and it reached to the 0.01 confident level from the late 1990s to the early 2010s."    before the second paragraph.

(54)P12, L5: replaces "Fig. 5" with "Fig. 6".

(55)P12, L7: replaces "Figs. 5a-b" with "Fig. 6a-b", replaced "decrease" with "decreasing".

(56)P12, L8: replaced "Fig. 5a" with "Fig. 6a".

(57)P12, L11: replaced "Fig. 5c-e" with "Fig. 6c-e".

(58)P12, L15: replaced "Fig. 5d, e" with "Fig. 6d, e".

(59)P12, L20: replaced "was" with "being".

(60)P12, L21: replaced "Fig. 5f-g" with "Fig. 6f-g".

(61)P12, L23: replaced "Fig. 5f" with "Fig. 6f".

(62)P12, L25: inserted "increasing" after "fluctuant", replaced "Fig. 5g" with "Fig. 6g".

(63)P12, L26: replaced "increased dramatically" with "sharply increased".

(64)P12, L28: inserted a new paragraph "In order to identify the monotonic trend in

monthly snow depth, we conducted the MK test (Fig. 7). In October, snow depth represented a decreasing trend and it reached to the 0.05 confident level only after 2010. The statistically significant changes of monthly snow depth in November during the period of the late 1980s through the early 2000s, though it was not statistically significant with the linear regression. From December through March, there were increasing trends in monthly snow depth and the abrupt change point appeared in the mid-1970s. In the linear regression analysis, the variation of snow depth was not significant in December. However, the results of M-K test showed that the increasing trend of monthly snow depth reached to the 0.01 confident level during the mid-1980s through the late 1990s, and then it decreased during the 2000s. From January to March, monthly snow depth increased significantly ($p \leqslant 0.01$) from the mid-1980s to the early 2010s. In April, the statistically significant increase was found from the late 1990s to the late 2000s, and it reached to the 0.01 confident level after 2000. Consistent with the linear regression, the trend in monthly snow depth was not significant in May." before the last paragraph.

(65) P12, L28: replaced "Fig. 6" with "Fig. 8".

(66) P13, L3: replaced "Fig. 6a" with "Fig. 8a".

(67) P13, L7: deleted "variability".

(68) P13, L12: replaced "Fig. 6b" with "Fig. 8b".

(69) P13, L18: replaced "Figs. 7a, b" with "Figs. 9a, b".

(70) P13, L26: replaced "7c-e" with "9c-e".

(71) P14, L2: replaced "fell" with "decreased".

(72) P14, L3: replaced "Figs. 7f-h" with "Figs. 9f-h".

(73) P14, L4: deleted "the" after "with".

(74) P14, L5: inserted "were found" after "trends", deleted "the".

(75) P14, L6: replaced "and" with "; for these,"

(76) P14, L10: replaced "and" with a comma, inserted "and Continentality" after "Elevation".

(77) P14, L11: replaced "features" with "variability"

(78) P14, L12: replaced "and elevation (Fig. 8)" with ", elevation and continentality

(Fig. 10)".

(79) P14, L14: replaced "Fig. 8a" with "Fig. 10a".

(80) P14, L15: deleted "mean".

(81) P14, L16: replaced "8a, c" with "10a, d".

(82) P14, L21: replaced "Fig. 8b" with "Fig. 10b".

(83) P14, L28: replaced "Fig. 8c" with "Fig. 10d".

(84) P14, L30: inserted a new paragraph "There was a significant positive relationship between snow depth and continentality, but the correlation coefficient was not high (r=0.1, Fig. 10c). This indicated that the continentality is not an important driving factor of snow cover climatology over Eurasia, though it will determine the snowfall rate."

(85) P15, L2: replaced "Variations" with "In addition to the terrain factors, variations", replaced "change" with "variability"

(86) P15, L5: replaced "Fig. 9" with "Fig. 11".

(87) P15, L7: replaced "decreased" with "decreases".

(88) P15, L9: replaced "9a" with "11a".

(89) P15, L10: replaced "Fig. 9b" with "Fig. 11b".

(90) P15, L17: replaced "increases" with "increasing trends".

(91) P15, L18: replaced "10" with "12", replaced "increase" with "increased".

(92) P15, L22: inserted "the bulk" after "in".

(93) P15, L23: replaced "fresh snow with low snow density" with "increasing snowfall should decrease the density of the surface snowpack, which lowed the whole density of snowpack".

(94) P15, L25: replaced "Figs. 10b-d" with "Figs. 12b-d".

(95) P15, L26: deleted "of changes".

(96) P15, L.27-30: deleted the two sentences "In fact,…in the future."

(97) P15, L30: inserted two new paragraphs "The partial correlation coefficients between snow cover and air temperature, as well as snow cover and snowfall were calculated to discuss the spatial relationship between them (Fig. 13). The significant

negative correlation (p≤0.05) between snow depth and air temperature presented in most areas of European Russia and the southern Siberia (Fig 13a). The stations with negative effects of air temperature on SWE were fewer, and there were no statistically significant correlation in the northern Siberia (Fig 13b). It was because the air temperature was below 0℃ in most areas of Siberia during December through March, the increasing temperature did not have an obvious effect on snow depth.

Consistent with the interannual variation, changes in snow depth and SWE were more affected by snowfall in most areas across the former USSR from December through March. The greater partial correlation coefficients (>0.6) between snow cover and snowfall appeared in the northern European Russia, the southern Siberia, the northeast and southeast of the Russian Far East. Variations in snow depth and SWE were more sensitive to snowfall and snowfall rate in these areas."

(98) P16, L7: replaced "differences" with "discrepancies".

(99) P16, L26: inserted "the" after "during", inserted "study" after "different".

(100)    P16, L30: inserted two new sentences "The results of our study showed that there was significant negative relationship between snow depth and air temperature in the southern Siberia, however, it did not exist in the northern Siberia. This may explain the difference in the results of these studies." after "(Ye et al., 1998; Kitaev et al., 2005)."

(101)    P17, L.9-14: deleted the sentences "This issue should…field measurements is required."

(102)    P17, L14: inserted new sentences "But we still got some interesting conclusions: There was a closely relationship between snow depth and elevation at the local scale. However, compared with latitude, the correlation between them was not so significant in the whole Eurasian Continent. Moreover, the continentality did not play a great role in spatial distribution of snow depth, especially on TP. The previous studies showed that the Tibetan Plateau's largest snow accumulation occurred in the winter, but the snowfall is the smallest of the year (Ma, 2008). This

was mainly due to majority of annual precipitation occurs during the summer monsoon season on TP which cause very less snowfall during winter half year (or snow accumulated season). Furthermore, the water vapor from the east and west was blocked by the Hengduan Mountains and Nyainqentanglha Mountains, respectively, which resulted in less snowfall. Although there was more snowfall in spring, snow cover was not easy to accumulate with higher temperatures. Therefore, snow depth was shallow on TP in general. In addition to topographic factors, spatial distribution of snow depth was also affected by atmospheric circulation. We will discuss this issue in the future studies."

(103)    P18, L1: inserted ", especially during the period of the mid-1980s through the 2000s" after "spring".

(104)    P18, L.9-12: deleted the sentences "The variations in snow depth…in future stuies."

(105)    P18, L12: inserted new sentences "Variations of snow depth were explained by air temperature and snowfall in most areas of the European Russia and some regions of the southern Siberia, the effects of the two factors on SWE only appeared in some of these areas; however, snowfall was the main driver force of the variance of snow depth and SWE in the former USSR."

(106)    P19, L.2-3: deleted the reference.

(107)    P19, L.27-30: deleted the reference.

(108)    P20, L23: inserted a new reference "Folland, C.K. and Karl, T.R.: Observed climate variability and change. Climate Change 2001: The Scientific Basis, edited by: J. T. Houghton et al., Cambridge University Press, Cambridge, UK, 99–181, 2001."

(109)    P20, L.26-27: deleted the reference.

(110)    P20, L30 - P21, L1: deleted the reference.

(111)    P21, L8: inserted a new reference "Groisman, P., Knight, R., Razuvaev, V., Bulygina, O., and Karl, T.: State of the Ground: Climatology and Changes during the Past 69 Years over Northern Eurasia for a Rarely Used Measure of Snow Cover and Frozen Land, J. Climate, 19, 4933–4955, 2006."

(112)    P21, L30 - P22, L3: deleted the reference.

(113)    P22, L.10-11: deleted the reference.

(114)    P22, L.14-19: deleted the two references.

(115)    P22, L20: replaced "Listion" with "Liston".

(116)    P22, L23: inserted a new reference "Ma, L.: Spatiotemporal variation in snow cover and the relationship between snow cover and atmospheric circulation in the Tibetan Plateau in recent 50 years, PhD dissertation, Graduate University of Chinese Academy of Sciences, 156pp, 2008."

(117)    P23, L.17-18: deleted the reference.

(118)    P24, L.5-8: deleted the two references.

(119)    P26: replaced figure 1 with a new figure

(120)    P30: replaced figure 4 with a new figure

(121) P30, L.5-7: deleted the last sentence "Y represents…annual mean snow depth".

(122) P31: added a new figure as figure 5

[Figure]

Figure 5. Mann-Kendall statistical curve of annual mean snow depth (a) and maximum snow depth (b) from 1966

through 2012 across the Eurasian continent. Straight line presents significance level at 0.05.

(123)    P31, L1: replaced "Figure 5" with "Figure 6".

(124)    P31: replaced figure 6 with a new figure

[Figure]

(125)   P32: added a new figure as figure 7

[Figure]

Figure 7. Mann-Kendall statistical curve of monthly mean snow depth (from October to May) from 1966 through 2012 across the Eurasian continent. (a) October, (b) November, (c) December, (d) January, (e) February, (f) March, (g) April, (h) May. Straight line presents significance level at 0.05.

(126)    P32, L2: replaced "Figure 6" with "Figure 8".

(127)    P34, L1: replaced "Figure 7" with "Figure 9".

(128)    P34, L8: replaced "Figure 8" with "Figure 10".

(129)    P34, L.8-10: replace the explanation of Figure 10 with "Figure 10. The relationship between annual mean snow depth and latitude (a), elevation (b) and continentality (c) for all stations across the Eurasian continent during 1966-2012. Asterisks show the mean snow depth of each station; the thick line is a linear regression trend; the different colors represent snow depth (cm) of each station (d)."

(130)    P34: replaced figure 10 with a new figure

(131)    P35, L2: replaced "Figure 9" with "Figure 11".

(132)    P35: replaced figure 11 with a new figure

(133)    P36, L2: replaced "Figure 10" with "Figure 12".

(134)    P36: replaced figure 12 with a new figure

(135)    P36: added a new figure as Figure 13

[revised manuscript text omitted]

---

## Referee Report (RR2)

Re-review of "Spatiotemporal variability of snow depth across the Eurasian continent from 1966 to 2012"

By X. Zhong, T. Zhang, L. Zheng, Y. Hu, H. Wang, and S. Kang

Submitted in revised form to *The Cryosphere*

Reference: tc-2016-182

Summary: In this paper, the authors develop a snow depth climatology across the Eurasian continent using ground-based observations over 1966-2012. A total of 1814 stations from 17 countries spanning Eurasia with snow data are used to assess mean annual and maximum snow depth and their trends for each site. The northern reaches of Eurasia typically have the greatest mean annual snow depth, revealing a latitudinal dependence on the results. Trends assessed from linear regressions show significant increases in snow depth poleward of 50°N. These trends are associated more so with increased snowfall rather than rising air temperatures.

This revised paper addresses some of the issues raised in an initial review of the article. However, there are outstanding issues that remain to be addressed before the paper can be considered for publication in *The Cryosphere*. The language requires considerable improvement throughout the paper and my specific comments highlight only a small fraction of these issues. The paper provides only a cursory discussion of the results and the authors appear to be missing an opportunity to place these results in a larger context. For instance, are the observational results consistent with simulations of snow parameters across Eurasia? Furthermore, the authors provide at times incomplete or unsatisfactory responses to comments provided (by both referees) during the initial review of their paper. My report therefore provides further comments the authors need to address in revising their manuscript.

General Comments:

1) In my initial review, I commented: "This study examines the characteristics and trends across the Eurasian continent from 1966 to 2012. To do so, the authors assemble snow depth data from 1103 stations across the study area. How representative are the station (point) snow depth data of the overall regional landscapes of interest? For instance, are snow depth data in forested areas collected at airports or other open areas, that may not represent the regional snow characteristics?" The authors acknowledge the shortcomings of the station distribution used in their study but do not address the point in question. Are the results based on point observations representative of the vast region under study?

2) The authors provide comprehensive information on snow data collection in the former USSR, but fail to report similar information for other countries. How is snow depth measured across Eurasia? Has sampling changed to automated sensors (e.g. sonic rangers) in recent decades? Little information is provided on the data collection process and the accuracy of the measurements. Further to this,

how is homogeneity in the time series of snow depth, SWE, and other variables assured if sampling techniques or instruments have changed over time? Have the time series been tested for homogeneity (i.e. discontinuities in the data)? Finally, no information is provided on how air temperature and precipitation measurements were made at the meteorological stations. Snowfall measurements are notoriously difficult to make and gauge undercatch correction factors must be applied to obtain improved estimates of snowfall, particularly in windy environments such as Arctic and alpine tundra. The entire section describing the observational data used in the present study must be improved and expanded. Such details may be provided in a supplementary document as necessary.

3) In response to another comment I made (as well as by Referee #2), the authors now employ the Mann-Kendall test to assess linear trends in addition to linear regressions. However, they fail to address the issue of serial correlation impacts on the trend analyses (as raised by Referee #2). This must be addressed before the paper can be considered for publication.

4) Further to this, the (revised) Figures 5 and 7 are confusing – what results do these figures represent? The Mann-Kendall trend analysis should give you one slope value over a period of study. No details are provided in the Data/Methods section on how the results presented in these figures are obtained. Further to this, what are "UF" and "UB" in these figures?

5) In my initial review, I commented: "Do the linear trends reported in Section 3.2 exceed the variability in the snow depth data? In other words, are there "detectable" trends in snow depth, i.e. with the signal greater than the noise in the system?" The authors' response does not fully address this issue, i.e. whether the slopes of the linear trends (signal) exceed the standard deviation (noise) in the snow parameters of interest.

6) The Discussion remains relatively brief and could be augmented by placing these results in a larger context. Do these results concord with modeling studies of snow across Eurasia? What are the prospects for future snow cover changes in Eurasia? What are the broader implications of the results to regional hydrology, permafrost distribution, ecology and society?

7) The names of countries or their abbreviations can be removed on all figures after Figure 1.

8) Please improve the language throughout the paper – there are portions of the text that are difficult to comprehend due to language issues, including all of Section 4.2. Furthermore, the verb tense in the introduction changes constantly and only one tense should be used consistently.

Specific Comments:

1) P. 3, line 13: Replace "reduced" with "declined".
2) P. 3, lines 15-18: The grammar in this sentence is poor – please rephrase.
3) P. 3, line 27: Replace "was" with "were".
4) P. 4, line 20: What aspect of "passive microwave" improved the algorithms?
5) P. 4, line 25-27: Language needs much improvement here.
6) P. 4, line 29: Delete the hyphen after "longer". Insert "the" before "climatology".

7) P. 5, line 26: Do you mean "and during the snowmelt period (every five days)"?
8) P. 6, line 7: Delete "the following Equation (1)"
9) P. 6, lines 25-27: Rephrase this sentence.
10) P. 6, line 28: Change to "study period".
11) P. 7, line 14: Replace the colon after "2012" with a period.
12) P. 7, lines 23-27: These sentences need to be rephrased.
13) P. 8, line 8: What do you mean with "Despite there is a nonlinearity".
14) P. 8, line 9: Delete "a" before "systematic".
15) P. 8, line 19: Delete "In order".
16) P. 8, line 20: Insert "a" before "single".
17) P. 8, lines 27-29: Rephrase this sentence. Insert a space after "(Fig. 2)".
18) P. 11, line 2: What do you mean by "fluctuating changed"?
19) P. 11, line 6 and elsewhere: Replace "confident level" with "confidence level".
20) P. 12, line 5: What do you mean by "fluctuant increasing trend"?
21) P. 12, line 10 and elsewhere: Replace "confident level" with "confidence level".
22) P. 12, line 30 and elsewhere: Delete spaces between the degree sign and North, i.e. "40°N".
23) P. 13, line 5: Replace "Eurasian areas" with "Eurasia".
24) P. 14, lines 12-13: Language must be improved here.
25) P. 14, line 25: Insert the p-value for the correlation coefficient.
26) P. 15, line 9: Change to "at most".
27) P. 15, line 22: Replace "lowed" with "lowered".
28) P. 16, line 2: Delete "the" before "northern".
29) P. 16, lines 3-5: This sentence must be re-written.
30) P. 17, line 12: Delete "the" before "southern".
31) P. 17, line 16: This entire section is poorly phrased and needs to be completely revised. Why does the font size change in the middle of the paragraph?
32) P. 18, line 18: Replace "increase" with "increasing".
33) P. 18, line 25: Delete "the" before "southern".
34) P. 20, line 16: Note spelling mistake in "Atmos."
35) P. 20, lines 27-28: Why are editors of a special journal issue listed here?
36) P. 21, line 18: Is this "Hydrol. Sci. J."?
37) P. 21, lines 22-23: Why are upper case letters provided for each word in the title of this article?
38) P. 22, line 19: Insert a hyphen in "Snow atmosphere".
39) P. 25, Table 1: Change to "snow courses".
40) P. 26, Figure 1: Why does the orientation of the triangles change across the figure? The top of the triangle should point directly northward to provide a consistent pattern across the figure.
41) P. 27, Figure 2 and subsequent figures: Delete all country names/abbreviations on the maps providing spatial results as this can be found on Figure 1.
42) P. 30, Figure 4: It is unclear why the authors use wavelets to extract low frequency in the time series of snow depth anomalies. Why not just use a running mean of the data?
43) P. 31, Figure 5: The results presented in this figure and in Figure 7 are difficult to interpret as details on what is being shown are not provided. Linear trends

inferred from the Mann-Kendall test should yield only one slope value for a period of record, so it is unclear what the time series in Figures 5 and 7 denote. What do the two lines "UF" and "UB" represent, the figure caption does not state what these are.

---

## Referee Report (RR3)

Re-review of "Spatiotemporal variability of snow depth across the Eurasian continent from 1966 to 2012"

By X. Zhong, T. Zhang, S. Kang, K. Wang, L. Zheng, Y. Hu and H. Wang

Submitted in revised form to *The Cryosphere*

Reference: tc-2016-182

Summary: In this paper, the authors develop a snow depth climatology across the Eurasian continent using ground-based observations over 1966-2012. A total of 1814 stations from 17 countries spanning Eurasia with snow data are used to assess mean annual and maximum snow depth and their trends for each site. The northern reaches of Eurasia typically have the greatest mean annual snow depth, revealing a latitudinal dependence on the results. Trends assessed from linear regressions show significant increases in snow depth poleward of 50°N. These trends are associated more so with increased snowfall rather than rising air temperatures.

This revised paper addresses in a satisfactory manner the issues raised in my previous reviews of the article. However, there are some minor technical issues that remain to be resolved as outlined in my report. I also note that the list of co-authors and their order has changed yet again, and so the authors must explain this change of co-authorship on their paper.

General Comments:

1) P. 4, second paragraph: Apart from remote sensing, numerical modeling is often used to obtain accurate and spatially-complete fields of snow depth and/ or snow water equivalent (SWE) (e.g., Liston and Hiemstra, 2011). Is there any reason why model simulations of snow depth and SWE are not mentioned in this paragraph, as they form another important source of cryospheric information in data sparse regions such as northern Eurasia?

2) P. 20, Appendix A: I appreciate the authors' consideration of the potential effects of serial correlation on their trend analyses. However, rather than the elaborate Durbin-Watson test, did the authors look simply at the lag 1 auto-regression (AR1) to examine if serial correlation was indeed present in their time series? How would those results compare to those obtained from the Durbin-Watson test?

Specific Comments:

1) P. 1, line 25: Replace "are huge" with "remain large" and replace "evening" with "even".
2) P. 2, line 6: Change the verb tense to the present, i.e. "provides".
3) P. 3, line 12: Insert "the" before "surface".
4) P. 4, line 8: Insert "the" before "NAO".

5) P. 4, line 9: Revise to: "fluctuations of snowfall amounts and snow depth".
6) P. 4, line 12: Change to: "however, the NAO index was…"
7) P. 4, line 25: Clouds do not interfere with microwave remote sensing of SWE, so this statement is misleading.
8) P. 4, line 26: What are "perfect algorithms"? Is there such a thing?
9) P. 5, line 18: Change to: "to develop a climatology and investigate the variability"
10) P. 6, line 16: Why the tentative language in this sentence? The air temperature measurements either have or do not have accuracy of 0.1°C. If not, then specify the exact accuracy of those measurements.
11) P. 6, line 26, Equation (1): Do not italicize the units of °C.
12) P. 8, line 6: Insert a space in "than 20".
13) P. 8, line 10: Replace semi-colon by a period at the end of the sentence.
14) P. 10, lines 6/7: The statement starting with "were mostly…" is incomplete – please rephrase.
15) P. 11, line 16: Replace "increased" with "increasing".
16) P. 12, lines 1-7: There's much repetition of ideas and text in this paragraph – please review and edit carefully.
17) P. 13, line 23: Delete "the" before "previous".
18) P. 13, line 24: Elsewhere, the Tibetan Plateau is abbreviated as "TP" but not here.
19) P. 13, line 24: Delete "the" before "winter" and "the" before "precipitation".
20) P. 13, line 25: Delete "the" before "winter".
21) P. 13, line 28: Replace "accumulated" with "accumulation".
22) P. 15, line 6: Insert a comma after "0°C,".
23) P. 15, line 23: Insert "a" before "different".
24) P. 16, line 5: "station" should be singular. Insert "a" before "critical".
25) P. 16, lines 11-14: The journal may prefer superscripts for all units, i.e. "cm yr$^{-1}$".
26) P. 16, line 13: Replace the comma after "Russia" with a semi-colon.
27) P. 17, line 4: Insert "Phase 5" after "Project".
28) P. 17, line 5: Here and elsewhere, the long name for the Tibetan Plateau is used again.
29) P. 17, lines 5-6: Delete "the" before "forested regions".
30) P. 17, line 24: The sentence starting with "Spring floods" is incomplete – please rephrase.
31) P. 17, line 29: Change to "forests".
32) P. 17, line 30: Change to "plant".
33) P. 20, line 22: Replace "anomalies" with "anomaly".
34) P. 20, line 24: Change to "its estimate value r:"
35) P. 21, line 7: Insert "at" before "approximately".
36) P. 21, line 14: See previous comment about the format for units.
37) P. 25, line 28: Note spelling mistake in "surface".
38) P. 26, lines 20-22: Please update with the appropriate volume and page numbers.
39) P. 29, line 6, Table A2: Insert "the" before "Dikson".
40) P. 31, lines 3-5, Figure 1: Replace "triangles" with "circles". The figure caption should explain all abbreviations used for the names of countries on the map. What does the inset map on the bottom right show? Is this inset map shown here and on Figures 2, 3, 6, and 7 needed, as no results are shown on these?

41) P. 36, Figure 5: The caption needs to explain why linear regressions are shown only on a few panels for this plot.
42) P. 42, Figure 11: The caption should specify that the results in this plot cover only Russia/former USSR.

---

## Author Response (AR2)

**Response to Referee #1**

**1. P.3: This is still a problem.**

The MK test solves for the normality problem but not for serial correlation, if present. The MK test assumes independent data! If serial correlation is present (other than trend) the variance of the MK statistic will be underestimated and the probability of detecting a false trend will be increased. See Khaled and Rao, Journal of Hydrology 1998, for example. The authors seem unaware that serial correlation, while not affecting the trend coefficient, will decrease the degree of freedom and possibly invalidate the statistical test on the slopes if undealt with (either the T test AND the MK test). There is much literature on this topic and applying parametric or non-parametric test on time-series MUST account for serial correlation (if present).

To quote H. Von Storch:

"There are, however, again and again cases in which people simply ignore this condition, in particular when dealing with more exotic tests such as the Mann-Kendall test, which is used to reject the null hypothesis of "no trends"."Von Storch and Navarra, 1999. Analysis of Climate Variability: Chapter 2: Misuses of Statistical Analysis in Climate Research, p.17.

I highly recommend that the authors read this chapter for guidance on how to correct for serial correlation, if necessary. I also mentioned several methodological papers on trend detection in time series in my first evaluation that have been ignored. I add another one that will be useful:

Khaliq, M. N., Ouarda, T. B. M. J., Gachon, P., Sushama, L., & St-Hilaire, A. (2009b). Identification of hydrological trends in the presence of serial and cross correlations: A review of selected methods and their application to annual flow regimes of Canadian rivers. Journal of Hydrology, 368(1-4),

Reply: Thank you very much for your detailed comments and concerns. We have read all recommended articles and book chapters. We have added the analysis of serial correlation as an appendix (another option is to directly add the appendix in the main text). We have used the Durbin-Watson test to check the serial correlation and the Cochrane-Orcutt method to correct the variable if serial correlation is present. Then, the trends in annual mean snow depth, maximum snow depth and monthly mean snow depth for each station were recalculated in the text and corrected in the figures throughout the study.

**"Appendix: Analysis of serial correlation**

In this research, the Kolmogorov-Smirnov (K-S) test was used to determine whether snow depth data could have come from a normal distribution. The results showed that all station data were in a normal distribution (such as annual mean snow depth for all station, Fig. 1). We used ordinary linear regression (OLR) to detect trends of changes in snow depth. Failure to consider the serial correlation of data could lead to erroneous results when detect the trends in time series of snow depth mainly because the probability of detecting false trends would be increased (Westherhead et al, 1998; Storch, 1999; Khaliq et al., 2009). In order to avoid this situation, we used the Durbin-Watson test to check the serial correlation (Neter et al., 1989; Tao et al., 2008):

$$\mathbf{d} = \frac{\sum_{t=2}^{n} (e_t - e_{t-1})^2}{\sum_{t=1}^{n} e_t^2} \tag{1}$$

where  $e_t$  is the residual estimated by the OLR.  $d_1$  is the lower limit,  $d_u$  is the upper limit. If  $d_u \le d \le 4 - d_u$  serial correlation is not absent, if  $d \le d_1$  or  $d \ge 4 - d_1$ serial correlation is present.

Figure 1. Normal distribution test of annual mean snow depth for all station by K-S test.

We used Cochrane-Orcutt method to correct the variable if serial correlation is present (Neter et al., 1989; Tao et al., 2008):

$$X_t' = X_t - \rho X_{t-1} \tag{2}$$

$$Y_t' = Y_t - \rho Y_{t-1} \tag{3}$$

where X' is the corrected year, Y' is the corrected anomalies in time series of snow depth for each station in this research, the autocorrelation coefficient  $\rho$  is replaced by its estimated r :

$$\mathbf{r} = \frac{\sum_{t=2}^{n} e_{t-1} e_t}{\sum_{t=2}^{n} e_{t-1}^2} \tag{4}$$

then, the Durbin-Watson test was used to check the serial correlation of the new snow depth anomalies, and recalculated the trends in time series of new data.

The Durbin-Watson test results showed that there were no serial correlations in the inter-annual trends of annual mean snow depth, maximum snow depth and monthly mean snow depth for all composite data ( $d_u \le d \le 4 - d_u$ ) (Table 1). However, the serial correlation was present in some stations when we calculated the linear trend of annual snow depth, maximum depth and monthly mean snow depth for each station. The percentage of the stations with serial correlation for annual snow depth and maximum depth were 18% and 21%, respectively. In the monthly test, the smallest proportion appeared in October, about 11%; the largest percentage of these stations for all stations was found in February, up to 21%. Then Cochrane-Orcutt method was used to correct the variables and re-estimated the trends in snow depth for these station (Fig. 6-7 in the text). Took Dikson site (73.5°N, 80.4°E, 42m a.s.l.) as an example: the serial correlation was present when the trend in annual mean snow depth was calculated. Comparing with the corrected result, the variance of the previous OLR statistic was overestimated (Table 2). The corrected result indicated that the variation of inter-annual mean snow depth was not significant (P' > 0.05). The serial correlation cannot be ignored for detecting trends in time series of snow cover variables, which possibly invalidate the statistical test on the slopes if undealt with.

|          | $d_u$  | d      | slope | P*     |
|----------|--------|--------|-------|--------|
| Mean     | 1.3525 | 1.6435 | 0.02  | 0.0016 |
| Maximum  | 1.6534 | 1.8824 | 0.06  | 0.0004 |
| October  | 1.6324 | 2.1377 | -0.01 | 0.0069 |
| November | 1.6277 | 2.3667 | 0.00  | 0.7408 |
| December | 1.6532 | 1.9684 | 0.02  | 0.0793 |
| January  | 1.3542 | 1.6326 | 0.04  | 0.0014 |
| February | 1.6521 | 1.8469 | 0.06  | 0.0000 |
| March    | 1.4536 | 1.9874 | 0.06  | 0.0003 |
| April    | 1.3242 | 1.6754 | 0.03  | 0.0187 |
| May      | 1.7726 | 2.0703 | 0.00  | 0.5811 |

Table1. Trends in snow depths with the Durbin-Watson test across Eurasia during 1966-2012

\*: P is the confidence level.

Table 2. Trends in annual mean snow depth with the Durbin-Watson test for Dikson site during

| 10   | 11 | 00  | 10 |  |
|------|----|-----|----|--|
| - 19 | 90 | -20 | Л2 |  |

| ID    | $d_u$  | d      | slope | Р     | $d'_u$ | d'     | slope' | P'    |
|-------|--------|--------|-------|-------|--------|--------|--------|-------|
| 20674 | 1.7234 | 1.2856 | 0.113 | 0.016 | 1.6345 | 2.0249 | 0.0942 | 0.055 |

**References**

- Khaliq, M.N., Ouarda, T.B.M.J., Gachon, P., Sushama, L., and St-Hilaire, A.: Indentification of hydrological trends in the presence of serial and cross correlations: A review of selected methods and their application to annual flow regimes of Canadian rivers, J. Hydrol., 368, 117-130, 2009.
- Neter, J., Wasserman, W., and Kutner, M.H.: Applied linear regression model, Boston, IRWIN, 1989.
- Storch, H.V.: Misuses of Statistical Analysis in Climate Research, in: Analysis of Climate Variability, edited by: Storch, H.V. and Navarra, A.: Springer Press, Berlin Heidelberg, Germany, 11–26, 1999.
- Tao, J., Zhang, X., Tao, J., and Shen, Q.: The checking and removing of the autocorrelation in climatic time series, Journal of Applied Meteorological Science, 19, 47-52, 2008.
- Weatherhead, E.C., Reinsel, G.C., Tiao, G.C., Meng, X., Choi, D., Cheang, W., Keller, T., DeLuisi, J., Wuebbles, D.J., Kerr, J.B., Miller, A.J., Oltmans, S.J., and Frederick, J.E.: Factors affecting the detection of trends: Statistical considerations and applications to environmental data, J. Geophys. Res., 103(D14), 17149-17161, 1998."

2. P.6: That is simply not true. The trend exceed the variability, assuming this variability is independently distributed over time, and, for the t-test, normally distributed. Hence if the noise is 'white' (not serial correlation) then the statement is true, otherwise not. and this was not tested and reported in the study.

Reply: Thank you for your comments. We have used the Durbin-Watson test to check the serial correlation and the Cochrane-Orcutt method to correct the variable if serial correlation is present. The details can be found in the first reply.

3. P.12: the first reply to referee #2 for the general comments, "the same kind of instruments" which one? You could describe measurement methods in one sentence in the text. Manual measurements with a ruler on a snow plate? Automatic ultrasonic range sensors? Incoroportate in the text

Reply: We have added more description of the measurement method:

"Snow depth was measured once a day at meteorological stations using a graduated stake installed at a fixed point location within the station or by a wooden ruler."

4. P. 13: See my long comments in response to a similar comment made by Ref1.

1) If, as you say, you have found your snow depth data to be normally distributed, then this one less concern for parametric statistical testing using the T-test on the regression slope. You would not have to use a non-parametric test such as the MK test. But fine, two tests gab strengthen your conclusions

**Reply: We have deleted the MK test because the snow depth data were normally distributed.**

2) Both test, the parametric t-test and the non-parametric MK test, assume independent data, and you have completed ignored this comment. The MK test does not overcome the assumption of independent data, which rarely holds for time-series. At the minimum you need to report, in the manuscript, the range of lag1 autocorrelation coefficient in order to support a claim that the data is approximately independent. I understand this is a tough issue that is often ignored in several studies, and without 'simple fixes' (see Von Storch and Navarra, 1999, p.17 for discussion, and other papes cited), but it MUST be address. Otherwise why bother applying a statistical test on the slopes?

Reply: We have added the analysis of serial correlation as an appendix. We have used the Durbin-Watson test to check the serial correlation and the Cochrane-Orcutt method to correct the variable if serial correlation is present. Then, the trends in annual mean snow depth, maximum snow depth and monthly mean snow depth for each station were recalculated in the text and corrected in the figures throughout the study.

**"Appendix: Analysis of serial correlation**

In this research, the Kolmogorov-Smirnov (K-S) test was used to determine whether snow depth data could have come from a normal distribution. The results showed that all station data were in a normal distribution (such as annual mean snow depth for all station, Fig. 1). We used ordinary linear regression (OLR) to detect trends of changes in snow depth. Failure to consider the serial correlation of data could lead to erroneous results when detect the trends in time series of snow depth mainly because the probability of detecting false trends would be increased (Westherhead et al, 1998; Storch, 1999; Khaliq et al., 2009). In order to avoid this situation, we used the Durbin-Watson test to check the serial correlation (Neter et al., 1989; Tao et al., 2008):

$$d = \frac{\sum_{t=2}^{n} (e_t - e_{t-1})^2}{\sum_{t=1}^{n} e_t^2}$$
(1)

where  $e_t$  is the residual estimated by the OLR.  $d_1$  is the lower limit,  $d_u$  is the upper limit. If  $d_u \le d \le 4 - d_u$  serial correlation is not absent, if  $d \le d_1$  or  $d \ge 4 - d_1$ serial correlation is present.

Figure 1. Normal distribution test of annual mean snow depth for all station by K-S test.

We used Cochrane-Orcutt method to correct the variable if serial correlation is present (Neter et al., 1989; Tao et al., 2008):

$$X'_{t} = X_{t} - \rho X_{t-1}$$
 (2)
 $Y'_{t} = Y_{t} - \rho Y_{t-1}$  (3)

where X' is the corrected year, Y' is the corrected anomalies in time series of snow depth for each station in this research, the autocorrelation coefficient  $\rho$  is replaced by its estimated r :

$$\mathbf{r} = \frac{\sum_{t=2}^{n} e_{t-1} e_t}{\sum_{t=2}^{n} e_{t-1}^2} \tag{4}$$

then, the Durbin-Watson test was used to check the serial correlation of the new snow depth anomalies, and recalculated the trends in time series of new data.

The Durbin-Watson test results showed that there were no serial correlations in the inter-annual trends of annual mean snow depth, maximum snow depth and monthly mean snow depth for all composite data ( $d_u \le d \le 4 - d_u$ ) (Table 1). However, the serial correlation was present in some stations when we calculated the linear trend of annual snow depth, maximum depth and monthly mean snow depth for each station. The percentage of the stations with serial correlation for annual snow depth and maximum depth were 18% and 21%, respectively. In the monthly test, the smallest proportion appeared in October, about 11%; the largest percentage of these stations for

all stations was found in February, up to 21%. Then Cochrane-Orcutt method was used to correct the variables and re-estimated the trends in snow depth for these station (Fig. 6-7 in the text). Took Dikson site (73.5°N, 80.4°E, 42m a.s.l.) as an example: the serial correlation was present when the trend in annual mean snow depth was calculated. Comparing with the corrected result, the variance of the previous OLR statistic was overestimated (Table 2). The corrected result indicated that the variation of inter-annual mean snow depth was not significant (P'>0.05). The serial correlation cannot be ignored for detecting trends in time series of snow cover variables, which possibly invalidate the statistical test on the slopes if undealt with.

|          | $d_u$  | d      | slope | $P^{*}$ |
|----------|--------|--------|-------|---------|
| Mean     | 1.3525 | 1.6435 | 0.02  | 0.0016  |
| Maximum  | 1.6534 | 1.8824 | 0.06  | 0.0004  |
| October  | 1.6324 | 2.1377 | -0.01 | 0.0069  |
| November | 1.6277 | 2.3667 | 0.00  | 0.7408  |
| December | 1.6532 | 1.9684 | 0.02  | 0.0793  |
| January  | 1.3542 | 1.6326 | 0.04  | 0.0014  |
| February | 1.6521 | 1.8469 | 0.06  | 0.0000  |
| March    | 1.4536 | 1.9874 | 0.06  | 0.0003  |
| April    | 1.3242 | 1.6754 | 0.03  | 0.0187  |
| May      | 1.7726 | 2.0703 | 0.00  | 0.5811  |

Table1. Trends in snow depths with the Durbin-Watson test across Eurasia during 1966-2012

\*: P is the confidence level.

Table 2. Trends in annual mean snow depth with the Durbin-Watson test for Dikson site during

|       | 1900-2012 |        |       |       |        |        |        |       |
|-------|-----------|--------|-------|-------|--------|--------|--------|-------|
| ID    | $d_u$     | d      | slope | Р     | $d'_u$ | ď      | slope' | Ρ'    |
| 20674 | 1.7234    | 1.2856 | 0.113 | 0.016 | 1.6345 | 2.0249 | 0.0942 | 0.055 |

**Response to Referee #2**

**General Comments:**

1. In my initial review, I commented: "This study examines the characteristics and trends across the Eurasian continent from 1966 to 2012. To do so, the authors assemble snow

depth data from 1103 stations across the study area. How representative are the station (point) snow depth data of the overall regional landscapes of interest? For instance, are snow depth data in forested areas collected at airports or other open areas, that may not represent the regional snow characteristics?" The authors acknowledge the shortcomings of the station distribution used in their study but do not address the point in question. Are the results based on point observations representative of the vast region under study?

Reply: For the instance, all the snow depth data in forest areas are collected just in forest, not at airport or open areas. The basic principle of site selection is as much as possible representing the surrounding environment. At the same time, the snow course data is also a supplement to the site data. Here for the first time, we present all data we can possibly collect from various countries over the continent and show snow depth spatial variations and temporal changes. These snow depth data can represent the regional variability and this in-situ dataset and its coverage is unprecedented. The purpose of our research is to present the spatiotemporal variations in snow depth. Therefore, we believe that the in-situ data can be used to achieve this goal.

2. The authors provide comprehensive information on snow data collection in the former USSR, but fail to report similar information for other countries. How is snow depth measured across Eurasia? Has sampling changed to automated sensors (e.g. sonic rangers) in recent decades? Little information is provided on the data collection process and the accuracy of the measurements.

Reply: snow depth is measured by a graduated stake installed at the station or a wooden ruler on a daily basis, and never change the measurement method. We have added the description of snow depth collection process:

"Snow depth was measured once a day at meteorological stations using a graduated stake installed at a fixed point location within the station or by a wooden ruler. Snow depth was measured using the same method across Eurasian continent since the meteorological observation standard was established by the former USSR and followed by all the former USSR republics, Mongolia and China. Snow depth is one of the standard elements to be measured on daily basis (WMO, 1996)."

Further to this, how is homogeneity in the time series of snow depth, SWE, and other variables assured if sampling techniques or instruments have changed over time?

Reply: the procedures for taking snow observation changed in 1965, and there has been no change in procedure and techniques since then. In this study, we only chose to use the data after 1965 (1996-2012) to ensure the homogeneity of the data. We explained this in the manuscript:

"Procedures and techniques for measuring snow depth may have changed over the course of station history. Consequently, snow depth data may have inhomogeneities in the time series over the period of record. Forturnately, there was no change in procedure and technique of snow depth measurements since 1965 in Russia and the other countries in this study (Bulygina et al., 2009). In this study, therefore, we chose to use snow depth data from 1966 to 2012."

Have the time series been tested for homogeneity (i.e. discontinuities in the data)?

Reply: We collected 2160 stations with snow depth data, however, we just selected 1814 stations in this study because of some stations with discontinuous data. The test had been described in the manuscript:

"We implemented additional quality control using the following requirements: (1) to ensure snow depth stability, at a given location, a month with less than 15 days of snow depth measurements is deleted. (2) Stations with sudden step changes of snow depth are eliminated from the list. (3) Stations with less than 20 years of data during the 1971-2000 period were excluded from the analysis. (4) At each station, we eliminated data points which exceed two standard deviations from their long-term (1971-2000) mean."

Finally, no information is provided on how air temperature and precipitation measurements were made at the meteorological stations. Snowfall measurements are notoriously difficult to make and gauge undercatch correction factors must be applied to obtain improved estimates of snowfall, particularly in windy environments such as Arctic and alpine tundra. The entire section describing the observational data used in the present study must be improved and expanded. Such details may be provided in a supplementary document as necessary.

Reply: We have added the description of the air temperature and precipitation

measurement. The snowfall data are estimated with air temperature and precipitation because there was no special snowfall observation. The original precipitation data were not corrected by considering the gauge undercatch, etc.

"Daily air temperature was measured by thermometer which was placed at a height of 1.5 m above the ground surface in an instrument shelter at meteorological station (WMO, 1996). Air temperature measurement should be accurate to 0.1°C. Air temperature was measured four times a day at 0200, 0800, 1400, and 2000 at local time. Daily mean air temperature was calculated by simple arithmetic average of the four measurements, while monthly mean was based on daily mean and annual mean was based on monthly mean. Precipitation was gathered and measured by a precipitation gauge and was reported with a 0.1-mm precision (Groisman and Rankova, 2001). Original precipitation data were not corrected by considering the gauge undercatch."

"Daily precipitation was partitioned into a solid and liquid fraction, based on daily mean temperature (Brown, 2000). The solid fraction of precipitation, Srat, was estimated by:

$$S_{rat} = \begin{cases} 1.0 & for \ T_{mean} \le -2.0^{\circ}C, \\ 0.0 & for \ T_{mean} \ge +2.0^{\circ}C, \\ 1.0 - 0.25(T_{mean} + 2.0) & for \ -2.0^{\circ}C

---

## Editor Decision (ED2)

[revised manuscript text omitted]
       | the former                           | 1044 | RIHMI-WDC, NSIDC                    |  |  |
| snow courses          | USSR                                 |      |                                     |  |  |
| Daily air temperature | the former                           | 386  | RIHMI-WDC                           |  |  |
| and precipitation     | USSR                                 |      |                                     |  |  |

3

4 **Table A1.** Trends in snow depths with the Durbin-Watson test across Eurasia during 1966-2012

|          | $d_1$  | $d_u$  | d      | slope* | $p^{*}$ |
|----------|--------|--------|--------|--------|---------|
| Mean     | 1.3034 | 1.3871 | 1.6435 | 0.02   | 0.0016  |
| Maximum  | 1.3034 | 1.3871 | 1.8824 | 0.06   | 0.0004  |
| October  | 1.3034 | 1.3871 | 2.1377 | -0.01  | 0.0069  |
| November | 1.4872 | 1.5739 | 2.3667 | 0.00   | 0.7408  |
| December | 1.4872 | 1.5739 | 1.9684 | 0.02   | 0.0793  |
| January  | 1.3034 | 1.3871 | 1.6326 | 0.04   | 0.0014  |
| February | 1.3034 | 1.3871 | 1.8469 | 0.06   | 0.0000  |
| March    | 1.3034 | 1.3871 | 1.9874 | 0.06   | 0.0003  |
| April    | 1.3034 | 1.3871 | 1.6754 | 0.03   | 0.0187  |
| May      | 1.4872 | 1.5739 | 2.0703 | 0.00   | 0.5811  |

5 \*: slope is the trend of changes in snow depth, the unit is cm yr-1; p is the confidence level.

6 **Table A2.** Trends in annual mean snow depth with the Durbin-Watson test for the Dikson site

| during 1966-2012 |        |        |        |       |       |              |        |        |         |        |
|------------------|--------|--------|--------|-------|-------|--------------|--------|--------|---------|--------|
| ID               | $d_1$  | $d_u$  | d      | slope | p     | $d_1^\prime$ | $d'_u$ | d′     | slope'* | $p'^*$ |
| 20674            | 1.3034 | 1.3871 | 1.2856 | 0.113 | 0.016 | 1.4872       | 1.5739 | 2.0249 | 0.0942  | 0.055  |

8 \*: slope' is the corrected trend of changes in snow depth, the unit is cm yr-1; p' is the corrected confidence level.

9

---

## Author Response (AR3)

**Response to Referee #1**

1. P.3: This is still a problem.

The MK test solves for the normality problem but not for serial correlation, if present. The MK test assumes independent data! If serial correlation is present (other than trend) the variance of the MK statistic will be underestimated and the probability of detecting a false trend will be increased. See Khaled and Rao, Journal of Hydrology 1998, for example. The authors seem unaware that serial correlation, while not affecting the trend coefficient, will decrease the degree of freedom and possibly invalidate the statistical test on the slopes if undealt with (either the T test AND the MK test). There is much literature on this topic and applying parametric or non-parametric test on time-series MUST account for serial correlation (if present).

To quote H. Von Storch:

"There are, however, again and again cases in which people simply ignore this condition, in particular when dealing with more exotic tests such as the Mann-Kendall test, which is used to reject the null hypothesis of "no trends"."Von Storch and Navarra, 1999. Analysis of Climate Variability: Chapter 2: Misuses of Statistical Analysis in Climate Research, p.17.

I highly recommend that the authors read this chapter for guidance on how to correct for serial correlation, if necessary. I also mentioned several methodological papers on trend detection in time series in my first evaluation that have been ignored. I add another one that will be useful:

Khaliq, M. N., Ouarda, T. B. M. J., Gachon, P., Sushama, L., & St-Hilaire, A. (2009b). Identification of hydrological trends in the presence of serial and cross correlations: A review of selected methods and their application to annual flow regimes of Canadian rivers. Journal of Hydrology, 368(1-4),

Reply: Thank you very much for your detailed comments and concerns. We have read all recommended articles and book chapters. We have added the analysis of serial correlation as an appendix (another option is to directly add the appendix in the main text). We have used the Durbin-Watson test to check the serial correlation and the Cochrane-Orcutt method to correct the variable if serial correlation is present. Then, the trends in annual mean snow depth, maximum snow depth and monthly mean snow depth for each station were recalculated in the text and corrected in the figures throughout the study.

"**Appendix A:** Analysis of serial correlation

[revised manuscript text omitted]

2. P.6: That is simply not true. The trend exceed the variability, assuming this variability is independently distributed over time, and, for the t-test, normally distributed. Hence if the noise is 'white' (not serial correlation) then the statement is true, otherwise not. and this was not tested and reported in the study.

Reply: Thank you for your comments. We have used the Durbin-Watson test to check the serial correlation and the Cochrane-Orcutt method to correct the variable if serial correlation is present. The details can be found in the first reply.

3. P.12: the first reply to referee #2 for the general comments, "the same kind of instruments" which one? You could describe measurement methods in one sentence in the text. Manual measurements with a ruler on a snow plate? Automatic ultrasonic range sensors? Incoroportate in the text

Reply: We have added more description of the measurement method:

"Snow depth was measured once a day at meteorological stations using a graduated stake installed at a fixed point location within the station or by a wooden ruler."

4. P. 13: See my long comments in response to a similar comment made by Ref1.

1) If, as you say, you have found your snow depth data to be normally distributed, then this one less concern for parametric statistical testing using the T-test on the regression slope. You would not have to use a non-parametric test such as the MK test. But fine, two tests gab strengthen your conclusions

Reply: We have deleted the MK test because the snow depth data were normally distributed.

2) Both test, the parametric t-test and the non-parametric MK test, assume independent data, and you have completed ignored this comment. The MK test does not overcome the assumption of independent data, which rarely holds for time-series. At the minimum

you need to report, in the manuscript, the range of lag1 autocorrelation coefficient in order to support a claim that the data is approximately independent. I understand this is a tough issue that is often ignored in several studies, and without 'simple fixes' (see Von Storch and Navarra, 1999, p.17 for discussion, and other papes cited), but it MUST be address. Otherwise why bother applying a statistical test on the slopes?

Reply: We have added the analysis of serial correlation as an appendix. We have used the Durbin-Watson test to check the serial correlation and the Cochrane-Orcutt method to correct the variable if serial correlation is present. Then, the trends in annual mean snow depth, maximum snow depth and monthly mean snow depth for each station were recalculated in the text and corrected in the figures throughout the study.

"**Appendix A:** Analysis of serial correlation

[revised manuscript text omitted]

**Response to Referee #2**

**General Comments:**

1. In my initial review, I commented: "This study examines the characteristics and trends across the Eurasian continent from 1966 to 2012. To do so, the authors assemble snow

depth data from 1103 stations across the study area. How representative are the station (point) snow depth data of the overall regional landscapes of interest? For instance, are snow depth data in forested areas collected at airports or other open areas, that may not represent the regional snow characteristics?" The authors acknowledge the shortcomings of the station distribution used in their study but do not address the point in question. Are the results based on point observations representative of the vast region under study?

Reply: For the instance, all the snow depth data in forest areas are collected just in forest, not at airport or open areas. The basic principle of site selection is as much as possible representing the surrounding environment. At the same time, the snow course data is also a supplement to the site data. Here for the first time, we present all data we can possibly collect from various countries over the continent and show snow depth spatial variations and temporal changes. These snow depth data can represent the regional variability and this in-situ dataset and its coverage is unprecedented. The purpose of our research is to present the spatiotemporal variations in snow depth. Therefore, we believe that the in-situ data can be used to achieve this goal.

2. The authors provide comprehensive information on snow data collection in the former USSR, but fail to report similar information for other countries. How is snow depth measured across Eurasia? Has sampling changed to automated sensors (e.g. sonic rangers) in recent decades? Little information is provided on the data collection process and the accuracy of the measurements.

Reply: snow depth is measured by a graduated stake installed at the station or a wooden ruler on a daily basis, and never change the measurement method. We have added the description of snow depth collection process:

"Snow depth was measured once a day at meteorological stations using a graduated stake installed at a fixed point location within the station or by a wooden ruler. Snow depth was measured using the same method across the Eurasian continent since the meteorological observation standard was established by the former Union of Soviet Socialist Republics (USSR) and followed by all of the former USSR republics, Mongolia and China. Snow depth is one of the standard elements to be measured on a

daily basis (WMO, 1996)."

Further to this, how is homogeneity in the time series of snow depth, SWE, and other variables assured if sampling techniques or instruments have changed over time?

Reply: the procedures for taking snow observation changed in 1965, and there has been no change in procedure and techniques since then. In this study, we only chose to use the data after 1965 (1996-2012) to ensure the homogeneity of the data. We explained this in the manuscript:

"Procedures and techniques for measuring snow depth may have changed over the course of station history. Consequently, snow depth data may not be homogeneous in the time series over the period of the record. Fortunately, there was no change in the procedure and technique of snow depth measurements since 1965 in Russia and the other countries in this study (Bulygina et al., 2009). Therefore, in this study, we chose to use snow depth data from 1966 to 2012."

Have the time series been tested for homogeneity (i.e. discontinuities in the data)?

Reply: We collected 2160 stations with snow depth data, however, we just selected 1814 stations in this study because of some stations with discontinuous data. The test had been described in the manuscript:

"We implemented additional quality control using the following requirements: (1) To ensure snow depth stability, at a given location, a month with less than 15 days of snow depth measurements was deleted; (2) Stations with sudden and steep changes in snow depth were eliminated from the list; (3) Stations with less than 20 years of data during the 1971-2000 period were excluded from the analysis; and (4) At each station, we eliminated data points that exceeded two standard deviations from their long-term (1971-2000) mean."

Finally, no information is provided on how air temperature and precipitation measurements were made at the meteorological stations. Snowfall measurements are notoriously difficult to make and gauge undercatch correction factors must be applied to obtain improved estimates of snowfall, particularly in windy environments such as Arctic and alpine tundra. The entire section describing the observational data used in the present study must be improved and expanded. Such details may be provided in a

supplementary document as necessary.

Reply: We have added the description of the air temperature and precipitation measurement. The snowfall data are estimated with air temperature and precipitation because there was no special snowfall observation. The original precipitation data were not corrected by considering the gauge undercatch, etc.

"Daily air temperature was measured using a thermometer, which was placed at a height of 1.5 m above the ground surface in an instrument shelter at the meteorological station (WMO, 1996). The air temperature measurement should be accurate to 0.1 ℃. Air temperature was measured four times a day at 0200, 0800, 1400, and 2000 local time. The daily mean air temperature was calculated by a simple arithmetic average of the four measurements, whereas the monthly mean was based on the daily mean and the annual mean was based on the monthly mean. Precipitation was gathered and measured by a precipitation gauge and was reported with a 0.1-mm precision (Groisman and Rankova, 2001). The original precipitation data were not corrected by considering the gauge undercatch."

"Daily precipitation was partitioned into a solid and liquid fraction based on daily mean temperature (Brown, 2000). The solid fraction of precipitation, Srat, was estimated by

$$S_{rat} = \begin{cases} 1.0 & \text{for } T_{mean} \leq -2.0°C, \\ 0.0 & \text{for } T_{mean} \geq +2.0°C, \\ 1.0 - 0.25(T_{mean} + 2.0) & \text{for } -2.0°C < T_{mean} < +2.0°C. \end{cases} \quad (1)$$

where $T_{mean}$ is the mean daily air temperature ( ℃)."

3. In response to another comment I made (as well as by Referee #2), the authors now employ the Mann-Kendall test to assess linear trends in addition to linear regressions. However, they fail to address the issue of serial correlation impacts on the trend analyses (as raised by Referee #2). This must be addressed before the paper can be considered for publication.

Reply: We have added the analysis of serial correlation as an appendix. We have used the Durbin-Watson test to check the serial correlation and the Cochrane-Orcutt method to correct the variable if serial correlation is present. Then, the trends in annual mean

snow depth, maximum snow depth and monthly mean snow depth for each station were recalculated in the text and corrected in the figures throughout the study.

"**Appendix A:** Analysis of serial correlation

[revised manuscript text omitted]

4. Further to this, the (revised) Figures 5 and 7 are confusing – what results do these figures represent? The Mann-Kendall trend analysis should give you one slope value over a period of study. No details are provided in the Data/Methods section on how the results presented in these figures are obtained. Further to this, what are "UF" and "UB" in these figures?

Reply: We have found snow depth data to be normally distributed, the ordinary linear regression can be used to analyze the trend in time series. Therefore, we decide not to use the MK test (non-parametric test) again. Fig.5 and Fig.7 have been deleted.

5. In my initial review, I commented: "Do the linear trends reported in Section 3.2 exceed the variability in the snow depth data? In other words, are there "detectable" trends in snow depth, i.e. with the signal greater than the noise in the system?" The authors' response does not fully address this issue, i.e. whether the slopes of the linear trends (signal) exceed the standard deviation (noise) in the snow parameters of interest.

Reply: We have analyzed the correlation between the slope of the linear trends and the standard deviation. The results present that the noise exceed the signal in many stations (Fig. 1). This is due to the variations of snow depth are effected by a variety of factors, which lead to the large interannual differences in snow depth. However, the long-term trends are not significant and slopes are not large. Then, we calculate the residuals of snow depth and analyze the "white" noise with QQplot for each station (Fig. 2). The results show residuals are the normal distribution, that is, the noise is "white". The

analysis of serial correlation also prove the same result.

[Figure]

Figure. 1 The correlation between the slope of the linear trends and the standard deviation for annual mean snow depth for each station.

[Figure]

Figure.2 The "white" noise test for site 20046, which the noise exceed the signal.

6. The Discussion remains relatively brief and could be augmented by placing these results in a larger context. Do these results concord with modeling studies of snow across Eurasia? What are the prospects for future snow cover changes in Eurasia? What are the broader implications of the results to regional hydrology, permafrost distribution, ecology and society?

Reply: We have added a comparison with the results of modeling studies and modified the Discussion section. The purpose of our research is to analyze and clarify the climatology and spatiotemporal variations in snow depth across Eurasia. The prediction of the prospects for future snow cover changes should be combined with the model simulation. We hope our results can provide important reference for estimating the simulation.

**"4 Discussion**

[revised manuscript text omitted]

7. The names of countries or their abbreviations can be removed on all figures after Figure 1.

Reply: All names and abbreviations are removed after Fig. 1.

8. Please improve the language throughout the paper – there are portions of the text that are difficult to comprehend due to language issues, including all of Section 4.2. Furthermore, the verb tense in the introduction changes constantly and only one tense should be used consistently.

Reply: We have the manuscript proofread and revised by the professional English editing service from American Journal Experts.

**Specific Comments:**

1. P. 3, line 13: Replace "reduced" with "declined".

Reply: Has been done.

2. P. 3, lines 15-18: The grammar in this sentence is poor – please rephrase.

Reply: We have rephrased the sentence:

"This may be explained in that the warmer air led to a greater moisture supply for snowfall in winter (Ye et al., 1998; Kitaev et al., 2005; Rawlins et al., 2010)."

3. P. 3, line 27: Replace "was" with "were".

Reply: Replaced "but there was" with "and showed large".

4. P. 4, line 20: What aspect of "passive microwave" improved the algorithms?

Reply: We have rephrased the sentence:

"…or developed and/or improved passive microwave snow algorithms"

5. P. 4, line 25-27: Language needs much improvement here.

Reply: We have rephrased the sentence:

"In addition, data acquisition from large airborne equipment or aerial systems is costly

and strict data use limitations apply."

6. P. 4, line 29: Delete the hyphen after "longer". Insert "the" before "climatology".

Reply: Has been done.

7. P. 5, line 26: Do you mean "and during the snowmelt period (every five days)"?

Reply: Yes, We have revised it.

8. P. 6, line 7: Delete "the following Equation (1)"

Reply: Has been done.

9. P. 6, lines 25-27: Rephrase this sentence.

Reply: we have rephrased the sentence:

"We defined a snow year starting from July 1$^{st}$ through June 30$^{th}$ of the following year to capture the entire seasonal snow cycle."

10. P. 6, line 28: Change to "study period".

Reply: Has been changed.

11. P. 7, line 14: Replace the colon after "2012" with a period.

Reply: we deleted the sentence.

12. P. 7, lines 23-27: These sentences need to be rephrased.

Reply: we have rephrased the sentence:

 "Anomalies of monthly, annual mean, and annual mean maximum snow depth from their long-term (1971-2000) records were calculated for each station across the Eurasian continent. Composite time series of monthly and annual anomalies were obtained by using all of the available station data across the study area."

13. P. 8, line 8: What do you mean with "Despite there is a nonlinearity".

Reply: we have rephrased the sentence:

"The linear trend analysis is also a useful approximation when systematic low-frequency variations emerged even though there is a nonlinearity."

14. P. 8, line 9: Delete "a" before "systematic".

Reply: Has been done.

15. P. 8, line 19: Delete "In order".

Reply: Has been deleted.

16. P. 8, line 20: Insert "a" before "single".

Reply: Deleted the sentences.

17. P. 8, lines 27-29: Rephrase this sentence. Insert a space after "(Fig. 2)".

Reply: we have rephrased the sentence:

"Distributions of long-term mean snow depth indicated a strong latitudinal zonality. Generally, snow depth increased with latitude northward across the Eurasian continent (Fig. 2)."

18. P. 11, line 2: What do you mean by "fluctuating changed"?

Reply: "fluctuating changed" means the changes in snow depth increased in some years, while decreased in the next period, alternating with each period.

19. P. 11, line 6 and elsewhere: Replace "confident level" with "confidence level".

Reply: Has been done.

20. P. 12, line 5: What do you mean by "fluctuant increasing trend"?

Reply: "fluctuant increasing trend" means there was a generally increasing trend in snow depth, but the changes in snow depth increased in some years, while decreased in the next period, alternating with each period.

21. P. 12, line 10 and elsewhere: Replace "confident level" with "confidence level".

Reply: Has been done.

22. P. 12, line 30 and elsewhere: Delete spaces between the degree sign and North, i.e. "40°N".

Reply: Has been deleted.

23. P. 13, line 5: Replace "Eurasian areas" with "Eurasia".

Reply: Has been done.

24. P. 14, lines 12-13: Language must be improved here.

Reply: we have deleted the sentence.

25. P. 14, line 25: Insert the p-value for the correlation coefficient.

Reply: We have inserted "P≤0.05,"

26. P. 15, line 9: Change to "at most".

Reply: Has been done.

27. P. 15, line 22: Replace "lowed" with "lowered".

Reply: Has been done.

28. P. 16, line 2: Delete "the" before "northern".

Reply: Has been deleted.

29. P. 16, lines 3-5: This sentence must be re-written.

Reply: we have rephrased the sentence:

"This was because there was no obvious effect of increasing temperature on snow depth when the air temperature was below 0 ℃ which occurred in most areas of Siberia from December through March."

30. P. 17, line 12: Delete "the" before "southern".

Reply: Has been deleted.

31. P. 17, line 16: This entire section is poorly phrased and needs to be completely

revised. Why does the font size change in the middle of the paragraph?

Reply: We have deleted the section and added the statement in "3.3 Variability of Snow Depth with Latitude, Elevation and Continentality" section:

"Topography is an important factor affecting the climatology of snow depth and is the main reason accounting for the inhomogeneity of data (Grünewald and Lehning, 2011, 2013; Grünewald et al., 2014). To explore the spatial variability of snow depth, we conducted a linear regression analysis of the annual mean snow depth with latitude, elevation and continentality (Fig. 8). Snow depth was positively correlated with latitude, i.e., snow depth generally increased with latitude (Fig. 8a). The increased rate of snow depth was approximately 0.81 cm per 1 °N across the Eurasian continent. A closer relationship between latitude and snow depth was found in regions north of 40 °N (Figs. 8a and d) where snow cover was relatively stable with the number of annual mean continuous snow cover days at more than 30 (Zhang and Zhong, 2014).

There was a negative correlation between snow depth and elevation across the Eurasian continent (Fig. 8b); with every 100 m increase in elevation, snow depth decreased by ~0.5 cm ($P \leqslant 0.05$). Annual mean snow depth was less than 1 cm in most areas, with an elevation greater than 2000 m because a snow depth of 0 cm was used to calculate the mean snow depth. Therefore, although the TP is at a high elevation, the shallow snow depth in this area resulted in a generally negative correlation between snow depth and elevation across the Eurasian continent. However, we also found that snow depth increased with elevation in most regions north of 45 °N (Fig. 8d).

There was a statistically significant positive relationship between snow depth and continentality over the Eurasian continent ($r=0.1$, $P \leqslant 0.05$, Fig. 8c). This indicated that the continentality may be not an important driving factor of snow depth distribution over Eurasia, especially on the TP. Although the previous studies showed that the Tibetan Plateau's largest snow accumulation occurred in the winter, the precipitation during the winter months was the smallest of the year (Ma, 2008). This

was mainly due to the majority of annual precipitation that occurs during the summer monsoon season on the TP, which causes much less precipitation during the winter half year (or the snow accumulated season). ”

32. P. 18, line 18: Replace "increase" with "increasing".

Reply: Has been done.

33. P. 18, line 25: Delete "the" before "southern".

Reply: Has been deleted.

34. P. 20, line 16: Note spelling mistake in "Atmos."

Reply: Has been revised.

35. P. 20, lines 27-28: Why are editors of a special journal issue listed here?

Reply: Has been revised.

"Callaghan, T. V., Johansson, M., Brown, R. D., Groisman, P. Ya., Labba, N., and Radionov, V.: The changing face of Arctic snow cover: A synthesis of observed and projected changes, Ambio, 40, 17-31. doi:10.1007/s13280-011-0212-y, 2011."

36. P. 21, line 18: Is this "Hydrol. Sci. J."?

Reply: Yes, We have revised it.

37. P. 21, lines 22-23: Why are upper case letters provided for each word in the title of this article?

Reply: Has been revised.

"Foster, J.L., Chang, A.T.C., and Hall, D.K.: Comparison of snow mass estimates from a prototype passive microwave snow algorithm, Remote Sens. Environ., 62, 132-142, 1997."

38. P. 22, line 19: Insert a hyphen in "Snow atmosphere".

Reply: Has been done.

39. P. 25, Table 1: Change to "snow courses"

Reply: Has been done.

40. P. 26, Figure 1: Why does the orientation of the triangles change across the figure? The top of the triangle should point directly northward to provide a consistent pattern across the figure.

Reply: The figure was drawn by ArcGIS, and the projection coordinate was used. Therefore, it seemed the top of triangle did not point directly northward. We have replaced triangle with circle.

[Figure]

41. P. 27, Figure 2 and subsequent figures: Delete all country names/abbreviations on the maps providing spatial results as this can be found on Figure 1.

Reply: Has been done.

42. P. 30, Figure 4: It is unclear why the authors use wavelets to extract low frequency in the time series of snow depth anomalies. Why not just use a running mean of the data?

Reply: running mean is the average statistics of the data, then simulated the trends in snow depth on the basis of average. This method will result in the missing information

of the former and later years.

43. P. 31, Figure 5: The results presented in this figure and in Figure 7 are difficult to interpret as details on what is being shown are not provided. Linear trends inferred from the Mann-Kendall test should yield only one slope value for a period of record, so it is unclear what the time series in Figures 5 and 7 denote. What do the two lines "UF" and "UB" represent, the figure caption does not state what these are.

Reply: We have found snow depth data to be normally distributed, the ordinary linear regression can be used to analyze the trend in time series. Therefore, we decide not to use the MK test (non-parametric test) again. Fig.5 and Fig.7 have been deleted.

**List of all relevant changes**

(1) P1, L.4: deleted "4" in superscript of the first author; replaced "6" with "4" in superscript of the third author; inserted a new author "Kang Wang[5]" as the fourth author; replaced "5" with "6" in superscript of the fourth author.

(2) P1, L.6: inserted "Key Laboratory of Remote Sensing of Gansu Province," before "Cold".

(3) P1, L.12: replace the fourth affiliation with the sixth affiliation.

(4) P1, L.13: inserted a new affiliation "[5] Institute of Arctic and Alpine Research, University of Colorado Boulder, Boulder, Colorado, 80309, USA"

(5) P1, L.13: replaced "5" with "6" in superscript of the fifth affiliation.

(6) P1, L.14: deleted the sixth affiliation.

(7) P1, L.19: deleted "the".

(8) P1, L.20: deleted "regional-and continental-scale".

(9) P1, L.21: inserted "from local community to regional industrial water supply" after "resources"; inserted new sentences "Data and knowledge on snow in general and snow depth/snow water equivalent in particular are prerequisites for climate change studies and local/regional development planning. Past studies by using in-situ data are mostly site-specific, while data from satellite remote sensing may cover a large area or in global scale, uncertainties are huge, evening misleading." before "In this study,"; deleted "a snow depth climatology and its"; replaced "variations were" with "change and variability in snow depth was".

(10) P1, L.22: deleted "the".

(11) P1, L.24: replaced "northeastern" with "north-eastern".

(12) P1, L.25: replaced "during 1966-2012" with "from 1966 through 2012".

(13) P1, L.26-27: deleted the comma; replaced "that period of time" with "the study period".

(14) P1, L.27: deleted "the"; deleted the space between the degree sign and North.

(15) P2, L.1: replaced "provides" with "provided".

(16) P2, L.2: replaced "changes in snow depth" with "snow depth climatology and changes"; replaced "are" with "were".

(17) P3, L.3: inserted ", including snow depth and snow area extent," after "cover"; inserted "an" after "as"; replaced "indicators" with "indicator".

(18) P3, L.3: replaced "circulation" with "circulations".

(19) P3, L.4: replaced "its" with "their".

(20) P3, L.10-13: replaced the sentence with "Changes in snow depth could have dramatic impacts on weather and climate through surface energy balance (Sturm et al., 2001), soil temperature and frozen ground (Zhang, 2005), spring runoff, water supply, and human activity (AMAP, 2011)."

(21) P3, L.13: inserted "the" after "Although"; replaced "reduced" with "declined".

(22) P3, L.14: replaced "still increased" with "showed an increasing trend"; deleted "the".

(23) P3, L.15-18: replaced the sentence by "This may be explained in that the warmer air led to a greater moisture supply for snowfall in winter (Ye et al., 1998; Kitaev et al., 2005; Rawlins et al., 2010)."

(24) P3, L.20: replaced "promoted" with "promotes".

(25) P3, L.20: inserted "local and regional" after "increased".

(26) P3, L.21: replaced "in-situ" with "in situ".

(27) P3, L.22: deleted "data".

(28) P3, L.23: deleted the comma; replaced "demonstrating" with "and demonstrated"; replaced "varies regionally:" with "varied differently over different regions."; deleted "overall, the"

(29) P3, L.24: replaced "annual" with "Annual".

(30) P3, L.25: deleted the comma before "and".

(31) P3, L.26: replaced "recent" with "last".

(32) P3, L.27: replaced "but there was" with "and showed large".

(33) P4, L.1: replaced "thus" with "and thus,".

(34) P4, L.2: inserted "the" before "snow".

(35) P4, L.3: replaced "are" with "were"; deleted "also".

(36) P4, L.5: replaced "is" with "was".

(37) P4, L.6: deleted "also"; replaced "other large" with "synoptic"; deleted the comma.

(38) P4, L.7: deleted the space after "Oscillation"; deleted "indices".

(39) P4, L.8: deleted the first "the".

(40) P4, L.10: replaced "is" with "was".

(41) P4, L.11: inserted "of Russia" after "Plain"; deleted "during the period".

(42) P4, L.12: deleted "the"; replaced "is" with "was".

(43) P4, L.13: replaced "indicated" with "demonstrated"; replaced "is" with "was".

(44) P4, L.14: inserted "between snow depth and" before "Niño-3".

(45) P4, L.15: replaced "in" with "on".

(46) P4, L.20: deleted "have"; inserted "/or" after "and"; deleted "the algorithms with", inserted "snow algorithms" after "microwave".

(47) P4, L.21-22: replaces "these observations" with "snow depth and snow water equivalent obtained by satellite remote sensing".

(48) P4, L.22: replaced "can" with "could"; deleted "the"; replaced "in-situ" with "the in situ".

(49) P4, L.23: replaced "the satellite data" with "they"; inserted a comma after "(25×25)".

(50) P4, L.24: replaced "inversion" with "perfect"; replaced the semicolon with the period; inserted the sentences "Using ground-based snow depth measurements across the Eurasian continent against snow depth obtained from passive microwave satellite remote sensing, Zheng et al. (2015) found that the mean percentage error was greater than 50% and can be up to approximately 200%. Utilization of snow depth obtained from satellite remote sensing has large uncertainties and is impractical." before "in"; replaced "in" with "In".

(51) P4, L.25: deleted "the".

(52) P4, L.26: deleted "always".

(53) P4, L.26-27: replaced "some of them need to obtain official permission before using in some countries" with "strict data use limitations apply".

(54) P4, L.27-30: replaced the sentence with "Ground-based measurements provide currently available and accurate snow depth over long time-series, which are critical data and information for investigating snow depth climatology and variability."

(55) P5, L.1: replaced "nearly" with "approximately".

(56) P5, L.2: inserted "the" after "over"; replaced "lands" with "land surfaces".

(57) P5, L.4-5: deleted "and large-scale" and "cover", deleted the comma.

(58) P5, L.6: replaced "cover" with "depth".

(59) P5, L.8-11: replaced the sentence with "Many studies on snow depth have focused on local and regional-scales over Russia (Ye et al., 1998; Kitaev et al., 2005; Bulygina et al., 2009, 2011; Brasnett, 1999) and on the TP (Li and Mi, 1983; Ma and Qin, 2012)."; deleted "However, due to the lack of data and information,"

(60) P5, L.13: replaced "is" with "was".

(61) P5, L.14: deleted "the"; deleted ", and analyze snow depth relationships with the topography and climate factors".

(62) P5, L.16: inserted a sentence "In addition, we analysed the spatial and temporal changes in snow depth with topography and climate factors over the study area." After "2012."

(63) P5, L.16-17: deleted the sentence "This study can provide basic information on climate system changes in the region."; deleted the comma.

(64) P5, L.21: inserted a sentence "The data used in this study include daily snow depth, snow water equivalent (SWE), air temperature and precipitation." before the first sentence.

(65) P5, L.22-23: deleted the sentence "Snow depth was measured at these stations on a daily basis", inserted the sentences "Snow depth was measured once a day at meteorological stations using a graduated stake installed at a fixed point location within the station or by a wooden ruler. Snow depth was measured using the same method across the Eurasian continent since the meteorological observation standard was established by the former USSR and followed by all of the former

USSR republics, Mongolia and China. Snow depth is one of the standard elements measured on a daily basis (WMO, 1996)." before "Historical".

(66) P5, L.26: inserted "the" before "during"; moved "period" to the back of "snowmelt".

(67) P5, L.28: deleted the comma.

(68) P5, L.30: replaced the first sentence with "SWE is an important parameter that is often used in water resource evaluation and hydroclimate studies."

(69) P6, L.1-3: deleted the sentence.

(70) P6, L.4: inserted "using a snow tube" after "measured".

(71) P6, L.5: inserted the sentences "Daily air temperature was measured using a thermometer, which was placed at a height of 1.5 m above the ground surface in an instrument shelter at the meteorological station (WMO, 1996). The air temperature measurement should be accurate to 0.1 ℃. Air temperature was measured four times a day at 0200, 0800, 1400, and 2000 local time. The daily mean air temperature was calculated by a simple arithmetic average of the four measurements, whereas the monthly mean was based on the daily mean and the annual mean was based on the monthly mean. Precipitation was gathered and measured by a precipitation gauge and was reported with a 0.1-mm precision (Groisman and Rankova, 2001). The original precipitation data were not corrected by considering the gauge undercatch." before "Daily" as a new paragraph.

(72) P6, L.6: deleted the comma.

(73) P6, L.7: deleted "the following Equation (1):".

(74) P6, L.13: inserted "a" before "quality"; deleted "the".

(75) P6, L.14: inserted "automatically" after "was".

(76) P6, L.14: inserted "and the National Meteorological Information Center (NMIC) of China Meteorological Administration (Ma and Qin, 2012)" after "(Veselov, 2002)".

(77) P6, L.16-24: replaced the sentences with "We implemented additional quality control using the following requirements: (1) To ensure snow depth stability, at a

given location, a month with less than 15 days of snow depth measurements was deleted; (2) Stations with sudden and steep changes in snow depth were eliminated from the list; (3) Stations with less than 20 years of data during the 1971-2000 period were excluded from the analysis; and (4) At each station, we eliminated data points that exceeded two standard deviations from their long-term (1971-2000) mean. After these four steps of snow depth quality control, we used data from 1814 stations to investigate the climatology and variability of snow depth over the Eurasian continent (Fig. 1 and Table 1)."

(78) P6, L.25-27: replaced the sentence with "We defined a snow year starting from July 1$^{st}$ through June 30$^{th}$ of the following year to capture the entire seasonal snow cycle."

(79) P6, L.27-P7, L3: replaced the sentences with "Procedures and techniques for measuring snow depth may have changed over the course of station history. Consequently, snow depth data may not be homogeneous in the time series over the period of the record. Fortunately, there was no change in the procedure and technique of snow depth measurements since 1965 in Russia and the other countries in this study (Bulygina et al., 2009). Therefore, in this study, we chose to use snow depth data from 1966 to 2012."

(80) P7, L5: replaced "In" with "in".

(81) P7, L6: replaced "way" with "method".

(82) P7, L7: replaced "in regular" with "based on"; inserted "the" before "World".

(83) P7, L10: replaced "the" with "an".

(84) P7, L11: replaced the period with the semicolon.

(85) P7, L.12-14: deleted the sentence.

(86) P7, L15: replaced the two "the" with "an".

(87) P7, L.17-18: replaced "from the annual snow depth for ≥20" with "for stations with more than 20"; inserted "the" after "during"; inserted "period" after "1966-2012"; replaced the period with the semicolon.

(88) P7, L19: replaced "the" with "an".

(89) P7, L21: replaced "values" with "value", inserted "the" before "annual"; deleted

the second "the".

(90) P7, L22: replaced "≥20" with "more than 20"; inserted "the" after "during"; inserted "period" after "1966-2012" ; replaced the period with the semicolon.

(91) P7, L.23-30: replaced the paragraph with "Anomalies of monthly, annual mean, and annual mean maximum snow depth from their long-term (1971-2000) records were calculated for each station across the Eurasian continent. Composite time series of monthly and annual anomalies were obtained by using all of the available station data across the study area."

(92) P8, L.2: replaced "of" with "in"; inserted "entire" before "study"; deleted "as a whole".

(93) P8, L.7: replaced "happened" with "occurred"; inserted "A" before "linear".

(94) P8, L.8: deleted "Despite there is a nonlinearity,"

(95) P8, L.9: deleted "a" after "when"; replaced "emerged" with "emerge"; inserted "even though there is a nonlinearity" after "emerge", deleted the period.

(96) P8, L.11: replaced "The Student T test" with "The Student's t-test".

(97) P8, L.12: deleted the first "the"; replaced "significant" with "significance".

(98) P8, L.13: replaced the second "the" with "a".

(99) P8, L.14: inserted "significant" after "considered"; inserted the sentences "The Durbin-Watson test was used to detect serial correlation of data in the time series, and the Cochrane-Orcutt test was used to correct the serial correlation. Then, the serial correlations of the new data were rechecked and recalculated trends in the time series of the new data. The methods and test results were described in the appendix." after "in our study."

(100) P8, L.14-23: deleted the sentences.

(101) P8, L.27-29: replaced the sentence with "Distributions of long-term mean snow depth indicated a strong latitudinal zonality. Generally, snow depth increased with latitude northward across the Eurasian continent (Fig. 2)."; inserted a space before "A".

(102) P8, L.30: deleted "in the".

(103) P9, L.1: deleted "of the".

(104)  P9, L.2: inserted "the" after "of"; replaced "gray" with "grey".

(105)  P9, L.3: replaced "Depths" with "Snow depths".

(106)  P9, L.4: replaced "northeastern" with "north-eastern".

(107)  P9, L.5: deleted the second comma; replaced "The regions" with "Regions".

(108)  P9, L.10: deleted the comma.

(109)  P9, L.11: replaced "north" with "northern part".

(110)  P9, L.12: inserted the comma after "China".

(111)  P9, L.13: replaced "northeastern" with "north-eastern".

(112)  P9, L.15-16: replaced the sentence with "Annual mean maximum snow depth (Fig. 2b) showed a similar spatial distribution pattern compared to the annual mean snow depth pattern."

(113)  P9, L.17-18: replaced the sentence with "The maximum value was approximately 201.8 cm in snow depth."

(114)  P9, L.19: deleted "the".

(115)  P9, L.20: deleted "located"; replaced "northeastern" with "north-eastern".

(116)  P9, L.24: inserted "the" after "of".

(117)  P9, L.25: deleted the comma; inserted "decreased to" before "6-10"; replaced "in the" with "when moving south to"; replaced "parts of the country" with "Mongolia".

(118)  P9, L.26-29: replaced the sentences with "Maximum snow depths were higher over the northern part of the Xinjiang Autonomous Region of China, Northeast China, and eastern and southwestern TP, were mostly greater than 10 cm and even greater than 20 cm in some areas. For the remaining regions of China, the maximum snow depths were relatively small and mostly less than 10 cm."

(119)  P9, L.30- P10, L.3: deleted the paragraph.

(120)  P10, L.6: deleted "the" and the comma.

(121)  P10, L.7: inserted "Moving southward," before "monthly"; replaced "Monthly" with "the monthly".

(122)  P10, L.8: deleted "in the"; deleted "most regions".

(123)  P10, L.9: replaced "areas covered by snow" with "snow cover extent".

(124)  P10, L.10: replaced "Most monthly" with "Monthly".

(125)  P10, L.11: replaced "in most regions" with "for the majority".

(126)  P10, L.12-13: replaced with "except the northern Xinjiang Autonomous Region of China, Northeast China, and south-western TP where snow depth exceeded 10 cm."

(127)  P10, L.14-15: replaced the sentence with "In spring (March through May), snow cover areas decreased significantly (Figs. 3g–i), which was mainly because of snow disappearance in the majority of China."

(128)  P10, L.18: inserted "in" before "the".

(129)  P10, L.21: inserted "both" after "in"; deleted "the"; inserted "snow depth" after "mean".

(130)  P10, L.22: inserted the period after "continent"; deleted "as a whole with".

(131)  P10, L.23-24: replaced the sentence with "Mean annual snow depth increased at a rate of approximately 0.2 cm decade$^{-1}$, whereas annual mean maximum snow depth increased at a rate of approximately 0.6 cm decade$^{-1}$ (Fig. 4)."

(132)  P10, L.26: deleted "the"; replaced "about" with "approximately".

(133)  P10, L.27: replaced "about" with "approximately".

(134)  P10, L.28: deleted the first "the"; deleted the second comma; deleted "it".

(135)  P10, L.29: deleted the comma.

(136)  P11, L.1: replaced "3.5" with "approximately 3 to 4".

(137)  P11, L.1-2: deleted the comma; replaced with "and then there was a large fluctuation without a significant trend from the late 1970s to the early 1990s."

(138)  P11, L.4-13: deleted the paragraph.

(139)  P11, L.14- P12, L.7: replaced the paragraphs with "Monthly snow depth changed significantly across the Eurasian continent from 1966 through 2012 (Fig. 5). Snow depth decreased in October at a rate of approximately -0.1 cm decade$^{-1}$ (Fig. 5a), and there were no significant trends in November and December with large inter-annual variations (Fig. 5b-c). From January through April, snow depth showed statistically increased trends with rates between 0.3 cm decade$^{-1}$ and 0.6

cm decade$^{-1}$ (Fig. 5d-g). Overall, snow depth decreased or there was no change in autumn and increased in winter and spring with large inter-annual variations over the study period."

(140)    P12, L.8-22: deleted the paragraph.

(141)    P12, L.23: replaced "Figure 8" with "Figure 6".

(142)    P12, L.24: deleted the comma.

(143)    P12, L.26: deleted "most of" and the first "the".

(144)    P12, L.27: replaced "Fig. 8a" with "Fig. 6a".

(145)    P12, L.29: deleted the first "the"; inserted "the" before "Russian".

(146)    P12, L.30: replaced "across" with "in".

(147)    P12, L.30 and P13, L1: deleted the space between the degree sign and North, replaced "the region" with "regions".

(148)    P13, L.2: replaced "indicating" with "which indicated".

(149)    P13, L.4: deleted "the".

(150)    P13, L.5: replaced "Eurasian areas" with "Eurasia"; replaced "but the change rates of the maximum snow depth" with "but the magnitude of changing rates in the maximum snow depth".

(151)    P13, L.7: replaced "8b" with "6b"; replaced "The significant" with "Significant".

(152)    P13, L.8-10: replaced the sentence with "Generally, the decreasing trends were found in the same regions where annual mean snow depth decreased and there were greater reductions in southern Siberia and the Far East."

(153)    P13, L.11: replaced "changes" with "increasing trends"; replaced "at the 95 % level" with "P≤0.05".

(154)    P13, L.12: replaced "Figs. 9a, b" with "Fig. 7a and b".

(155)    P13, L.13: inserted "although the magnitudes were generally small" after "October".

(156)    P13, L.13-17: replaced the sentences with "Over November, the increasing trends in snow depth only appeared in Siberia and the Russian Far East, whereas decreasing trends occurred in monthly mean snow depth over eastern European

Russia, the southern West Siberian Plain, and the northeast Russian Far East."

(157) P13, L.18-24: replaced the paragraph with "In winter months (December-February), there was a gradual expansion in areas with increasing trends in monthly mean snow depth variation with P≤0.05 (Figs. 7c–e), and this mainly occurred in eastern European Russia, southern Siberia, the northern Xinjiang Autonomous Region of China, and Northeast China. In contrast, significant decreasing trends were observed in northern and western European Russia and were scattered in Siberia, the northeast Russian Far East, and northern China."

(158) P13, L.25-26: replaced "at the 95 % level" with "P≤0.05".

(159) P13, L.27: replaced "Figs. 9f-h" with "Figs. 7f-h".

(160) P13, L.28: inserted the comma after "USSR".

(161) P13, L.30: replaced "of" with "in"; inserted "stations" after "these".

(162) P14, L.1: replaced "in" with "at".

(163) P14, L.1-2: replaced the sentence with "Compared with regions south of 50 °N, changes in monthly mean snow depth were more significant over regions north of 50 °N."

(164) P14, L.5: inserted the sentence "Topography is an important factor affecting the climatology of snow depth and is the main reason accounting for the inhomogeneity of data (Grünewald and Lehning, 2011, 2013; Grünewald et al., 2014)." before the first sentence.

(165) P14, L.6: inserted "the" before "annual".

(166) P14, L.7: replaced "10" with "8"; replaced "is" with "was".

(167) P14, L.8: replaced "increases" with "increased"; replaced "10a" with "8a"; replaced "increase" with "increased"; replaced "about" with "approximately".

(168) P14, L.9: deleted the space between the degree sign and North; inserted "across the Eurasian continent" after "1 °N".

(169) P14, L.9-13: replaced the sentences with "A closer relationship between latitude and snow depth was found in regions north of 40 °N (Figs. 8a and d) where snow cover was relatively stable with the number of annual mean

continuous snow cover days at more than 30 (Zhang and Zhong, 2014)."

(170)    P14, L.15: replaced "(Fig. 10b)" with "(Fig. 8b);".

(171)    P14, L.17: deleted the second comma.

(172)    P14, L.18: inserted "a" after "at".

(173)    P14, L.19: replaced "the" with "a".

(174)    P14, L.20: replaced "determined" with "found".

(175)    P14, L.21: deleted the space between the degree sign and North, replaced "10d" with "8d".

(176)    P14, L.22-23: deleted the sentence.

(177)    P14, L.24: inserted "statistically" before "significant".

(178)    P14, L.25: inserted "over the Eurasian continent" after "continentality"; inserted "P≤0.05," before "Fig.", replaced "10c" with "8c".

(179)    P14, L.26: replaced "is" with "may be".

(180)    P14, L.26-27: replaced "snow cover climatology" with "snow depth distribution".

(181)    P14, L.27: inserted "especially on the TP" after "Eurasia", deleted "though it will determine the snowfall rate", and inserted the sentences "Although the previous studies showed that the Tibetan Plateau's largest snow accumulation occurred in the winter, the precipitation during the winter months was the smallest of the year (Ma, 2008). This was mainly due to the majority of annual precipitation that occurs during the summer monsoon season on the TP, which causes much less precipitation during the winter half year (or the snow accumulated season)." after the last sentence.

(182)    P15, L.3: inserted "former" before "USSR"; replaced "Fig. 11" with "Fig. 9".

(183)    P15, L.3-5: replaced the sentence with "The period (snow cover years) spanned from 1966 through 2009 using available data."

(184)    P15, L.6: deleted the comma.

(185)    P15, L.7: replaced "11a" with "9a", deleted "the".

(186)    P15, L.8: replaced "better" with "strong"; replaced "11b" with "9b".

(187)    P15, L.9: replaced "in" with "at"; replaced "the" with "an"; replaced "being"

with "of".

(188)    P15, L.10-11: replaced sentence with "Snow depth increased with an increase in accumulated snowfall, and the thickest snow depth of approximately 120 cm had a maximum cumulative snowfall of approximately 350 mm."

(189)    P15, L.12: replaced "Comparing" with "Compared with"; replaced "of changes" with "in change".

(190)    P15, L.13: replaced "variability of" with "variabilities in".

(191)    P15, L.16: replaced "Fig. 12" with "Fig. 10".

(192)    P15, L.18: inserted "a" after "to"; replaced the comma with "and".

(193)    P15, L.19: replaced "The significant" with "Significant".

(194)    P15, L.20: deleted the second "the".

(195)    P15, L.22: replaced "lowed" with "lowered"; replaced "whole" with "entire"; inserted "the" before "snowpack".

(196)    P15, L.23: replaced "of" with "in"

(197)    P15, L.24: replaced "12b-d" with "10b-d".

(198)    P15, L.26: deleted "the".

(199)    P15, L.27: deleted the comma; replaced "as well as" with "and".

(200)    P15, L.29: replaced "Fig. 13" with "Fig. 11"; replaced "The" with "A".

(201)    P15, L.30: replaced "presented" with "was present"; deleted "the".

(202)    P16, L.1: replaced "13a" with "11a".

(203)    P16, L.2: replaced "correlation" with "correlations"; deleted "the".

(204)    P16, L.3: replaced "13b" with "11b"; replaced "It" with "This"; inserted "there was no obvious effect of increasing temperature on snow depth when" after "because"; inserted "which occurred" after "0 ℃".

(205)    P16, L.4-5: replaced "during" with "from"; deleted ", the increasing temperature did not have an obvious effect on snow depth".

(206)    P16, L.9: deleted the first two "the"; inserted "and" before the third "the".

(207)    P16, L.11: inserted "the" after "and".

(208)    P16, L.14: deleted "4.1 Comparison with Previous Results".

(209)    P16, L.15-25: replaced the paragraph with "Studies on changes in snow

[revised manuscript text omitted]

(219)    P17, L.16- P18, L6: deleted the paragraph.

(220)    P18, L.13: replaced "northeastern" with "north-eastern".

(221)    P18, L.16: inserted "the" after "of".

(222)    P18, L.17: replaced "of" with "in".

(223)    P18, L.18: inserted "entire" before "Eurasian"; deleted "as a whole"; replaced "increase" with "increasing".

(224)  P18, L.21: inserted "a" after "presented".

(225)  P18, L.22: replaced "while" with "whereas"; replaced the two "of" with "in"; inserted "the" before "variations".

(226)  P18, L.25: deleted the first "the"; inserted "the" before "northern".

(227)  P18, L.26: replaced "northeastern" with "north-eastern".

(228)  P18, L.28: replaced "northeastern" with "north-eastern".

(229)  P18, L.30: deleted "the".

(230)  P19, L.1: replaced "of" with "in".

(231)  P19, L.2: deleted the two "the".

(232)  P19, L.3: replaced the comma with "and".

(233)  P19, L.4: replaced "driver" with "driving".

(234)  P19, L.6: inserted an appendix:

"**Appendix A:** Analysis of serial correlation

[revised manuscript text omitted]

(257)    P26: replaced figure 1 with a new figure

(258) P27: replaced figure 2 with a new figure

(259)    P28-29: replaced figure 3 with a new figure

(260)    P30: deleted figure 5.

(261)    P32: replaced "Figure 6" with "Figure 5".

(262)    P34: deleted figure 7.

(263)    P35: replaced figure 8 with a new figure, replaced "Figure 8" with "Figure
         6".

[Figure]

(264)    P36-37: replaced figure 9 with a new figure, replaced "Figure 9" with "Figure 7", replaced "Figure 10" with "Figure 8".

[Figure]

[Figure]

● -1 - -0.7 ● -0.7 - -0.5 ● -0.5 - -0.3 ● -0.3 - -0.1 ∙ -0.1 - 0 · 0 - 0.1 ● 0.1 - 0.3 ● 0.3 - 0.5 ● 0.5 - 0.7 ● 0.7 - 1

(265)  P38: replaced "Figure 11" with "Figure 9".

(266)  P39: replaced "Figure 12" with "Figure 10".

(267)  P40: replaced figure 13 with a new figure, replaced "Figure 13" with "Figure 11", replaced "confident level" with "confidence level".

[Figure]

● -1 - -0.8 ● -0.8 - -0.6 ● -0.6 - -0.4 ● -0.4 - -0.2 · -0.2 - 0 · 0 - 0.2 ● 0.2 - 0.4 ● 0.4 - 0.6 ● 0.6 - 0.8 ● 0.8 - 1

(268)  P40, L.6: inserted a new figure as Figure A1.

[revised manuscript text omitted]

---

## Author Response (AR4)

**Response to Referee #1**

This revised paper addresses in a satisfactory manner the issues raised in my previous reviews of the article. However, there are some minor technical issues that remain to be resolved as outlined in my report. I also note that the list of co-authors and their order has changed yet again, and so the authors must explain this change of co-authorship on their paper.

Reply: the list of co-authors is ordered by the individual contribution to the article: Xinyue Zhong, Tingjun Zhang, and Shichang Kang designed the article structure. Kang Wang performed the analysis of serial correlation. Lei Zheng and Yuantao Hu conducted snow depth data analysis. Huijuan Wang analyzed interannual trend in snowfall. Xinyue Zhong prepared the manuscript with contributions from all co-authors.

**General Comments:**

1. P. 4, second paragraph: Apart from remote sensing, numerical modeling is often used to obtain accurate and spatially-complete fields of snow depth and/ or snow water equivalent (SWE) (e.g., Liston and Hiemstra, 2011). Is there any reason why model simulations of snow depth and SWE are not mentioned in this paragraph, as they form another important source of cryospheric information in data sparse regions such as northern Eurasia?

Reply: Thank you very much for your comments. We have added the statement of numerical modeling and spatial interpolation data, and pointed out their merits and weaknesses.

 "Apart from remote sensing, numerical modeling is often used to obtain accurate and spatially-complete fields of snow depth and/ or snow water equivalent (SWE) (Liston and Hiemstra, 2011; Terzago et al., 2014; Wei and Dong, 2015). However, remote sensing data with coarse-scale measurement is an important input parameter that affects simulation accuracy and does not provide a sufficient time series length. Spatial interpolation is a common method for estimates in areas with devoid data. However, uncertainty and potential bias in spatial interpolation can be introduced due to specific algorithms especially in complex terrain areas."

2. P. 20, Appendix A: I appreciate the authors' consideration of the potential effects of serial correlation on their trend analyses. However, rather than the elaborate Durbin-Watson test, did the authors look simply at the lag 1 auto-regression (AR1) to examine if serial correlation was indeed present in their time series? How would those results compare to those obtained from the Durbin-Watson test?

Reply: the Durbin-Watson test just check serial correlation at the lag 1 auto-regression. AR1 formula is $Y = aX + e_t$, where $e_t$ is the noise term or residue. Examining $e_t$ just to check if serial correlation is present in AR1. Therefore, the Durbin-Watson test

is used to check the residue and then test serial correlation.

**Specific Comments:**

1. P. 1, line 25: Replace "are huge" with "remain large" and replace "evening" with "even".

Reply: Has been done.

2. P. 2, line 6: Change the verb tense to the present, i.e. "provides".

Reply: Has been changed.

3. P. 3, line 12: Insert "the" before "surface".

Reply: Has been inserted.

4. P. 4, line 8: Insert "the" before "NAO".

Reply: Has been inserted.

5. P. 4, line 9: Revise to: "fluctuations of snowfall amounts and snow depth".

Reply: We have deleted the sentence.

6. P. 4, line 12: Change to: "however, the NAO index was…"

Reply: Has been changed.

7. P. 4, line 25: Clouds do not interfere with microwave remote sensing of SWE, so this statement is misleading.

Reply: We have deleted clouds.

8. P. 4, line 26: What are "perfect algorithms"? Is there such a thing?

Reply: We have replaced "perfect algorithms" with "algorithms".

9. P. 5, line 18: Change to: "to develop a climatology and investigate the variability"

Reply: Has been changed.

10. P. 6, line 16: Why the tentative language in this sentence? The air temperature measurements either have or do not have accuracy of 0.1 ℃. If not, then specify the

exact accuracy of those measurements.

Reply: We have replaced "should be" with "was".

11. P. 6, line 26, Equation (1): Do not italicize the units of ℃.

Reply: Has been changed.

12. P. 8, line 6: Insert a space in "than 20".

Reply: Has been inserted.

13. P. 8, line 10: Replace semi-colon by a period at the end of the sentence.

Reply: Has been changed.

14. P. 10, lines 6/7: The statement starting with "were mostly…" is incomplete – please rephrase.

Reply: We have inserted "which" before "were mostly".

15. P. 11, line 16: Replace "increased" with "increasing".

Reply: Has been done.

16. P. 12, lines 1-7: There's much repetition of ideas and text in this paragraph – please review and edit carefully.

Reply: We have deleted this paragraph.

17. P. 13, line 23: Delete "the" before "previous".

Reply: Has been deleted.

18. P. 13, line 24: Elsewhere, the Tibetan Plateau is abbreviated as "TP" but not here.

Reply: Has been abbreviated.

19. P. 13, line 24: Delete "the" before "winter" and "the" before "precipitation".

Reply: Has been deleted.

20. P. 13, line 25: Delete "the" before "winter".

Reply: Has been deleted.

21. P. 13, line 28: Replace "accumulated" with "accumulation".

Reply: Has been done.

22. P. 15, line 6: Insert a comma after "0°C,".
Reply: Has been inserted.

23. P. 15, line 23: Insert "a" before "different".

Reply: Has been inserted.

24. P. 16, line 5: "station" should be singular. Insert "a" before "critical".

Reply: Has been done.

25. P. 16, lines 11-14: The journal may prefer superscripts for all units, i.e. "cm yr-1".

Reply: Has been done.

26. P. 16, line 13: Replace the comma after "Russia" with a semi-colon.

Reply: Has been replaced.

27. P. 17, line 4: Insert "Phase 5" after "Project".

Reply: Has been inserted.

28. P. 17, line 5: Here and elsewhere, the long name for the Tibetan Plateau is used again.

Reply: Has been modified.

29. P. 17, lines 5-6: Delete "the" before "forested regions".

Reply: Has been deleted.

30. P. 17, line 24: The sentence starting with "Spring floods" is incomplete – please rephrase.

Reply: We have rephrased the sentence:

"Spring floods are generated by melting snow, and freshwater derives are from snowmelt in some snow-dominated basins (Barnett et al., 2005)."

31. P. 17, line 29: Change to "forests".

Reply: Has been changed.

32. P. 17, line 30: Change to "plant".

Reply: Has been changed.

33. P. 20, line 22: Replace "anomalies" with "anomaly".

Reply: Has been replaced.

34. P. 20, line 24: Change to "its estimate value r:"

Reply: Has been changed.

35. P. 21, line 7: Insert "at" before "approximately".

Reply: Has been inserted.

36. P. 21, line 14: See previous comment about the format for units.

Reply: Has been changed.

37. P. 25, line 28: Note spelling mistake in "surface".

Reply: Has been modified.

38. P. 26, lines 20-22: Please update with the appropriate volume and page numbers.

Reply: Has been updated.

39. P. 29, line 6, Table A2: Insert "the" before "Dikson".

Reply: Has been inserted.

40. P. 31, lines 3-5, Figure 1: Replace "triangles" with "circles". The figure caption should explain all abbreviations used for the names of countries on the map. What does the inset map on the bottom right show? Is this inset map shown here and on Figures 2, 3, 6, and 7 needed, as no results are shown on these?

Reply: We have replaced "triangles" with "circles" and explained all abbreviations. The inset map shows the Chinese territory that cannot be displayed in the large map. We think it is necessary to represent.

41. P. 36, Figure 5: The caption needs to explain why linear regressions are shown only on a few panels for this plot.

Reply: We have added the explanation:

"Linear regression was only shown when the rate of change was at the 95% level."

42. P. 42, Figure 11: The caption should specify that the results in this plot cover only Russia/former USSR.

Reply: We have added:

"Figure 11. Spatial distributions of partial correlation coefficients of snow depth and air temperature (a), snow depth and snowfall (b), SWE and air temperature (c), SWE and snowfall from November through March during 1966-2009 across the former USSR."

**Response to Referee #3**

**General Comments:**

1. Lack of guiding science questions/hypotheses: A fundamental weakness of the paper is related to the lack of clear science questions guiding the analysis which results in a descriptive level analysis without any particularly interesting or relevant conclusions that help advance understanding of key questions such as: Is Eurasian fall snow cover increasing as shown in the NOAA-CDR dataset (e.g. Cohen et al. 2012) and subsequently disputed by Brown and Derksen (2013) and Mudry et al. (2017)? Is there evidence of an accelerating hydrologic cycle (e.g. Syed et al. 2010) in the snow cover data? Do climate models underestimate snow cover temperature sensitivity (e.g. Mudryk et al. 2017) or is this an artifact of the NOAA-CDR dataset? Are precipitation trends consistent with observed changes in snow depth?

Reply: Thank you very much for your detailed comments and concerns. There are many

studies focus on the variations in snow depth over Eurasia, but some problems still exist in those studies: 1) Research scale. Most of studies on snow depth have focused on local and regional scales over Russia and on the Tibetan Plateau, however, there are few information of snow depth at continental scale over Eurasia. 2) Data. Data from in situ, remote sensing, numerical modeling and interpolation are used to investigate variations in snow depth. These data have their advantages and disadvantages. Although snow depth and snow water equivalent obtained by satellite remote sensing could mitigate regional deficiency of the in situ snow depth observations, they have low spatial resolution (25×25 km), and the accuracy is always affected by underlying surface conditions and algorithms. Numerical modeling is often used to obtain accurate and spatially-complete fields of snow depth and/ or snow water equivalent (SWE). However, remote sensing data with coarse-scale measurement is an important input parameter that affects simulation accuracy, however, it does not provide a sufficient time series length. Spatial interpolation is a common method for estimates in areas with devoid data. Uncertainty and potential bias in spatial interpolation can be introduced due to specific algorithms especially in complex terrain areas. Although the number of ground-based observation sites is limited, it can provide currently available and accurate snow depth over long time-series, and provide the data base for the verifications of remote sensing and model simulation.

Based on the above problems, we develop a climatology and investigate the variability of snow depth over Eurasia, and discuss the impacts of topography and climate factors on snow accumulation, and the potential effects of variations in snow depth. The results showed that snow accumulation increased significantly when snow cover extent and snow cover duration decreased in response to climate change over Eurasia. Interannual trend in snow depth is consistent with snowfall and heavy snowfall. This indicates that extreme snowfall events may be the main cause of the increase in snow depth.

The authors chose to analyse snow depth and snow cover in two separate papers which is a strategic error in my opinion. Understanding snow cover variability requires at least four essential snow cover variables: the start/end date of snow cover, and the date and depth of the annual maximum accumulation, together with information on rainfall, snowfall and temperature. For example, this paper shows increasing snow depths over polar latitudes occurring with a shortened snow cover season (the other paper). The only way this can happen is from more intense snowfall during the shorter accumulation period. Is this hypothesis supported by the precipitation data? Analysis of the melt period (SnowOff Date - SDmax date) could also provide insights into melt dynamics and possibly additional evidence of an accelerating hydrologic cycle. Separating snow depth and snow cover precludes examining these kinds of questions.

Reply: The focus of the two paper are different. This paper mainly studies the spatiotemporal change and variability in snow depth and its influencing factors at continental scale. We adds some analysis of changes in snow depth in the other paper.

2. Paper organization and language: The organization of the paper suffers because the authors do not have a clear storyline (i.e. science questions) to build on. This results in the inclusion of often irrelevant material in the introduction, and overly descriptive material in the results section. I urge the authors to look at examples of published papers in journals such a GRL or JGR to see how the papers are structured. Issues with the English language become relatively minor if the paper has a solid science foundation.

Reply: We have modified Introduction and Discussion:

[revised manuscript text omitted]

3. Methodological issues: The paper contains a number of methodological issues that may have implications for some of the study conclusions:

- The first is the 20-year minimum years of data requirement for a station to be included in the regional average for 1966-2012. This has the potential to generate a temporally varying network of stations that can have a major impact on the trend analysis results of the regional average. The authors should provide a time series plot of the number of stations included in the Eurasia regional average each year to verify that a relatively even spatial distribution of stations is maintained over the full 47 years. You can test the robustness of the regional average to varying minimum data length for a range of years e.g. 20, 30, 40, 50.

Reply: the 20-year minimum years of data requirement for a station not to be included in the regional average for 1966-2012, but for 1971-2000. The World Meteorological Organization common approach to calculate anomalies is based on a 30-years climate normal period. In our study, 1971-2000 was used as the normal period. To ensure data continuity, according to probability and mathematical statistics, the station with more than two-thirds of 30-years data during 1971 to 2000 was used to analyze.

- A related issue is the generation of a regional average from all stations which gives a result that is weighted toward the region with the densest observing network (i.e. west of ~90E). Interpolation of station data to a grid would help avoid this potential bias.

Reply: Spatial interpolation is a common method for estimates in areas with devoid data. However, uncertainty and potential bias in spatial interpolation can be introduced due to specific algorithms especially in complex terrain areas. At present, there is no mature snow depth interpolation method in China, especially on the Tibetan-Plateau. Therefore, interpolation of station data to grid is not used in our study.

- Ignoring homogeneity and undercatch issues with the precipitation data (page 6, lines 22-23) may also have implications for the study conclusions. For example, with corrected precipitation data Groisman et al. (2014) found no evidence of increasing cold season precipitation over most of the Russian Federation, and significant decreases over the Arctic sector.

Reply: The snowfall data are estimated with air temperature and precipitation because we cannot possess the snowfall observations. And there are no wind speed or wind direction data can be obtained, therefore, we do not correct precipitation data to account for the bias in the long-term interannual precipitation time series.

- The analysis of elevation influence on snow depth (dSD/dZ) is contaminated by other influences such as climate region (e.g. dry interior regions are likely to have a different elevation response than mountains in maritime locations). One way to isolate dSD/dZ would be to use a moving spatial window.

Reply: Raster data can be used in the moving spatial window method. We should interpolate station data to a grid if we use this method. However, uncertainty and potential bias in interpolation can be introduced due to specific algorithms especially in complex terrain areas. At present, there is no mature snow depth interpolation method in China, especially on the Tibetan-Plateau. To avoid bias, we do not use moving spatial window. We have added more detail of the relationship between elevation and snow depth:

"Snow depths are averaged to 200 m elevation bands and then discussed the relation to elevation level for the former USSR and China. Snow depths were deeper in the lower elevation bands between 0 and 600 m across the former USSR (Fig. 8c). However, there were shallow snow accumulation between 600 and 1000 m due to most accumulation areas located in forest. Then snow depth was followed by a significant positive trend and reached a peak. Snow depths represented marked decrease in the highest elevation band (2600~2900 m). There were only two stations in this band and more snow accumulation difference between the two stations because of terrain and climate factors. Snow depths were deeper in three elevation bands across China: 200~1000 m, 1600~1800 m and 2400~2600 m. Greater snow accumulation were attributed to heavy

snowfall and severe cold in these regions. An increasing trend of snow depth presented in the higher elevations above 2600 m on the TP."

- The discussion of Liston and Hiemstra snow depth data on page 16-17 incorrectly states that it is an assimilation (it is a reconstruction with no observational input), and that SnowModel was driven with surface observations. The model was driven with downscaled MERRA reanalysis fields that do not incorporate surface observations.

Reply: In Liston and Hiemstra's study, they use two models to analyze the trends in snow cover. Precipitation, wind speed and direction, air temperature, and relative humidity obtained from meteorological stations and/or an atmospheric model located within near the simulation domain are inputs in SnowModel. MicroMet is a data assimilation and interpolation model that utilizes meteorological station datasets and/or gridded atmospheric model or (re)analyses datasets. Therefore, we state "the SnowModel input data included ground-based measured air temperature, precipitation, wind conditions and in part snow depth."

- The wavelet analysis only seems to have served as a low-pass filter for the regional averaged time series plots. What happened to the wavelet spectrum plot showing wavelet coefficients versus time like Figure 6 in De Jongh et al. (2006)?

Reply: Thank you very much for your suggestion. We have plotted the wavelet spectrum of snow depth and maximum snow depth. The results showed that there are no specific periodic cycle is dominant over the time series for average snow depth and maximum snow depth. The yearly cycle is about half a year from the mid-1960s though the late 1990s. A 1.5-year component process is represented in the period 1993-2010 for average snow depth and 2003-2010 for maximum snow depth.

[Figure]

[Figure]

Fig.1 The wavelet spectrum of snow depth during 1966 through 2010.

**List of all relevant changes**

(1) P1, L.10: replaced "Cryosphere" with "Cryospheric".

(2) P1, L.25: replaced "are huge" with "remain large", replaced "evening" with "even".

(3) P2, L.6: replaced "provided" with "provides".

(4) P3, L.2-3: deleted the first sentence.

(5) P3, L.7-10: moved this sentence to the first sentence of the paragraph.

(6) P3, L.11-14: moved this sentence to the last sentence of the first paragraph.

(7) P3, L.12: inserted "the" before "surface".

(8) P3, L.14-20: deleted those sentences.

(9) P3, L.21-P5, L.22: rewrote and reordered these paragraphs:

"During winter, the average maximum terrestrial snow cover is approximately 47 $\times 10^6$ km$^2$ over the Northern Hemisphere land surfaces (Robinson et al., 1993; IGOS, 2007). A large fraction of the Eurasian continent is covered by snow during the winter season, and some areas are covered by snow for more than half a year. There are long-term snow measurements and observations across the Eurasian continent with the first snow depth record dating back to 1881 in Latvia (Armstrong, 2001). These measurements provide valuable data and information for snow cover phenology and snow cover change detection. Many studies on snow depth have focused on local and regional scales over Russia (Ye et al., 1998; Kitaev et al., 2005; Bulygina et al., 2009, 2011; Brasnett, 1999) and on the Tibetan Plateau (TP) (Li and Mi, 1983; Ma and Qin, 2012), which have revealed the significant regional characteristics in the changes in snow depth. Annual mean snow depth has increased in northern Eurasia and the Arctic during the last 70 years (Ye et al., 1998; Kitaev et al., 2005; Callaghan et al., 2011a; Liston and Hiemstra, 2011) and showed large regional differences (Bulygina et al., 2009, 2011; Ma and Qin, 2012; Stuefer et al., 2013; Terzago et al., 2014). Changes in snow depth are primarily affected by air temperature and precipitation. Ye et al. (1998) and Kitaev et al. (2005) showed that higher air temperatures caused an

increase in snowfall in winter from 1936 through 1995, and thus, greater snow depth was observed in northern Eurasia in response to global warming. Furthermore, the snow depth distribution and variation are controlled by terrain (i.e., elevation, slope, aspect, and roughness) and vegetation (Lehning et al., 2011; Grünewald et al., 2014; Revuelto et al., 2014; Rees et al., 2014; Dickerson-Lange et al., 2015). Snow depth is closely related to synoptic-scale atmospheric circulation indices such as the North Atlantic Oscillation/Arctic Oscillation (NAO/AO). For example, Kitaev et al. (2002) reported that the NAO index was positively related to snow depth in the northern part of the East European Plain of Russia and over western Siberia from 1966 to 1990; however, the NAO index was negatively correlated with snow depth in most southern regions of northern Eurasia. You et al. (2011) demonstrated that there was a positive relationship between snow depth and the winter AO/NAO index and between snow depth and Niño-3 region sea surface temperature (SST) on the eastern and central TP from 1961 through 2005. However, most snow depth studies are at regional scale, information of snow depth at continental scale is required over the Eurasian continent.

To increase the spatial coverage of snow depth, researchers have used different instruments (e.g., LIDAR, airborne laser scanning (ALS), and unmanned aerial systems (UASs)) (Hopkinson et al., 2004; Grünewald et al., 2013; Bühler et al., 2016) or developed and/or improved passive microwave snow algorithms (Foster et al., 1997; Derksen et al., 2003; Grippaa et al., 2004; Che et al., 2016). Although snow depth and snow water equivalent obtained by satellite remote sensing could mitigate regional deficiency of the in situ snow depth observations, they have low spatial resolution (25×25 km), and the accuracy is always affected by underlying surface conditions and algorithms. Using ground-based snow depth measurements over the Eurasian continent against snow depth obtained from passive microwave satellite remote sensing, Zheng et al. (2015) found that the mean percentage error was greater than 50% and can be up to approximately 200%. Utilization of snow depth obtained from satellite remote sensing has large uncertainties and is impractical. Apart from remote sensing, numerical modeling is often used to obtain accurate and spatially-complete fields of snow depth and/ or snow water equivalent (SWE) (Liston

and Hiemstra, 2011; Terzago et al., 2014; Wei and Dong, 2015). However, remote sensing data with coarse-scale measurement is an important input parameter that affects simulation accuracy and does not provide a sufficient time series length. Spatial interpolation is a common method for estimates in areas with devoid data. However, uncertainty and potential bias in spatial interpolation can be introduced due to specific algorithms especially in complex terrain areas. In addition, data acquisition from large airborne equipment or aerial systems is costly and strict data use limitations apply. Ground-based measurements provide currently available and accurate snow depth over long time-series, which are critical data and information for investigating snow depth climatology and variability and can provide the data base for the verifications of remote sensing and model simulation.

The objective of this study is to develop a climatology and investigate the variability of snow depth over the Eurasian continent from 1966 to 2012. In addition, we analyse the effects of topography and climate factors (i.e., air temperature and snowfall) on the changes in snow depth over the study area. This study is unique in snow cover analysis using the most comprehensive daily snow depth observational network at continental scale over Eurasia. The dataset and methodology are described in Section 2 with the results, discussion, and conclusions presented in Sections 3, 4, and 5, respectively."

(10) P7, L.10: replaced semi-colon by a period.

(11) P7, L.11: replaced semi-colon by a period.

(12) P7, L.11-12: replaced the sentence with "The World Meteorological Organization common approach to calculate anomalies is based on a 30-years climate normal period (IPCC, 2013). In our study, 1971-2000 was used as the normal period. To ensure data continuity, stations with less than 20-years data during the 1971-2000 period were excluded."

(13) P7, L.12: deleted "and".

(14) P7, L.14: deleted a space before "After".

(15) P8, L.6: inserted a space in "than 20".

(16) P8, L.10: replaced semi-colon by a period at the end of the sentence.

(17)P10, L.6: inserted "which" before "were".

(18)P11, L.16: replaced "increased" with "increasing".

(19)P12, L.1-7: deleted the paragraph.

(20)P12, L.30-P15, L.13: deleted the paragraphs.

(21)P15, L.16: inserted "4.1 Comparisons with previous results" before the paragraph.

(22)P15, L.23: inserted "a" after "using".

(23)P16, L.5: replaced "stations" with "station"; inserted "a" after "also".

(24)P16, L.11: replaced "cm/yr" with "cm yr$^{-1}$".

(25)P16, L.13: replaced the comma after "Russia" with a semi-colon.

(26)P16, L.14: replaced "cm/yr" with "cm yr$^{-1}$".

(27)P16, L.16-25: deleted the paragraph.

(28)P17, L.4: inserted "Phase 5" after "Project"; replaced the comma after "2014" with a semi-colon.

(29)P17, L.5: replaced "Qinghai-Tibetan Plateau" with "TP"; deleted the last "the".

(30)P17, L.8: inserted these paragraphs before the paragraph:

[revised manuscript text omitted]

(31) P17, L.18: replaced "Qinghai-Tibetan Plateau" with "TP".

(32) P17, L.24: inserted "and" before "freshwater"; inserted "are" before "from".

(33) P17, L.29: replaced "forest" with "forests".

(34) P17, L.30: replaced "plants" with "plant".

(35) P18, L.30: inserted "especially heavy snowfall" after "snowfall"

(36) P20, L.22: replaced "anomalies" with "anomaly".

(37) P20, L.24: replaced "estimated" with "estimate"; inserted "value" after "estimate".

(38) P21, L.7: inserted "at" after "October".

(39) P21, L.14: replaced "cm/yr" with "cm yr$^{-1}$".

(40) P22, L.11-12: deleted the reference.

(41) P22, L.17-18: deleted the reference.

(42) P22, L.21: inserted a new reference "Brown, R. D. and Robinson, D. A.: Northern Hemisphere spring snow cover variability and change over 1922-2010 including an assessment of uncertainty, The Cryosphere, 5, 219-229, 2011."

(43) P23, L.19-20: deleted the reference.

(44) P24, L.8-9: deleted the reference.

(45) P25, L.28: replaced "suiface" with "surface".

(46) P26, L.21-22: replaced the doi number by updated volume and page numbers "44, 2873-2895, 2015".

(47) P26, L.27-29: deleted the reference.

(48) P28, L.1: inserted a new reference "Xu, W., Ma, L., Ma, M., Zhang, H., and Yuan, W.: Spatial-temporal variability of snow cover and depth in the Qinghai-Tibetan Plateau, J. Climate, 30, 1521-1533, 2017."

(49) P29, Table A1: replaced the value of $d_l$ and $d_u$ in April with "1.3034" and "1.3871", respectively.

(50) P29, L.5: replaced "cm/yr" with "cm yr$^{-1}$".

(51) P29, L.6: inserted "the" before "Dikson".

(52) P30, Table A1: replaced the value of $d_l$ and $d_u$ with "1.3034" and "1.3871", respectively.

(53) P30, L.1: replaced "cm/yr" with "cm yr$^{-1}$".

(54) P31, L.3-5: replaced "trianles" with "circles".

(55) P31, L.6: inserted "The abbreviations of countries represented separately: ARM-Armenia, AZE-Azerbaijan, BLR-Belarus, EST-Estonia, GEO-Georgia, KAZ-Kazakhstan, KGZ-Kyrgyzstan, LTU-Lithuania, LVA-Latvia, MDA-Moldova, TJK-Tajikistan, TKM-Turkmenistan, UKR- Ukraine, UZB-Uzbekistan." after the last sentence.

(56) P37, L.4: inserted "Linear regression was only shown when the rate of change was at the 95% level." after the last sentence.

(57) P39: replaced figure 8 with a new figure

(58) P39, L.8: inserted "and c" after "b".

(59) P39, L.9: replaced "c" with "d".

[revised manuscript text omitted]

---

## Author Response (AR5)

Dear Dr. Guillaume Chambon,

Thank you very much for your comments on this manuscript. Your help and guidance during this long review process is much appreciated. We have followed your instructions and made necessary changes. Please see details for our reply for each item. Please let us know if you have further concerns.

Best wishes,

Tingjun Zhang

**Response to Editor**

1. P. 1, lines 24-26: unclear sentence.

Reply: We have modified the sentences:

"Previous studies by using in-situ measurements were mostly site-specific (Bulygina et al., 2009, 2011; Ma and Qin, 2012); data from satellite remote sensing may cover a large area or in global scale, but uncertainties remain large, even misleading (Zheng et al., 2015). In this study, we obtained snow depth data of ground-based measurements from 1814 stations across Eurasian continent from 1966 to 2012. The main objective of this study is to investigate the spatial and temporal changes and variabilities in snow depth over the Eurasian continent in this study.

2. P. 2, lines 6-7: unclear sentence.

Reply: We have modified the sentences:

"This study provides a baseline for snow depth climatology and snow depth changes, using in-situ measured snow depth data for investigating climate system changes over the Eurasian continent."

3. P. 3, line 9: replace "could" by "can".

Done.

4. P. 3, line 22: delete "on".

Done.

5. P. 3, line 23: replace "which" by "and".

Done.

6. P. 3, lines 23-24: not very clear: try to be more specific. Is this sentence connected to the previous one? Is so, make it more evident.

Reply: we have modified the sentences:

"Previous studies on snow depth have focused at local and regional scales over Russia (Ye et al., 1998; Kitaev et al., 2005; Bulygina et al., 2009, 2011; Brasnett, 1999) and the Tibetan Plateau (TP) (Li and Mi, 1983; Ma and Qin, 2012). These studies demonstrated that that annual mean snow depth has increased in northern Eurasia and the Arctic during the last 70 years (Ye et al., 1998; Kitaev et al., 2005; Callaghan et al., 2011a; Liston and Hiemstra, 2011). However, there are large regional differences at various scales (Bulygina et al., 2009, 2011; Ma and Qin, 2012; Stuefer et al., 2013; Terzago et al., 2014)."

7. P. 4, line 4: insert "In addition,".

Done.

8. P. 4, line 14: Why? Explain more clearly why this information is required.

Reply: We have deleted the sentence.

9. P. 4, line 26: delete "approximately"

Reply: Has been deleted.

10. P. 4, line 26: insert "Therefore," before "utilization".

Reply: We have deleted the sentence.

11. P. 4, line 27: This statement appears a bit strong. Consider toning down.

Reply: We have deleted the sentence.

12. P. 4, line 28: delete "accurate and".

Reply: Has been deleted.

13. P. 4, line 30-P. 5, line 2: unclear sentence.

Reply: We have modified the sentences:

"However, low-resolution satellite remote sensing data is used as input parameter, which can affect simulation accuracy and does not provide a sufficient time series length."

14. P. 5 line 7: insert "Hence," before "ground-based", delete "and"

Reply: Has been modified.

15. P. 5, line 8: insert "most" before "accurate", replace "depth" by "depths".

Reply: Has been modified.

16. P. 5, line 10: replace "simulation" by "simulations".

Reply: Has been replaced.

17. P. 5, lines 11-12: To connect with what is said above, you should already briefly describe here the type of data considered in the study.

Reply: We have modified the paragraph:

"The objective of this study is to (i) establish snow depth climatology (1971-2000), (ii) investigate snow depth variability at various scales from 1966 to 2012, and (iii) analyze factors controlling snow depth distribution and changes over Eurasian continent. Snow depth data used in this study are daily or 10-day interval ground-based measurements from 1814 stations. Detailed description of in-situ measurements and methodology are described in Section 2 with major results, discussions, and conclusions presented in Sections 3, 4, and 5, respectively."

18. P. 6, lines 7-9: This sentence is not useful here. Consider moving it - if not too redundant - in the introduction. Only very few results concerning SWE are presented in the sequel. Why?

Reply: We have deleted the analysis of the relation between SWE and climate factors in the sequel because there are similar results in snow depth.

19. P. 6, lines 18-19: Explain briefly why.

Reply: We have deleted the sentence.

20. P. 6, lines 24-25: This should probably be said earlier, after lines 5-6.

Reply: We have moved the sentence after lines 5-6:

"…Snow depth was measured every 10 m in the forest and every 20 m in open terrain. The final snow depth at each station was determined as the average of all measurements in each snow course survey (Bulygina et al., 2011)."

21. P. 7, lines 17-18: Seems contradictory with what is said at the beginning of the section: please be more precise here and there.

Reply: We have deleted the contradictory sentence at the beginning of the section and added the period here:

"Procedures and techniques for measuring snow depth may have changed over the course of station history before the 1950s."

22. P. 8, lines 2-5: Note very clear. Is the annual mean snow depth a data computed for each year, or a single value averaged over the whole 1966-2012 period?

Reply: We have modified the definition:

"(3) Annual mean snow depth: annual mean snow depth was calculated as an arithmetic sum of monthly mean snow depth divided by the number of available snow months for each snow year.
(6) Long-term mean annual snow depth: it was averaged from annual mean snow depths over the 1971-2000 period. "

23. P. 8, lines 6-9: Idem. It is not very clear here whether this annual mean max snow depth is an average value over the 1966-2012 period or a data available for each year (and averaged over what in this case?). In fact, both quantities are used in the sequel, so you should be more precise here (and use different denominations?).

Reply: We have modified the definition:

"(4) Annual maximum snow depth: the annual maximum snow depth is defined as the maximum daily snow depth within each snow year.
(7) Long-term mean maximum snow depth: it was averaged from annual maximum snow depth over the 1971-2000 period."

24. P. 8, lines 15-16: Unnecessary sentence.

Reply: We have deleted the sentence.

25. P. 8, lines 23-24: Was it calculated on the low-pass filtered signal or the raw data?

Reply: We have modified the sentence:

"The linear trend coefficient of the raw snow depth was calculated to represent the rate of change at each station."

26. P. 8, line 26: Which partial correlation coefficients? They have not yet been introduced at this stage.

Reply: We have added as following:

"The Student's t-test was used to assess statistical significance of the slope in the linear regression analysis and the partial correlation coefficients of snow depth, air temperature and snowfall,…"

27. P. 8, line 30: consider replacing by: "...trends in the time series of the corrected data were recalculated"

Reply: Has been replaced.

28. P. 9, line 1: replace "were" by "are"

Reply: Has been replaced.

29. P. 9, line 7: Unclear terminology: could be confused with the "annual maximum mean snow depth" defined earlier.

Reply: We have modified it as "The maximum value of 109.3"

30. P. 9, line 11: Why do you use past tense in all this description? Consider using present tense instead in the whole "Results" section.

Reply: We have used present tense in the whole "Results" section.

31. P. 9, line 29: replace "however," by "in contrast".

Reply: Has been replaced.

32. P. 10, line 3: higher than what?

Reply: We have replaced "higher" by "high".

33. P. 10, line 5: insert "in" and "they".

Reply: Has been inserted.

34. P. 10, line 19: replace by "due mainly to snow disappearance..."

Reply: Has been replaced.

35. P. 11, line 25: replace by "some regions south of...".

Reply: Has been replaced.

36. P. 12, line 8: replaced by "p"

Reply: Has been replaced.

37. P. 12, line 21: Consider adding here a brief summary of the more significant trends observed in the data.

Reply: We have added:

"Overall, it presents significant increasing trends in annual mean snow depth, annual maximum snow depth and monthly mean snow depth over Eurasia, especially in European Russia, south of Siberia, the northern Xinjiang Autonomous Region of China, and Northeast China. Compared with regions south of 50°N, changes in snow depth are more significant over regions north of 50°N."

38. P. 12, line 28: replace by "the present".

Reply: Has been replaced.

39. P. 12, line 29: suppress "that".

Reply: Has been deleted.

40. P. 13, line 4: replace by "the present" or "our"

Reply: Has been replaced.

41. P. 13, line 20: replace by "the present".

Reply: Has been replaced.

42. P. 13, line 21: missing words here?

Reply: We have added:

"…however, we found winter monthly mean snow depth increased at a rate of 0.42 cm yr$^{-1}$ in southern Siberia during the period from 1966 to 2012."

43. P. 13, line 29: No data of SWE have been presented up to now. Hence, focus only on snow depth?

Reply: Yes, we focus only on snow depth, and have deleted the analysis of SWE.

44. P. 13, line 30: not clear: try using similar geographic references as those used in the description of results.

Reply: We have deleted the second "the western portion of".

45. P. 13, line 30: What is "this" here?

Reply: We have replaced "This" by "The similar result".

46. P. 14, line 22: Not clear. Where is this calculation procedure described before?

Reply: We described the calculation procedure in the definition of daily snow depth:

"(1) Daily snow depth: we defined a snow cover day with snow depth equal to or greater than 0 cm according to the standard method for deriving monthly mean snow depth based on the World Meteorological Organization (WMO) climatological products (Ma and Qin, 2012). Daily snow depth is the original in-situ measurements of snow depth."

47. P. 14, lines 25-27: Unclear sentence, should be rephrased.

Reply: We have modified the sentence:

"Snow depths were averaged at each 200 m elevation band."

48. P. 14, line 30-P.15 line1: unclear formulation.

Reply: We have modified the sentence:

"Annual mean snow depths increase with elevation and reach to the peak at 1600 m."

49. P. 15, line 4: This whole paragraph is unclear and would need to be rephrased.

Reply: We have modified the paragraph:

"There is a negative correlation between annual mean snow depth and elevation across the Eurasian continent (Fig. 8b); with every 100 m increases in elevation, annual mean snow depth decreases by ~0.5 cm (p≤0.05). Annual mean snow depth is less than 1 cm in regions with elevation greater than 2000 m because a snow depth of 0 cm was used to calculate the annual mean snow depth. Therefore, although the TP is at a high elevation, the shallow annual mean snow depth results in a generally negative correlation between snow depth and elevation across the Eurasian continent. Snow depths were averaged at each 200 m elevation band. Annual mean snow depths are deeper in the lower elevation bands (between 0 and 600 m) across the former USSR (Fig. 8c). However, there are shallow annual mean snow depth between 600 and 1000 m due mainly to forest effect. Annual mean snow depths increase with elevation and reach to the peak at 1600 m. Annual mean snow depths show marked decrease in the highest elevation bands (2600~2900 m). There are only two stations in this band and more annual mean snow depth difference between the two stations because of terrain and climate factors. Snow is deeper in three elevation bands across China: 200~1000 m, 1600~1800 m and 2400~2600 m. Greater snow depth is attributed to more snowfall and severe cold weather in these regions. An increasing trend with elevation presents above 2600 m on the TP."

50. P. 15, line 9: You should define continentality here.

Reply: We have added the definition of continentality:

"Continentality is a measure of the difference between continental and marine climates. It is roughly a measure of distance from oceans. Continentality affects precipitation, thus determines snowfall rate and snow depth."

51. P. 15, line 10: Appears contradictory with the previous sentence and the significant positive correlation.

Reply: We have modified the sentences:

"Although there is a statistically significant positive relationship between annual mean snow depth and continentality over the Eurasian continent, the Goodness of Fit is only 1% (Fig. 8d). This indicates that the continentality may not be an important driving factor of annual mean snow depth distribution compared with latitude and elevation over Eurasia, especially on the TP."

52. P. 15, lines 11-16: What is the relation between these last 2 sentences and the influence of continentality? Shouldn't these sentences be rather moved to section 4.3 about climate factors?

Reply: We have deleted the sentences.

53. P. 16, line 1: remove "snow depth" here, since you cannot compare snow depth to snow depth!

Reply: Has been deleted.

54. P. 16, line 1: Where is this evident?

Reply: We have deleted the sentence.

55. P. 16, line 11: unclear statement: what is the "entire density"?

Reply: We have replaced "entire" by "bulk".

56. P. 16, line 12: How is "heavy snowfall" defined?

Reply: We have added the definition of heavy snowfall:

"In addition, there are similar inter-annual variations in snowfall and heavy snowfall (daily snowfall amount is between 5-10 mm)."

57. P. 16, line 18: This is the first time that data concerning SWE are presented. You should at least also present, and comment, SWE data in Fig. 9 and 10.

Reply: We have deleted the analysis of the relation between SWE and climate factors in the sequel because there are similar results in snow depth.

58. P. 16, line 19: between what and what?

Reply: We have modified the sentences:

"A significant negative correlation ($p \leqslant 0.05$) between annual mean snow depth and air temperature is present in most areas of European Russia and southern Siberia (Fig 11a). However, there is no statistically significant correlation among them in northern Siberia."

59. P. 17, line 6: Where is this demonstrated? This conclusion does not seem to by fully supported by the presented results. Hence, consider either expanding the argument or toning down.

Reply: We have modified the sentences:

"The present study shows that there are similar inter-annual variations in annual mean snow depth and heavy snowfall, which implies that extreme snowfall may be the main reason for snow thickening."

59. P. 17, line 6: Idem: none of the presented results directly concern atmospheric circulation.

Reply: We have deleted.

60. P. 17, line 17: replace by "the present".

Reply: Has been replaced.

61. P. 43, line 5: was referred to as the "95% confidence level" in previous figures: be consistent.

Reply: Has been replaced.

**List of relevant changes**

According to editor's comments, we have made relevant changes in this manuscript. The main changes are followed:

1.  We have reorganized the objective of our study:

    "The objective of this study is to (i) establish snow depth climatology (1971-2000), (ii) investigate snow depth variability at various scales from 1966 to 2012, and (iii) analyze factors controlling snow depth distribution and changes over Eurasian continent. Snow depth data used in this study are daily or 10-day interval ground-based measurements from 1814 stations. Detailed description of in-situ measurements and methodology are described in Section 2 with major results, discussions, and conclusions presented in Sections 3, 4, and 5, respectively."

2.  All of the snow depth variable have been redefined and have replaced all the words in the manuscript:

    "(1) Daily snow depth: we defined a snow cover day with snow depth equal to or greater than 0 cm according to the standard method for deriving monthly mean snow depth based on the World Meteorological Organization (WMO) climatological products (Ma and Qin, 2012). Daily snow depth is the original in-situ measurements of snow depth.
    (2) Monthly mean snow depth: monthly mean snow depth was computed as an arithmetic sum of daily snow depth divided by the number of days with snow on the ground within each month.
    (3) Annual mean snow depth: annual mean snow depth was calculated as an arithmetic sum of monthly mean snow depth divided by the number of available snow months for each snow year.
    (4) Annual maximum snow depth: the annual maximum snow depth was defined as the maximum daily snow depth within each snow year.
    (5) Long-term mean monthly snow depth: it was averaged from each monthly mean snow depth over the 1971-2000 period.
    (6) Long-term mean annual snow depth: it was averaged from annual mean snow depths over the 1971-2000 period.
    (7) Long-term mean maximum snow depth: it was averaged from annual maximum snow depth over the 1971-2000 period."

3.  The data of long-term mean annual snow depth, long-term mean maximum snow depth and long-term mean monthly snow depth from 1971 through 2000 are used to reanalyze the climatology of snow depth. This is because the analysis of the anomalies of annual mean snow depth, annual maximum snow depth and monthly mean snow depth from 1966 through 2012 with respect to the 1971-2000 mean across the Eurasian continent.

4. We focus only on snow depth in this manuscript, therefore, we have deleted the analysis of the relation between SWE and climate factors in the sequel because there are similar results in snow depth.

5. We have added a brief summary of the more significant trends observed in the "3.2 Variability of Snow Depth" section:

[revised manuscript text omitted]

---

## Author Response (AR6)

**Response to Editor**

1. P. 4, line 30: insert "However,"

Done.

2. P. 5, line 2: insert "In addtion,"

Done.

3. P. 12, line 6: replace by "the data present".

Done.

4. P. 13, line 29: Here and in what follows, shouldn't it be the "long-term mean annual snow depth"? Please check (as well as in the corresponding figures captions).

Reply: it is long-term mean annual snow depth, we have modified.

5. P. 14, line 12: This statement is still not clear to me: Why is a snow depth of 0cm used to calculate the mean snow depth above 2000 m? Where is this explained?

Reply: we defined a snow cover day with snow depth equal to or greater than 0 cm according to the standard method for deriving monthly mean snow depth based on the World Meteorological Organization (WMO) climatological products, which is stated in "Data and Methodology" section. We have deleted the sentence.

6. P.14, line 15: insert "In Fig. 8c,"

Reply: Has been inserted.

7. P.14, line 15: replace by "are averaged over 200 m elevation bands".

Done.

8. P.14, line 16: replace by "higher".

Done.

9. P.14, line 19: insert "then"

Reply: Has been inserted.

10. P.14, line 21: which band? More than what?

Reply: We have inserted "(2600~3000 m)" after band, replaced "more annual mean snow depth difference" by "long-term mean annual snow depths are very different".

11. P. 14, line 22: insert "In China".

Done.

12. P. 14, line 25: Replace by "is present". Furthermore, isn't this statement is contradiction with what is said above: "although the TP is at high elevation, the shallow annual mean snow depth results in a generally negative correlation between snow depth and elevation across the Eurasian continent"?

Reply: Has been inserted. What is said above indicates shallow long-term mean annual snow depth on the TP results in long-term mean annual snow depth decreases with elevation on the whole continent scale. However, this statement here means that there is a positive relationship between long-term mean annual snow depth and elevation above 2600 m when only considers snow cover on the TP.

13. P. 14, lines 26-27: You could more simply say: "Continentality is roughly a measure of distance from oceans."

Reply: Has been modified.

14. P. 15 line 11: Here also, shouldn't it be the "long-term mean annual snow depth"?

Reply: Has been modified.

15. P. 15, line 17: replace by "corresponds to".

Done.

16. P. 15, lines 19-20: Consider rephrasing as: "Overall, the mean annual long-term air temperature, snowfall and snow depth display increasing trends..."

Reply: Has been rephrased.

17. P. 15, line 21: replace by "fall".

Done.

18. P. 15, line 28: please further clarify this definition

Reply: We have added the definition:

[revised manuscript text omitted]